# The Sample-Communication Complexity Trade-off in Federated Q-Learning

**Sudeep Salgia**
Carnegie Mellon University
`ssalgia@andrew.cmu.edu`

**Yuejie Chi**
Carnegie Mellon University
`yuejiechi@cmu.edu`

## Abstract

We consider the problem of Federated Q-learning, where $M$ agents aim to collaboratively learn the optimal Q-function of an unknown infinite horizon Markov Decision Process with finite state and action spaces. We investigate the trade-off between sample and communication complexity for the widely used class of intermittent communication algorithms. We first establish the converse result, where we show that any Federated Q-learning that offers a linear speedup with respect to number of agents in sample complexity needs to incur a communication cost of at least $\Omega(\frac{1}{1-\gamma})$, where $\gamma$ is the discount factor. We also propose a new Federated Q-learning algorithm, called Fed-DVR-Q, which is the first Federated Q-learning algorithm to simultaneously achieve order-optimal sample and communication complexities. Thus, together these results provide a complete characterization of the sample-communication complexity trade-off in Federated Q-learning.

## 1 Introduction

Reinforcement Learning (RL) [Sutton and Barton, 2018] refers to an online sequential decision making paradigm where the learning agent aims to learn an optimal policy, i.e., a policy that maximizes the long-term reward, through repeated interactions with an unknown environment. RL finds applications across a diverse array of fields including, but not limited to, autonomous driving, games, recommendation systems, robotics and Internet of Things (IoT) [Kober et al., 2013, Yurtsever et al., 2020, Silver et al., 2016, Lim et al., 2020].

The primary hurdle in RL applications is often the high-dimensional nature of the decision space that necessitates the learning agent to have to access to an enormous amount of data in order to have any hope of learning the optimal policy. Moreover, the sequential collection of such an enormous amount of data through a single agent is extremely time-consuming and often infeasible in practice. Consequently, practical implementations of RL involve deploying multiple agents to collect data in parallel. This decentralized approach to data collection has fueled the design and development of distributed or federated RL algorithms that can collaboratively learn the optimal policy without actually transferring the collected data to a centralized server. Such a federated approach to RL, which does not require the transfer of local data, is gaining interest due to lower bandwidth requirements and lower security and privacy risks. In this work, we focus on federated variants of Q-learning algorithms where the agents collaborate to directly learn the optimal Q-function without forming an estimate of the underlying unknown environment.

A particularly important aspect of designing Federated RL algorithms, including Federated Q-learning algorithms, is to address the natural tension between sample and communication complexity. At one end of the spectrum lies the naïve approach of running a centralized algorithm with optimal sample complexity after transferring and combining all the collected data at a central facility/server. Such an approach trivially achieves the optimal sample complexity while suffering from a very high and infeasible communication complexity. On the other hand, several recently proposed

38th Conference on Neural Information Processing Systems (NeurIPS 2024).

algorithms [Khodadadian et al., 2022, Woo et al., 2023] operate in more practical regimes, offering significantly lower communication complexities as compared to the naïve approach at the cost of sub-optimal sample complexities. These results suggest the existence of underlying trade-off between sample and communication complexities of Federated RL algorithms. The primary goal of this work is to better understand this trade-off in context of Federated Q-learning by investigating these following fundamental questions:

- *Fundamental limit of communication: What is the minimum amount of communication required by a federated Q-learning algorithm to achieve any statistical benefit of collaboration?*
- *Optimal algorithm design: How does one design a federated Q-Learning algorithm that simultaneously offers optimal order sample and communication complexity guarantees i.e., operates on the optimal frontier of sample-communication complexity trade-off?*

## 1.1 Main Results

We consider a setup where $M$ distributed agents collaborate to learn the optimal Q-function of an infinite horizon Markov Decision Process which is defined over a finite state space $\mathcal{S}$ and a finite action set $\mathcal{A}$, and has a discount factor of $\gamma \in (0, 1)$. We consider a commonly considered setup in federated learning called the intermittent communication setting, where the clients intermittently share information among themselves with the help of a central server. In this work, we provide a complete characterization of the trade-off between sample and communication complexity under the aforementioned setting by providing answers to both the questions. The main result of this work is twofold and is summarized below.

- *Fundamental bounds on communication complexity of Federated Q-learning*: We establish lower bounds on the communication complexity of Federated Q-learning, both in terms of number of communication rounds and the overall number of bits that need to be transmitted in order to achieve any speed up in convergence with respect to the number of agents. Specifically, we show that in order for an intermittent communication algorithm to obtain *any* benefit of collaboration, i.e., *any* order of speed up w.r.t. the number of agents, the number of communication rounds must be least $\Omega(\frac{1}{(1-\gamma)\log^2 N})$ and the number of *bits* sent by each agent to the server must be least $\Omega(\frac{|\mathcal{S}||\mathcal{A}|}{(1-\gamma)\log^2 N})$, where $N$ denotes the number of samples taken by the algorithm for each state-action pair.
- *Achieving the optimal sample-communication complexity trade-off*: We propose a new Federated Q-Learning algorithm called Federated Doubly Variance Reduced Q Learning, Fed-DVR-Q for short, that simultaneously achieves optimal order of sample complexity and the minimal order of communication as dictated by the lower bound. We show that Fed-DVR-Q learns an $\varepsilon$-optimal Q-function in the $\ell_\infty$ sense with $\tilde{\mathcal{O}}\left(\frac{|\mathcal{S}||\mathcal{A}|}{M\varepsilon^2(1-\gamma)^3}\right)$ i.i.d. samples from the generative model at each agent while incurring a total communication cost of $\tilde{\mathcal{O}}\left(\frac{|\mathcal{S}||\mathcal{A}|}{(1-\gamma)}\right)$ *bits* per agent across $\tilde{\mathcal{O}}\left(\frac{1}{(1-\gamma)}\right)$ rounds of communication. Thus, Fed-DVR-Q not only improves upon both the sample and communication complexities of existing algorithms, but also is the *first algorithm* to achieve both order-optimal sample and communication complexities (See Table 1 for a comparison).

## 1.2 Related Work

**Single agent Q-Learning.** Q-Learning has been extensively studied in the single-agent setting in terms of both its asymptotic convergence [Jaakkola et al., 1993, Tsitsiklis, 1994, Szepesvári, 1997, Borkar and Meyn, 2000] and its finite-time sample complexity in both synchronous [Even-Dar and Mansour, 2004, Beck and Srikant, 2012, Wainwright, 2019a, Chen et al., 2020, Li et al., 2023] and asynchronous settings [Chen et al., 2021b, Li et al., 2023, 2021, Qu and Wierman, 2020].

**Distributed RL.** There has also been a considerable effort towards developing distributed and federated RL algorithms. The distributed variants of the classical TD learning algorithm have been investigated in a series of studies [Chen et al., 2021c, Doan et al., 2019, 2021, Sun et al., 2020, Wai, 2020, Wang et al., 2020, Zeng et al., 2021b]. The impact of environmental heterogeneity in federated TD learning was studied in Wang et al. [2023]. A distributed version of actor-critic

| Algorithm/Reference | Number of Agents | Sample Complexity | Communication Complexity |
|---|---|---|---|
| Q-learning [Li et al., 2023] | 1 | $\frac{|\mathcal{S}||\mathcal{A}|}{(1-\gamma)^4\varepsilon^2}$ | N/A |
| Variance Reduced Q-learning [Wainwright, 2019b] | 1 | $\frac{|\mathcal{S}||\mathcal{A}|}{(1-\gamma)^3\varepsilon^2}$ | N/A |
| Fed-SynQ [Woo et al., 2023] | $M$ | $\frac{|\mathcal{S}||\mathcal{A}|}{M(1-\gamma)^5\varepsilon^2}$ | $\frac{M}{1-\gamma}$ |
| Fed-DVR-Q (This work) | $M$ | $\frac{|\mathcal{S}||\mathcal{A}|}{M(1-\gamma)^3\varepsilon^2}$ | $\frac{1}{1-\gamma}$ |
| Lower bound ([Azar et al., 2013], This work) | $M$ | $\frac{|\mathcal{S}||\mathcal{A}|}{M(1-\gamma)^3\varepsilon^2}$ | $\frac{1}{1-\gamma}$ |

Table 1: Comparison of sample and communication complexity of various single-agent and Federated Q-learning algorithms for learning an $\varepsilon$-optimal Q-function under the synchronous setting. We hide logarithmic factors and burn-in costs for all results for simplicity of presentation. In the above table, $\mathcal{S}$ and $\mathcal{A}$ represent state and action spaces respectively and $\gamma$ denotes the discount factor. We report the communication complexity only in terms of number of rounds as other algorithms assume transmission of real numbers and hence do not report bit level costs. For the lower bound, Azar et al. [2013] and this work establish the bound for sample and communication complexity respectively.

algorithms was studied by Shen et al. [2023] where the authors established convergence of their algorithm and demonstrated a linear speed up in the number of agents in their sample complexity bound. Chen et al. [2022] proposed a new distributed actor-critic algorithm which improved the dependence of sample complexity on the error $\varepsilon$ and incurs a communication cost of $\tilde{\mathcal{O}}(\varepsilon^{-1})$. Chen et al. [2021a] have proposed a communication efficient distributed policy gradient algorithm and have analyzed its convergence and established a communication complexity of $\mathcal{O}(1/(M\varepsilon))$. Xie and Song [2023] adopts a distributed policy optimization perspective, which is different from the Q-learning paradigm considered in this work. Moreover, the algorithm in Xie and Song [2023] obtains a linear communication cost, which is worse than that obtained in our work. Similarly, Zhang et al. [2024] focuses on on-policy learning and incurs a communication cost that depends polynomially on the required error $\varepsilon$. Several other studies [Yang et al., 2023, Zeng et al., 2021a, Lan et al., 2024] have also developed and analyzed other distributed/federated variants of the classical natural policy gradient method [Kakade, 2001]. Assran et al. [2019], Espeholt et al. [2018], Mnih et al. [2016] have developed distributed algorithms to train deep RL networks more efficiently.

**Distributed Q-learning.** Federated Q-learning has been explored relatively recently. Khodadadian et al. [2022] proposed and analyzed a federated Q-learning algorithm in the asynchronous setting with a sample complexity of $\tilde{\mathcal{O}}\left(\frac{|\mathcal{S}|^2}{M\mu_{\min}^5(1-\gamma)^9\varepsilon^2}\right)$, where $\mu_{\min}$ is the minimum entry of the stationary state-action occupancy distribution of the sample trajectories over all agents. Jin et al. [2022] study the impact of environmental heterogeneity across clients in Federated Q-learning. They propose an algorithm where the local environments are different at each client but each client knows their local environment. Under this setting, they propose an algorithm that achieves a sample and communication complexity of $\mathcal{O}(\frac{1}{(1-\gamma)^3\varepsilon})$ and $\mathcal{O}(\frac{1}{(1-\gamma)^3\varepsilon})$ rounds respectively. Woo et al. [2023] proposed new algorithms with improved analysis for Federated Q-learning under both synchronous and asynchronous settings. Their proposed algorithm achieves a sample complexity and communication complexity of $\tilde{\mathcal{O}}(\frac{|\mathcal{S}||\mathcal{A}|}{M(1-\gamma)^5\varepsilon^2})$ and $\tilde{\mathcal{O}}(\frac{M|\mathcal{S}||\mathcal{A}|}{1-\gamma})$ real numbers respectively under the synchronous setting and that of $\tilde{\mathcal{O}}(\frac{1}{M\mu_{\text{avg}}(1-\gamma)^5\varepsilon^2})$ and $\tilde{\mathcal{O}}\left(\frac{M|\mathcal{S}||\mathcal{A}|}{1-\gamma}\right)$ real numbers respectively under the asynchronous setting. Here, $\mu_{\text{avg}}$ denotes the minimum entry of the average stationary state-action occupancy distribution of all agents. In a follow up work, Woo et al. [2024] propose a Federated Q-learning for offline RL in finite horizon setting and establish a sample and communication complexity of $\tilde{\mathcal{O}}(\frac{H^7|\mathcal{S}|C_{\text{avg}}}{M\varepsilon^2})$ and $\tilde{\mathcal{O}}(H)$, where $H$ denotes the length of the horizon and $C_{\text{avg}}$ denotes the average single-policy concentrability coefficient of all agents.

**Accuracy-Communication Trade-off in Federated Learning.** The trade-off between communication complexity and accuracy (equivalently, sample complexity) has been studied in various federated and distributed learning problems, including stochastic approximation algorithms for convex optimization. Duchi et al. [2014], Braverman et al. [2016] establish the celebrated inverse linear relationship between the error and the communication cost the problem of distributed mean estimation. Similar trade-off for distributed stochastic optimization, multi-armed bandits and linear bandits has been studied and established across numerous studies [Woodworth et al., 2018, 2021, Tsitsiklis and Luo, 1987, Shi and Shen, 2021, Salgia and Zhao, 2023].

## 2 Problem Formulation and Preliminaries

In this section, we provide a brief background of Markov Decision Processes, outline the performance measures for Federated Q-learning algorithms and describe the class of intermittent communication algorithms considered in this work.

### 2.1 Markov Decision Processes

In this work, we focus on an infinite-horizon Markov Decision Process (MDP), denoted by $\mathcal{M}$, over a state space $\mathcal{S}$ and an action space $\mathcal{A}$ and with a discount factor $\gamma \in (0, 1)$. Both the state and action spaces are assumed to be finite sets. In an MDP, the state $s$ evolves dynamically under the influence of actions based on a probability transition kernel, $P : (\mathcal{S} \times \mathcal{A}) \times \mathcal{S} \to [0, 1]$. The entry $P(s'|s, a)$ denotes the probability of moving to state $s'$ when an action $a$ is taken in the state $s$. An MDP is also associated with a deterministic reward function $r : \mathcal{S} \times \mathcal{A} \to [0, 1]$, where $r(s, a)$ denotes the immediate reward obtained for taking the action $a$ in the state $s$. Thus, the transition kernel $P$ along with the reward function $r$ completely characterize an MDP. In this work, we consider the synchronous setting, where each agent has access to an independent generative model or simulator from which they can draw independent samples from the unknown underlying distribution $P(\cdot|s, a)$ for each state-action pair $(s, a)$ [Kearns and Singh, 1998].

A policy $\pi : \mathcal{S} \to \Delta(\mathcal{A})$ is a rule for selecting actions across different states, where $\Delta(\mathcal{A})$ denotes the simplex over $\mathcal{A}$ and $\pi(a|s)$ denotes the probability of choosing action $a$ in a state $s$. Each policy $\pi$ is associated with a state value function and a state-action value function, or the Q-function, denoted by $V^\pi$ and $Q^\pi$ respectively. $V^\pi$ and $Q^\pi$ measure the expected discounted cumulative reward attained by $\pi$ starting from a particular state $s$ and state-action pair $(s, a)$ respectively. Mathematically, $V^\pi$ and $Q^\pi$ are given as

$$V^\pi(s) := \mathbb{E}\left[\sum_{t=0}^{\infty} \gamma^t r(s_t, a_t) \middle| s_0 = s\right]; \quad Q^\pi(s, a) := \mathbb{E}\left[\sum_{t=0}^{\infty} \gamma^t r(s_t, a_t) \middle| s_0 = s, a_0 = a\right], \quad (1)$$

where $a_t \sim \pi(\cdot|s_t)$ and $s_{t+1} \sim P(\cdot|s_t, a_t)$ for all $t \geq 0$. The expectation is taken w.r.t. the randomness in the trajectory $\{s_t, a_t\}_{t=1}^{\infty}$. Since the rewards lie in $[0, 1]$, it follows immediately that both the value function and Q-function lie in the range $[0, \frac{1}{1-\gamma}]$.

An optimal policy $\pi^\star$ is a policy that maximizes the value function uniformly over all the states and it has been shown that such an optimal policy $\pi^\star$ always exists [Puterman, 2014]. The optimal value and Q-functions are those corresponding to that of an optimal policy $\pi^\star$ are denoted as $V^\star := V^{\pi^\star}$ and $Q^\star := Q^{\pi^\star}$ respectively. The optimal Q-function, $Q^\star$, is also the unique fixed point of the Bellman operator $\mathcal{T} : \mathcal{S} \times \mathcal{A} \to \mathcal{S} \times \mathcal{A}$, given by

$$(\mathcal{T}Q)(s, a) = r(s, a) + \gamma \cdot \mathbb{E}_{s' \sim P(\cdot|s, a)}\left[\max_{a' \in \mathcal{A}} Q(s', a')\right]. \quad (2)$$

Q-learning [Watkins and Dayan, 1992] aims to learn the optimal policy by first learning $Q^\star$ as the solution to the fixed point equation $\mathcal{T}Q = Q$ and then obtain a deterministic optimal policy via the maximization $\pi^\star(s) = \arg\max_a Q^\star(s, a)$.

Let $Z \in \mathcal{S}^{|\mathcal{S}||\mathcal{A}|}$ be a random vector whose $(s, a)^{\text{th}}$ coordinate is drawn from the distribution $P(\cdot|s, a)$, independently of all other coordinates. We define the random operator $\mathcal{T}_Z : (\mathcal{S} \times \mathcal{A}) \to (\mathcal{S} \times \mathcal{A})$ as

$$(\mathcal{T}_Z Q)(s, a) = r(s, a) + \gamma V(Z(s, a)), \quad (3)$$

where $V(s') = \max_{a' \in \mathcal{A}} Q(s', a')$. The operator $\mathcal{T}_Z$ can be interpreted as the sample Bellman Operator, where we have the relation $\mathcal{T}Q = \mathbb{E}_Z[\mathcal{T}_Z Q]$ for all Q-functions.

Lastly, the federated learning setup considered in this work consists of $M$ agents, where all the agents face a common, unknown MDP, i.e., the transition kernel and the reward functions are the same across agents, which is popularly known as the homogeneous setting. For a given value of $\varepsilon \in (0, \frac{1}{1-\gamma})$, the objective of agents is to collaboratively learn an $\varepsilon$-optimal estimate (in the $\ell_\infty$ sense) of the optimal Q-function of the unknown MDP.

## 2.2 Performance Measures

We measure the performance of a Federated Q-learning algorithm $\mathscr{A}$ using two metrics — sample complexity and communication complexity. For a given MDP $\mathcal{M}$, let $\widehat{Q}_{\mathcal{M}}(\mathscr{A}, N, M)$ denote the estimate of $Q^\star_{\mathcal{M}}$, the optimal Q-function of the MDP $\mathcal{M}$, returned by an algorithm $\mathscr{A}$, when given access to $N$ i.i.d. samples from the generative model for each $(s, a)$ pair at all the $M$ agents. The minimax error rate of the algorithm $\mathscr{A}$, denoted by $\mathsf{ER}(\mathscr{A}; N, M)$, is defined as

$$\mathsf{ER}(\mathscr{A}; N, M) := \sup_{\mathcal{M}=(P,r)} \mathbb{E}\left[\|\widehat{Q}_{\mathcal{M}}(\mathscr{A}, N, M) - Q^\star_{\mathcal{M}}\|_\infty\right], \tag{4}$$

where the expectation is taken over the samples and any randomness in the algorithm. Given a value of $\varepsilon > 0$, the sample complexity of $\mathscr{A}$, denoted by $\mathsf{SC}(\mathscr{A}; \varepsilon, M)$ is given as

$$\mathsf{SC}(\mathscr{A}; \varepsilon, M) := |\mathcal{S}||\mathcal{A}| \cdot \min\{N \in \mathbb{N} : \mathsf{ER}(\mathscr{A}; N, M) \le \varepsilon\}. \tag{5}$$

Similarly, we can also define a high-probability version for any $\delta \in (0, 1)$ as follows:

$$\mathsf{SC}(\mathscr{A}; \varepsilon, M, \delta) := |\mathcal{S}||\mathcal{A}| \cdot \min\{N \in \mathbb{N} : \Pr(\sup_{\mathcal{M}} \|\widehat{Q}_{\mathcal{M}}(\mathscr{A}, N, M) - Q^\star_{\mathcal{M}}\|_\infty \le \varepsilon) \ge 1 - \delta\}.$$

We measure the communication complexity of any federated learning algorithm both in terms of frequency of information exchange and total number of bits uploaded by the agents. For each agent $m$, let $C^m_{\mathsf{round}}(\mathscr{A}; N)$ and $C^m_{\mathsf{bit}}(\mathscr{A}; N)$ respectively denote the number of times agent $m$ sends a message to the server and the total number of bits uploaded by agent $m$ to the server when an algorithm $\mathscr{A}$ is run with $N$ i.i.d. samples from the generative model for each $(s, a)$ pair at each agent. The communication complexity of $\mathscr{A}$, when measured in terms of frequency of communication and total number of bits exchanged, is given by

$$\mathsf{CC}_{\mathsf{round}}(\mathscr{A}; N) := \frac{1}{M} \sum_{m=1}^{M} C^m_{\mathsf{round}}(\mathscr{A}; N); \quad \mathsf{CC}_{\mathsf{bit}}(\mathscr{A}; N) := \frac{1}{M} \sum_{m=1}^{M} C^m_{\mathsf{bit}}(\mathscr{A}; N), \tag{6}$$

respectively. Similarly, for a given value of $\varepsilon \in (0, \frac{1}{1-\gamma})$, we can also define $\mathsf{CC}_{\mathsf{round}}(\mathscr{A}; \varepsilon)$ and $\mathsf{CC}_{\mathsf{bit}}(\mathscr{A}; \varepsilon)$ based on when $\mathscr{A}$ is run to guarantee a minimax error of at most $\varepsilon$.

## 2.3 Intermittent Communication Algorithms

In this work, we consider a popular class of federated learning algorithms referred to as algorithms with intermittent communication. The intermittent communication setting provides a natural framework to extend single agent Q-learning algorithms to the distributed setting. As the name suggests, under this setting, the agents intermittently communicate with each other, sharing their updated beliefs about $Q^\star$. Between two communication rounds, each agent updates their belief about $Q^\star$ using stochastic fixed point iteration based on the locally available data, similar to a single agent setup. Such intermittent communication algorithms

---

**Algorithm 1:** A generic algorithm $\mathscr{A}$

1: Input : $T, R, \{\eta_t\}_{t=1}^T, \mathcal{C} = \{t_r\}_{r=1}^R, B$
2: Set $Q_0^m \leftarrow 0$ for all agents $m$
3: **for** $t = 1, 2, \ldots, T$ **do**
4:     **for** $m = 1, 2, \ldots, M$ **do**
5:         Compute $Q_{t-\frac{1}{2}}^m$ according to Eqn. 7
6:         Compute $Q_t^m$ according to Eqn. 8
7:     **end for**
8: **end for**
9: **return** $Q_T$

---

have been extensively studied and used to establish lower bounds on communication complexity of distributed stochastic convex optimization [Woodworth et al., 2018, 2021].

A generic Federated Q-learning algorithm with intermittent communication is outlined in Algorithm 1. It is characterized by the following five parameters: (i) total number of updates $T$; (ii) the number of communication rounds $R$; (iii) a step size schedule $\{\eta_t\}_{t=1}^{T}$; (iv) a communication schedule $\{t_r\}_{r=1}^{R}$; (v) batch size $B$. During the $t^{\text{th}}$ iteration, each agent $m$ computes $\{\widehat{\mathcal{T}}_{Z_b}(Q_{t-1}^m)\}_{b=1}^{B}$, a minibatch of sample Bellman operators at the current estimate $Q_{t-1}^m$, using $B$ samples from the generative model for each $(s, a)$ pair, and obtains an intermediate local estimate using the Q-learning update as follows:

$$Q_{t-\frac{1}{2}}^m = (1 - \eta_t)Q_{t-1}^m + \frac{\eta_t}{B} \sum_{b=1}^{B} \mathcal{T}_{Z_b}(Q_{t-1}^m). \tag{7}$$

Here $\eta_t \in (0, 1]$ is the step-size chosen corresponding to the $t^{\text{th}}$ time step. The intermediate estimates are averaged based on a communication schedule $\mathcal{C} = \{t_r\}_{r=1}^{R}$ consisting of $R$ rounds, i.e.,

$$Q_t^m = \begin{cases} \frac{1}{M} \sum_{j=1}^{M} Q_{t-\frac{1}{2}}^j & \text{if } t \in \mathcal{C}, \\ Q_{t-\frac{1}{2}}^m & \text{otherwise.} \end{cases} \tag{8}$$

In the above equation, the averaging step can also be replaced with any distributed mean estimation routine that includes compression to control the bit level costs. Without loss of generality, we assume that $Q_0^m = 0$ for all agents $m$ and $t_R = T$, i.e., the last iterates are always averaged. It is straightforward to note that the number of samples taken by an intermittent communication algorithm is $BT$, i.e, $N = BT$ and the communication complexity $\text{CC}_{\text{round}} = R$.

## 3  Lower Bound

In this section, we investigate the first of the two questions regarding the lower bound on communication complexity. The following theorem establishes a lower bound on the communication complexity of a Federated Q-learning algorithm with intermittent communication.

**Theorem 1.** *Assume that $\gamma \in [5/6, 1)$ and the state and action spaces satisfy $|\mathcal{S}| \geq 4$ and $|\mathcal{A}| \geq 2$. Let $\mathscr{A}$ be a Federated Q-learning algorithm with intermittent communication that is run for $T \geq \max\{16, \frac{1}{1-\gamma}\}$ steps with a step size schedule of either $\eta_t := \frac{1}{1+c_\eta(1-\gamma)t}$ or $\eta_t := \eta$ for all $1 \leq t \leq T$. If*

$$R = \text{CC}_{\text{round}}(\mathscr{A}; N) \leq \frac{c_0}{(1-\gamma)\log^2 N}; \text{ or } \text{CC}_{\text{bit}}(\mathscr{A}; N) \leq \frac{c_1 |\mathcal{S}||\mathcal{A}|}{(1-\gamma)\log^2 N}$$

*for some universal constants $c_0, c_1 > 0$ then, for all choices of communication schedule, batch size $B$, $c_\eta > 0$ and $\eta \in (0, 1)$, the minimax error of $\mathscr{A}$ satisfies*

$$\text{ER}(\mathscr{A}; N, M) \geq \frac{C_\gamma}{\log^3 N \sqrt{N}},$$

*for all $M \geq 2$ and $N = BT$. Here $C_\gamma > 0$ is a constant that depends only on $\gamma$.*

The above theorem states that in order for an intermittent communication algorithm to obtain *any* benefit of collaboration, i.e., for the error rate $\text{ER}(\mathscr{A}; N, M)$ to decrease w.r.t. number of agents, the number of communication rounds must be least $\Omega(\frac{1}{(1-\gamma)\log^2 N})$. This implies that any Federated Q-learning algorithm that offers order optimal sample complexity, and thereby also a linear speed up with respect to the number of agents, must have at least $\Omega(\frac{1}{(1-\gamma)\log^2 N})$ rounds of communication and transmit $\Omega(\frac{|\mathcal{S}||\mathcal{A}|}{(1-\gamma)\log^2 N})$ bits of information per agent. This characterizes the converse relation for the sample-communication tradeoff in Federated Q-learning. We would like to point out that our lower bound extends to the asynchronous setting as the assumption of i.i.d. noise corresponding to a generative model is a special case of Markovian noise observed in asynchronous setting.

The lower bound on the communication complexity of Federated Q-learning is a consequence of the bias-variance trade-off that governs the convergence of the algorithm. While a careful choice of step-sizes alone is sufficient to balance this trade-off in the centralized setting, the choice of communication schedule also plays an important role in balancing this trade-off in the federated setting. The local steps between two communication rounds induce a positive estimation bias that

depends on the standard deviation of the iterates and is a well-documented issue of "over-estimation" in Q-learning [Hasselt, 2010]. Since such a bias is driven by *local* updates, it does not reflect any benefit of collaboration. During a communication round, the averaging of iterates across agents allows the algorithm an opportunity to counter this bias by reducing the effective variance of the updates through averaging. In our analysis, we show that if the communication is infrequent, the local bias becomes the dominant term and averaging of iterates is insufficient to counter the impact of the positive bias induced by the local steps. As a result, we do not observe any statistical gains when the communication is infrequent. The analysis is inspired the analysis of Q-learning by Li et al. [2023] and is based on analyzing the convergence of an intermittent communication algorithm on a specifically chosen "hard" instance of MDP. Please refer to Appendix B for a detailed proof.

***Remark*** 1 (Communication complexity of policy evaluation). Several recent studies [Liu and Olshevsky, 2023, Tian et al., 2024] established that a single round of communication is sufficient to achieve linear speedup of TD learning for *policy evaluation*, which do not contradict with our results focusing on Q-learning for *policy learning*. The latter is more involved due to the nonlinearity of the Bellman optimality operator. Specifically, if the operator whose fixed point is to be found is linear in the decision variable (e.g., the value function in TD learning) then the fixed point update only induces a variance term corresponding to the noise. However, if the operator is non-linear, then in addition to the variance term, we also obtain a *bias* term in the fixed point update. While the variance term can be controlled with one-shot averaging, more frequent communication is necessary to ensure that the bias term is small enough.

***Remark*** 2 (Extension to asynchronous Q-learning). We would like to point out that our lower bound extends to the asynchronous setting [Li et al., 2023] as the assumption of i.i.d. noise corresponding to a generative model is a special case of Markovian noise observed in the asynchronous setting.

## 4 The **Fed-DVR-Q** algorithm

Having characterized the lower bound on the communication complexity of Federated Q-learning, we explore our second question of interest — designing a federated Q-learning algorithm that achieves this lower bound while simultaneously offering an optimal order of sample complexity.

We propose a new Federated Q-learning algorithm, **Fed-DVR-Q**, that achieves not only a communication complexity of $\mathsf{CC}_{\mathsf{round}} = \tilde{\mathcal{O}}(\frac{1}{1-\gamma})$ and $\mathsf{CC}_{\mathsf{bit}} = \tilde{\mathcal{O}}(\frac{|\mathcal{S}||\mathcal{A}|}{1-\gamma})$ but also the optimal order of sample complexity (upto logarithmic factors), thereby providing a tight characterization of the achievability frontier that matches with the converse result derived in the previous section.

### 4.1 Algorithm Description

Fed-DVR-Q proceeds in epochs. During an epoch $k \geq 1$, the agents collaboratively update $Q^{(k-1)}$, the estimate of $Q^\star$ obtained at the end of previous epoch, to a new estimate $Q^{(k)}$, with the aid of the sub-routine called REFINEESTIMATE. The sub-routine REFINEESTIMATE is designed to ensure that the suboptimality gap, $\|Q^{(k)} - Q^\star\|_\infty$, reduces by a factor of 2 at the end of every epoch. Thus, at the end of $K = \mathcal{O}(\log(1/\varepsilon))$ epochs, Fed-DVR-Q obtains a $\varepsilon$-optimal estimate of $Q^\star$, which is then set to be the output of the algorithm. Please refer to Alg. 2 for a pseudocode.

---

**Algorithm 2:** Fed-DVR-Q

1: Input : Error bound $\varepsilon > 0$, failure probability $\delta > 0$
2: $k \leftarrow 1, Q^{(0)} \leftarrow \mathbf{0}$
3: // Set parameters as described in Sec. 4.1.3
4: **for** $k = 1, 2, \ldots, K$ **do**
5:     $Q^{(k)} \leftarrow$ REFINEESTIMATE$(Q^{(k-1)}, B, I, L_k, D_k, J)$
6:     $k \leftarrow k + 1$
7: **end for**
8: **return** $Q^{(K)}$

---

#### 4.1.1 The REFINEESTIMATE sub-routine

REFINEESTIMATE, starting from $\overline{Q}$, an initial estimate of $Q^\star$, uses variance reduced Q-learning updates to obtain an improved estimate of $Q^\star$. It is characterized by four parameters — the initial estimate $\overline{Q}$, the number of local iterations $I$, the recentering sample size $L$ and the batch size $B$,

which can be appropriately tuned to control the quality of the returned estimate. Additionally, it also takes input two parameters $D$ and $J$ required by the compressor.

The first step in REFINEESTIMATE is to collaboratively approximate $\mathcal{T}\overline{Q}$ for the variance reduced updates. To this effect, each agent $m$ builds an approximation of $\mathcal{T}\overline{Q}$ as follows:

$$\widetilde{\mathcal{T}}_L^{(m)}(\overline{Q}) := \frac{1}{\lceil L/M \rceil} \sum_{l=1}^{\lceil L/M \rceil} \mathcal{T}_{Z_l^{(m)}}(\overline{Q}), \tag{9}$$

where $\{Z_1^{(m)}, Z_2^{(m)}, \ldots, Z_{\lceil L/M \rceil}^{(m)}\}$ are $\lceil L/M \rceil$ i.i.d. samples with $Z_1^{(m)} \sim Z$. Each agent sends $\mathscr{C}\left(\widetilde{\mathcal{T}}_L^{(m)}(\overline{Q}) - \overline{Q}; D, J\right)$, a compressed version of the difference $\widetilde{\mathcal{T}}_L^{(m)}(\overline{Q}) - \overline{Q}$, to the server, which collects all the estimates from the agents and constructs the estimate

$$\widetilde{\mathcal{T}}_L(\overline{Q}) = \overline{Q} + \frac{1}{M} \sum_{m=1}^{M} \mathscr{C}\left(\widetilde{\mathcal{T}}_L^{(m)}(\overline{Q}) - \overline{Q}; D, J\right) \tag{10}$$

and sends it back to the agents for the variance reduced updates. We defer the description of the compression routine to the end of this section. Equipped with the estimate $\widetilde{\mathcal{T}}_L(\overline{Q})$, REFINEESTIMATE constructs a sequence $\{Q_i\}_{i=1}^{I}$ using the following iterative update scheme initialized with $Q_0 = \overline{Q}$. During the $i^{\text{th}}$ iteration, each agent $m$ carries out the following update:

$$Q_{i-\frac{1}{2}}^m = (1-\eta)Q_{i-1} + \eta \left[\widehat{\mathcal{T}}_i^{(m)}Q_{i-1} - \widehat{\mathcal{T}}_i^{(m)}\overline{Q} + \widetilde{\mathcal{T}}_L(\overline{Q})\right]. \tag{11}$$

In the above equation, $\eta \in (0,1)$ is the step size and $\widehat{\mathcal{T}}_i^{(m)}Q := \frac{1}{B} \sum_{z \in \mathcal{Z}_i^{(m)}} \mathcal{T}_z Q$, where $\mathcal{Z}_i^{(m)}$ is the minibatch of $B$ i.i.d. random variables drawn according to $Z$, independently at each agent $m$ for all iterations $i$. Each agent then sends a compressed version of the update, i.e., $\mathscr{C}\left(Q_{i-\frac{1}{2}}^m - Q_{i-1}; D, J\right)$, to the server, which uses them to compute the next iterate

$$Q_i = Q_{i-1} + \frac{1}{M} \sum_{m=1}^{M} \mathscr{C}\left(Q_{i-\frac{1}{2}}^m - Q_{i-1}; D, J\right), \tag{12}$$

and broadcast it to the clients. After $I$ such updates, the obtained iterate $Q_I$ is returned by the routine. A pseudocode of the REFINEESTIMATE routine is given in Algorithm 3 in Appendix A.

### 4.1.2 The Compression Operator

The compressor, $\mathscr{C}(\cdot; D, J)$, used in the proposed algorithm Fed-DVR-Q is based on the popular stochastic quantization scheme. In addition to the input vector $Q$ to be quantized, the quantizer $\mathscr{C}$ takes two input parameters $D$ and $J$. $D$ corresponds to an upper bound on $\ell_\infty$ norm of $Q$, i.e., $\|Q\|_\infty \leq D$. $J$ corresponds to the resolution of the compressor, i.e., number of bits used by the compressor to represent each coordinate of the output vector.

The compressor first splits the interval $[0, D]$ into $2^J - 1$ intervals of equal length where $0 = d_1 < d_2, \cdots < d_{2^J} = D$ correspond to end points of the intervals. Each coordinate of $Q$ is then separately quantized as follows. The value of the $n^{\text{th}}$ coordinate, $\mathscr{C}(Q)[n]$, is set to be $d_{j_n-1}$ with probability $\frac{d_{j_n} - Q[n]}{d_{j_n} - d_{j_n-1}}$ and to $d_{j_n}$ with the remaining probability, where $j_n := \min\{j : d_j < Q[i] \leq d_{j+1}\}$. It is straightforward to note that each coordinate of $\mathscr{C}(Q)$ can be represented using $J$ bits.

### 4.1.3 Setting the parameters

The desired convergence of the iterates $\{Q^{(k)}\}$ is obtained by carefully choosing the parameters of the sub-routine REFINEESTIMATE and the compression operator $\mathscr{C}$. For all epochs $k \geq 1$, we set the number of iterations $I$ and the batch size $B$ of REFINEESTIMATE and the number of bits $J$ of the compressor $\mathscr{C}$ to be $\lceil \frac{2}{\eta(1-\gamma)} \rceil$, $\lceil \frac{2}{M}(\frac{12\gamma}{(1-\gamma)})^2 \log(\frac{8KI|\mathcal{S}||\mathcal{A}|}{\delta}) \rceil$ and $\lceil \log_2(\frac{70}{\eta(1-\gamma)} \sqrt{\frac{2}{M} \log(\frac{8KI|\mathcal{S}||\mathcal{A}|}{\delta})}) \rceil$ respectively. The total number of epochs is set to $K = \lceil \frac{1}{2} \log_2(\frac{1}{1-\gamma}) \rceil + \lceil \frac{1}{2} \log_2(\frac{1}{(1-\gamma)\varepsilon^2}) \rceil$. The recentering sample sizes $L_k$ and bounds $D_k$ are set to be the following functions of epoch index $k$:

$$L_k := \frac{19600}{(1-\gamma)^2} \log\left(\frac{8KI|\mathcal{S}||\mathcal{A}|}{\delta}\right) \cdot \begin{cases} 4^k & \text{if } k \leq K_0 \\ 4^{k-K_0} & \text{if } k > K_0 \end{cases}; \quad D_k := 16 \cdot \frac{2^{-k}}{1-\gamma}, \tag{13}$$

where $K_0 = \lceil \frac{1}{2} \log_2(\frac{1}{1-\gamma}) \rceil$. The piecewise definition of $L_k$ is crucial to obtain the optimal dependence with respect to $\frac{1}{1-\gamma}$, similar to the two-step procedure outlined in Wainwright [2019b].

## 4.2 Performance Guarantees

The following theorem characterizes the sample and communication complexity of Fed-DVR-Q.

**Theorem 2.** *Consider any $\delta \in (0,1)$ and $\varepsilon \in (0,1]$. Under the federated learning setup described in Section 2.1, the sample and communication complexities of the* Fed-DVR-Q *algorithm, when run with the choice of parameters described in Sec. 4.1.3 and a learning rate $\eta \in (0,1)$, satisfy the following relations for some universal constant $C_1 > 0$:*

$$\mathsf{SC}(\mathsf{Fed\text{-}DVR\text{-}Q}; \varepsilon, M, \delta) \leq \frac{C_1}{\eta M(1-\gamma)^3 \varepsilon^2} \log_2\left(\frac{1}{(1-\gamma)\varepsilon}\right) \log\left(\frac{8KI|\mathcal{S}||\mathcal{A}|}{\delta}\right),$$

$$\mathsf{CC}_{\mathsf{round}}(\mathsf{Fed\text{-}DVR\text{-}Q}; \varepsilon, \delta) \leq \frac{16}{\eta(1-\gamma)} \log_2\left(\frac{1}{(1-\gamma)\varepsilon}\right),$$

$$\mathsf{CC}_{\mathsf{bit}}(\mathsf{Fed\text{-}DVR\text{-}Q}; \varepsilon, \delta) \leq \frac{32|\mathcal{S}||\mathcal{A}|}{\eta(1-\gamma)} \log_2\left(\frac{1}{(1-\gamma)\varepsilon}\right) \log_2\left(\frac{70}{\eta(1-\gamma)}\sqrt{\frac{2}{M}\log\left(\frac{8KI|\mathcal{S}||\mathcal{A}|}{\delta}\right)}\right).$$

A proof of Theorem 2 can be found in Appendix C. A few implications of the theorem are in order.

**Optimal Sample-Communication complexity trade-off.** As shown by the above theorem, Fed-DVR-Q offers a linear speed up in the sample complexity with respect to the number of agents while simultaneously achieving the same order of communication complexity as dictated by the lower bound derived in Theorem 1, both in terms of frequency and bit level complexity. Moreover, Fed-DVR-Q is the *first* Federated Q-Learning algorithm that achieves a sample complexity with optimal dependence on all the salient parameters, i.e., $|\mathcal{S}|, |\mathcal{A}|$ and $\frac{1}{1-\gamma}$, in addition to linear speedup w.r.t. to number of agents and thereby bridges the existing gap between upper and lower bounds on sample complexity for Federated Q-learning. Thus, Theorem 1 and 2 together provide a characterization of optimal operating point of the sample-communication complexity trade-off in Federated Q-learning.

**Role of Minibatching.** The commonly adopted approach in intermittent communication algorithm is to use a local update scheme that takes multiple small (i.e., $B = \mathcal{O}(1)$), noisy updates between communication rounds, as evident from the algorithm design in Khodadadian et al. [2022], Woo et al. [2023] and even numerous FL algorithms for stochastic optimization McMahan et al. [2017], Haddadpour et al. [2019], Khaled et al. [2020]. In Fed-DVR-Q, we replace the local update scheme of taking multiple small, noisy updates by a single, large update with smaller variance, obtained by averaging the noisy updates over a minibatch of samples. The use of updates with smaller variance in variance reduced Q-learning yields the algorithm its name. While both the approaches result in similar sample complexity guarantees, the local update scheme requires more frequent averaging across clients to ensure that the bias of the estimate, also commonly referred to as "client drift", is not too large. On the other hand, the minibatching approach does not encounter the problem of bias accumulation from local updates and hence can afford more infrequent averaging allowing Fed-DVR-Q to achieve optimal order of communication complexity.

**Compression.** Fed-DVR-Q is the first algorithm in Federated Q-Learning to analyze and establish communication complexity at the bit level. All existing studies on Federated RL focus only on the frequency of communication and assume transmission of real numbers with infinite bit precision. On the other hand, the our analysis provides a more holistic view point of communication complexity and provides bounds at the bit level, which is of great practical significance. While some recent other studies [Wang et al., 2023] also consider quantization in Federated RL, their objective is to understand the impact of message size on convergence with no constraint on the frequency of communication, unlike the holistic viewpoint adopted in this work.

# 5 Conclusion and Future Directions

We presented a complete characterization of the sample-communication trade-off for Federated Q-learning algorithms with intermittent communication. We showed that no Federated Q-learning algorithm with intermittent communication can achieve a linear speedup with respect to the number of agents if its number of communication rounds are sublinear in $\frac{1}{1-\gamma}$. We also proposed a new Federated Q-learning algorithm called Fed-DVR-Q that uses variance reduction along with minibatching to achieve optimal-order sample and communication complexities. In particular, we showed that Fed-DVR-Q has a sample complexity of $\tilde{\mathcal{O}}(\frac{|\mathcal{S}||\mathcal{A}|}{M(1-\gamma)^3\varepsilon^2})$, which is order-optimal in all salient problem parameters, and a communication complexity of $\tilde{\mathcal{O}}(\frac{1}{1-\gamma})$ rounds and $\tilde{\mathcal{O}}(\frac{|\mathcal{S}||\mathcal{A}|}{1-\gamma})$ bits.

The results in this work raise several interesting questions that are worth exploring. While we focus on the tabular setting in this work, it is of great interest to investigate to the trade-off in other settings where we use function approximation to model the $Q^\star$ and $V^\star$ functions. Moreover, it is interesting to explore the trade-off in the finite horizon setting, where there is no discount factor. Furthermore, it is also worthwhile to explore if the communication complexity can be further reduced by going beyond the class of intermittent communication algorithms.

## Acknowledgement

We would like to thank the anonymous reviewers for their constructive feedback. This work is supported in part by the grants NSF CCF-2007911, CCF-2106778, CNS-2148212, ECCS-2318441, ONR N00014-19-1-2404 and AFRL FA8750-20-2-0504, and in part by funds from federal agency and industry partners as specified in the Resilient & Intelligent NextG Systems (RINGS) program.

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

# A Additional details about REFINEESTIMATE

We outline below the pseudocode of the REFINEESTIMATE routine described in Sec. 4.1.1.

---

**Algorithm 3:** REFINEESTIMATE$(\overline{Q}, B, I, L, D, J)$

---

1: Input: Initial estimate $\overline{Q}$, batch size $B$, Number of iterations $I$, recentering sample size $L$, quantization bound $D$, message size $J$
2: // Build an approximation for $\mathcal{T}\overline{Q}$ which is to be used for variance reduced updates
3: **for** $m = 1, 2, \ldots, M$ **do**
4:    Draw $\lceil L/M \rceil$ i.i.d. samples from the generative model for each $(s, a)$ pair and evaluate $\widetilde{\mathcal{T}}_L^{(m)}(\overline{Q})$ according to Eqn. (9)
5:    Send $\mathscr{C}(\widetilde{\mathcal{T}}_L^{(m)}(\overline{Q}) - \overline{Q}; D, J)$ to the server
6:    Receive $\frac{1}{M} \sum_{m=1}^M \mathscr{C}(\widetilde{\mathcal{T}}_L^{(m)}(\overline{Q}) - \overline{Q}; D, J)$ from the server and compute $\widetilde{\mathcal{T}}_L(\overline{Q})$ according to Eqn. (10)
7: **end for**
8: $Q_0 \leftarrow \overline{Q}$
9: // Variance reduced updates with minibatching
10: **for** $i = 1, 2, \ldots, I$ **do**
11:    **for** $m = 1, 2, \ldots, M$ **do**
12:       Draw $B$ i.i.d. samples from the from the generative model for each $(s, a)$ pair
13:       Compute $Q_{i-\frac{1}{2}}^m$ according to Eqn. (11)
14:       Send $\mathscr{C}(Q_{i-\frac{1}{2}}^m - Q_{i-1}; D, J)$ to the server
15:       Receive $\frac{1}{M} \sum_{m=1}^M \mathscr{C}(Q_i^m - Q_{i-1}; D, J)$ from the server and compute $Q_i$ according to Eqn. (12)
16:    **end for**
17: **end for**
18: **return** $Q_I$

---

# B Proof of Theorem 1

In this section, we prove the main result of the paper, the lower bound on the communication complexity of federated Q-learning algorithms. At a high level, the proof consists of the following three steps.

**Introducing the "hard" MDP instance.** The proof builds upon analyzing the behavior of a generic algorithm $\mathscr{A}$ outlined in Algorithm 1 over a particular instance of MDP. The particular choice of MDP is inspired by, and borrowed from, other lower bound proofs in the single-agent setting [Li et al., 2023] and helps highlight core issues that lie at the heart of the sample-communication complexity trade-off. Following Li et al. [2023], the construction is first over a small state-action space that allows us to focus on a simpler problem before generalizing it to larger state-action spaces.

**Establishing the performance of intermittent communication algorithms.** In the second step, we analyze the error of the iterates generated by an intermittent communication algorithm $\mathscr{A}$. The analysis is inspired by the single-agent analysis in Li et al. [2023], which highlights the underlying bias-variance trade-off. Through careful analysis of the algorithm dynamics in the federated setting, we uncover the impact of communication on the bias-variance trade-off and the resulting error of the iterates to obtain the lower bound on the communication complexity.

**Generalization to larger MDPs.** As the last step, we generalize our construction of the "hard" instance to more general state-action space and extend our insights to obtain the statement of the theorem.

### B.1 Introducing the "hard" instance

We first introduce an MDP instance $\mathcal{M}_h$ that we will use throughout the proof to establish the result. Note that this MDP is identical to the one considered in Li et al. [2023] to establish the lower bounds on the performance of single-agent Q-learning algorithm. It consists of four states $\mathcal{S} = \{0, 1, 2, 3\}$. Let $\mathcal{A}_s$ denote the action set associated with the state $s$. The probability transition kernel and the reward function of $\mathcal{M}_h$ is given as follows:

$$\mathcal{A}_0 = \{1\} \quad P(0|0,1) = 1 \quad\quad\quad\quad\quad\quad\quad\quad\quad\quad r(0,1) = 0, \quad (14a)$$
$$\mathcal{A}_1 = \{1,2\} \quad P(1|1,1) = p \quad\quad P(0|1,1) = 1-p \quad r(1,1) = 1, \quad (14b)$$
$$\quad\quad\quad\quad\quad P(1|1,2) = p \quad\quad P(0|1,2) = 1-p \quad r(1,2) = 1, \quad (14c)$$
$$\mathcal{A}_2 = \{1\} \quad P(2|2,1) = p \quad\quad P(0|2,1) = 1-p \quad r(2,1) = 1, \quad (14d)$$
$$\mathcal{A}_3 = \{1\} \quad P(3|3,1) = 1 \quad\quad\quad\quad\quad\quad\quad\quad\quad\quad r(3,1) = 1, \quad (14e)$$

where the parameter $p = \dfrac{4\gamma - 1}{3\gamma}$. We have the following results about the optimal $Q$ and $V$ functions of this hard MDP instance.

**Lemma 1** ([Li et al., 2023, Lemma 3]). *Consider the MDP $\mathcal{M}_h$ constructed in Eqn. (14). We have,*

$$V^\star(0) = Q^\star(0,1) = 0$$
$$V^\star(1) = Q^\star(1,1) = Q^\star(1,2) = V^\star(2) = Q^\star(2,1) = \frac{1}{1-\gamma p} = \frac{3}{4(1-\gamma)}$$
$$V^\star(3) = Q^\star(3,1) = \frac{1}{1-\gamma}.$$

Throughout the next section of the proof, we focus on this MDP with four states and two actions. In Appendix B.4, we generalize the proof to larger state-action spaces.

### B.2 Notation and preliminary results

For convenience, we first define some notation that will be used throughout the proof.

**Useful relations of the learning rates.** We consider two kinds of step size sequences that are commonly used in Q-learning — the constant step size schedule, i.e., $\eta_t = \eta$ for all $t \in \{1, 2, \ldots, T\}$ and the rescaled linear step size schedule, i.e., $\eta_t = \frac{1}{1+c_\eta(1-\gamma)t}$, where $c_\eta > 0$ is a universal constant that is independent of the problem parameters.

We define the following quantities:

$$\eta_k^{(t)} = \eta_k \prod_{i=k+1}^{t} (1 - \eta_i(1 - \gamma p)) \quad\quad \text{for all } 0 \le k \le t, \quad (15)$$

where we take $\eta_0 = 1$ and use the convention throughout the proof that if a product operation does not have a valid index, we take the value of that product to be 1. For any integer $0 \le \tau < t$, we have the following relation, which will be proved at the end of this subsection for completeness:

$$\prod_{k=\tau+1}^{t} (1 - \eta_k(1 - \gamma p)) + (1 - \gamma p) \sum_{k=\tau+1}^{t} \eta_k^{(t)} = 1. \quad (16)$$

Similarly, we also define,

$$\widetilde{\eta}_k^{(t)} = \eta_k \prod_{i=k+1}^{t} (1 - \eta_i) \quad\quad \text{for all } 0 \le k \le t, \quad (17)$$

which satisfies the relation

$$\prod_{k=\tau+1}^{t} (1 - \eta_k) + \sum_{k=\tau+1}^{t} \widetilde{\eta}_k^{(t)} = 1. \quad (18)$$

for any integer $0 \leq \tau < t$. The claim follows immediately by plugging $p = 0$ in (16). Note that for constant step size, the sequence $\widetilde{\eta}_k^{(t)}$ is clearly increasing. For the rescaled linear step size, we have,

$$\frac{\widetilde{\eta}_{k-1}^{(t)}}{\widetilde{\eta}_k^{(t)}} = \frac{\eta_k}{\eta_{k-1}(1 - \eta_k)} = 1 - \frac{(1 - c_\eta(1 - \gamma))\eta_k}{1 - c_\eta(1 - \gamma)\eta_k} \leq 1 \tag{19}$$

whenever $c_\eta \leq \frac{1}{1-\gamma}$. Thus, $\widetilde{\eta}_k^{(t)}$ is an increasing sequence as long as $c_\eta \leq \frac{1}{1-\gamma}$. Similarly, $\eta_k^{(t)}$ is also clearly increasing for the constant step size schedule. For the rescaled linear step size schedule, we have,

$$\frac{\eta_{k-1}^{(t)}}{\eta_k^{(t)}} = \frac{\eta_k}{\eta_{k-1}(1 - \eta_k(1 - \gamma p))} \leq \frac{\eta_k}{\eta_{k-1}(1 - \eta_k)} \leq 1,$$

whenever $c_\eta \leq \frac{1}{1-\gamma}$. The last bound follows from Eqn. (19).

**Proof of** (16). We can show the claim using backward induction. For the base case, note that,

$$(1 - \gamma p)\eta_t^{(t)} + (1 - \gamma p)\eta_{t-1}^{(t)} = (1 - \gamma p)\eta_t + (1 - \gamma p)\eta_{t-1}(1 - (1 - \gamma p)\eta_t)$$

$$= 1 - (1 - \eta_t(1 - \gamma p))(1 - \eta_{t-1}(1 - \gamma p)) = 1 - \prod_{k=t-1}^{t}(1 - \eta_k(1 - \gamma p)),$$

as required. Assume (16) is true for some $\tau$. We have,

$$(1 - \gamma p) \sum_{k=\tau}^{t} \eta_k^{(t)} = (1 - \gamma p)\eta_\tau^t + (1 - \gamma p) \sum_{k=\tau+1}^{t} \eta_k^{(t)}$$

$$= (1 - \gamma p)\eta_\tau \prod_{k=\tau+1}^{t}(1 - \eta_k(1 - \gamma p)) + 1 - \prod_{k=\tau+1}^{t}(1 - \eta_k(1 - \gamma p))$$

$$= 1 - \prod_{k=\tau}^{t}(1 - \eta_k(1 - \gamma p)),$$

thus completing the induction step.

**Sample transition matrix.** Recall $Z \in \mathcal{S}^{|\mathcal{S}||\mathcal{A}|}$ is a random vector whose $(s, a)$-th coordinate is drawn from the distribution $P(\cdot|s, a)$. We use $\widehat{P}_t^m$ to denote the sample transition at time $t$ and agent $m$ obtained by averaging $B$ i.i.d. samples from the generative model. Specifically let $\{Z_{t,b}^m\}_{b=1}^B$ denote a collection of $B$ i.i.d. copies of $Z$ collected at time $t$ at agent $m$. Then, for all $s, a, s'$,

$$\widehat{P}_t^m(s'|s, a) = \frac{1}{B} \sum_{b=1}^{B} P_{t,b}^m(s'|s, a), \tag{20}$$

where $P_{t,b}^m(s'|s, a) = \mathbb{1}\{Z_{t,b}^m(s, a) = s'\}$ for $s' \in \mathcal{S}$.

**Preliminary relations of the iterates.** We state some preliminary relations regarding the evolution of the Q-function and the value function across different agents that will be helpful for the analysis later.

We begin with the state 0, where we have $Q_t^m(0, 1) = V_t^m(0) = 0$ for all agents $m \in [M]$ and $t \in [T]$. This follows almost immediately from the fact that state 0 is an absorbing state with zero reward. Note that $Q_0^m(0, 1) = V_0^m(0) = 0$ holds for all clients $m \in [M]$. Assuming that $Q_{t-1}^m(0, 1) = V_{t-1}^m(0) = 0$ for all clients for some time instant $t - 1$, by induction, we have,

$$Q_{t-1/2}^m(0, 1) = (1 - \eta_t)Q_{t-1}^m(0, 1) + \eta_t(\gamma V_{t-1}^m(0)) = 0.$$

Consequently, $Q_t^m(0, 1) = 0$ and $V_t^m(0) = 0$, for all agents $m$, irrespective of whether there is averaging.

For state 3, the iterates satisfy the following relation:

$$Q_{t-1/2}^m(3,1) = (1-\eta_t)Q_{t-1}^m(3,1) + \eta_t(1+\gamma V_{t-1}^m(3))$$
$$= (1-\eta_t)Q_{t-1}^m(3,1) + \eta_t(1+\gamma Q_{t-1}^m(3,1))$$
$$= (1-\eta_t(1-\gamma))Q_{t-1}^m(3,1) + \eta_t,$$

where the second step follows by noting $V_t^m(3) = Q_t^m(3,1)$. Once again, one can note that averaging step does not affect the update rule implying that the following holds for all $m \in [M]$ and $t \in [T]$:

$$V_t^m(3) = Q_t^m(3,1) = \sum_{k=1}^{t} \eta_k \left( \prod_{i=k+1}^{t} (1-\eta_i(1-\gamma)) \right) = \frac{1}{1-\gamma}\left[ 1 - \prod_{i=1}^{t}(1-\eta_i(1-\gamma)) \right], \tag{21}$$

where the last step follows from Eqn. (16) with $p = 1$.

Similarly, for state 1 and 2, we have,

$$Q_{t-1/2}^m(1,1) = (1-\eta_t)Q_{t-1}^m(1,1) + \eta_t(1+\gamma \widehat{P}_t^m(1|1,1)V_{t-1}^m(1)), \tag{22}$$

$$Q_{t-1/2}^m(1,2) = (1-\eta_t)Q_{t-1}^m(1,2) + \eta_t(1+\gamma \widehat{P}_t^m(1|1,2)V_{t-1}^m(1)), \tag{23}$$

$$Q_{t-1/2}^m(2,1) = (1-\eta_t)Q_{t-1}^m(2,1) + \eta_t(1+\gamma \widehat{P}_t^m(2|2,1)V_{t-1}^m(2)). \tag{24}$$

Since the averaging makes a difference in the update rule, we further analyze the update as required in later proofs.

## B.3   Main analysis

We first focus on establishing a bound on the number of communication rounds, i.e., $\mathsf{CC}_{\mathsf{round}}(\mathscr{A})$ (where we drop the dependency with other parameters for notational simplicity), and then use this lower bound to establish the bound on the bit level communication complexity $\mathsf{CC}_{\mathsf{bit}}(\mathscr{A})$.

To establish the lower bound on $\mathsf{CC}_{\mathsf{round}}(\mathscr{A})$ for any intermittent communication algorithm $\mathscr{A}$, we analyze the convergence behavior of $\mathscr{A}$ on the MDP $\mathcal{M}_h$. We assume that the averaging step in line 6 of Algorithm 1 is carried out exactly. Since the use of compression only makes the problem harder, it is sufficient for us to consider the case where there is no loss of information in the averaging step for establishing a lower bound. Lastly, throughout the proof, without loss of generality we assume that

$$\log N \leq \frac{1}{1-\gamma}, \tag{25}$$

otherwise, the lower bound in Theorem 1 reduces to the trivial lower bound.

We divide the proof into following three parts based on the choice of learning rates and batch sizes:

1. Small learning rates: For constant learning rates, $0 \leq \eta < \frac{1}{(1-\gamma)T}$ and for rescaled linear learning rates, the constant $c_\eta$ satisfies $c_\eta \geq \log T$.

2. Large learning rates with small $\eta_T/(BM)$: For constant learning rates, $\eta \geq \frac{1}{(1-\gamma)T}$ and for rescaled linear learning rates, the constant $c_\eta$ satisfies $0 \leq c_\eta \leq \log T \leq \frac{1}{1-\gamma}$ (c.f. (25)). Additionally, the ratio $\frac{\eta_T}{BM}$ satisfies $\frac{\eta_T}{BM} \leq \frac{1-\gamma}{100}$.

3. Large learning rates with large $\eta_T/(BM)$: We have the same condition on the learning rates as above. However, in this case the ratio $\frac{\eta_T}{BM}$ satisfies $\frac{\eta_T}{BM} > \frac{1-\gamma}{100}$.

We consider each of the cases separately in the following three subsections.

### B.3.1   Small learning rates

In this subsection, we prove the lower bound for small learning rates, which follow from similar arguments in Li et al. [2023].

For this case, we focus on the dynamics of state 2. We claim that the same relation established in Li et al. [2023] continues to hold, which will be established momentarily:

$$\mathbb{E}[V_T^m(2)] = \left( \frac{1}{M} \sum_{j=1}^M \mathbb{E}[V_T^j(2)] \right) = \sum_{k=1}^T \eta_k^{(t)} = \frac{1 - \eta_0^{(T)}}{1 - \gamma p}. \tag{26}$$

Consequently, for all $m \in [M]$, we have

$$V^\star(2) - \mathbb{E}[V_T^m(2)] = \frac{\eta_0^{(T)}}{1 - \gamma p}. \tag{27}$$

To obtain lower bound on $V^\star(2) - \mathbb{E}[V_T^m(2)]$, we need to obtain a lower bound on $\eta_0^{(T)}$, which from [Li et al., 2023, Eqn. (120)] we have

$$\log(\eta_0^{(T)}) \geq -1.5 \sum_{t=1}^T \eta(1 - \gamma p) \geq -2 \sum_{t=1}^T \frac{1}{t \log T} \geq -2 \quad \implies \quad \eta_0^{(T)} \geq e^{-2}$$

when $T \geq 16$ for both choices of learning rates. On plugging this bound in (27), we obtain,

$$\mathbb{E}[\|Q_T^m - Q^\star\|_\infty] \geq \mathbb{E}[|Q^\star(2) - Q_T^m(2)|] \geq V^\star(2) - \mathbb{E}[V_T^m(2)] \geq \frac{3}{4e^2(1 - \gamma)\sqrt{N}} \tag{28}$$

holds for all $m \in [M]$, $N \geq 1$ and $M \geq 2$. Thus, it can be noted that the error rate $\mathsf{ER}(\mathscr{A}; N, M)$ is bounded away from a constant value irrespective of the number of agents and the number of communication rounds. Thus, even with $\mathsf{CC}_{\mathsf{round}} = \Omega(T)$, we will not observe any collaborative gain if the step size is too small.

**Proof of** (26). Recall that from (24), we have,

$$Q_{t-1/2}^m(2, 1) = (1 - \eta_t)V_{t-1}^m(2) + \eta_t(1 + \gamma \widehat{P}_t^m(2|2, 1)V_{t-1}^m(2)).$$

Here, $Q_{t-1}^m(2, 1) = V_{t-1}^m(2)$ as the second state has only a single action.

- When $t$ is not an averaging instant, we have,

$$V_t^m(2) = Q_t^m(2, 1) = (1 - \eta_t)V_{t-1}^m(2) + \eta_t(1 + \gamma \widehat{P}_t^m(2|2, 1)V_{t-1}^m(2)). \tag{29}$$

  On taking expectation on both sides of the equation, we obtain,

$$\begin{aligned}
\mathbb{E}[V_t^m(2)] &= (1 - \eta_t)\mathbb{E}[V_{t-1}^m(2)] + \eta_t(1 + \gamma\mathbb{E}[\widehat{P}_t^m(2|2, 1)V_{t-1}^m(2)]) \\
&= (1 - \eta_t)\mathbb{E}[V_{t-1}^m(2)] + \eta_t \left( 1 + \gamma\mathbb{E}[\widehat{P}_t^m(2|2, 1)]\mathbb{E}[V_{t-1}^m(2)] \right) \\
&= (1 - \eta_t)\mathbb{E}[V_{t-1}^m(2)] + \eta_t \left( 1 + \gamma p\mathbb{E}[V_{t-1}^m(2)] \right) \\
&= (1 - \eta_t(1 - \gamma p))\mathbb{E}[V_{t-1}^m(2)] + \eta_t. \tag{30}
\end{aligned}$$

  In the second step, we used the fact that $\widehat{P}_t^m(2|2, 1)$ is independent of $V_{t-1}^m(2)$.

- Similarly, if $t$ is an averaging instant, we have,

$$\begin{aligned}
V_t^m(2) = Q_t^m(2, 1) &= \frac{1}{M} \sum_{j=1}^M Q_{t-1/2}^j(2, 1) \\
&= (1 - \eta_t)\frac{1}{M} \sum_{j=1}^M V_{t-1}^j(2) + \frac{1}{M} \sum_{j=1}^M \eta_t(1 + \gamma\widehat{P}_t^j(2|2, 1)V_{t-1}^j(2)). \tag{31}
\end{aligned}$$

  Once again, upon taking expectation we obtain,

$$\mathbb{E}[V_t^m(2)] = (1 - \eta_t)\frac{1}{M} \sum_{j=1}^M \mathbb{E}[V_{t-1}^j(2)] + \frac{1}{M} \sum_{j=1}^M \eta_t(1 + \gamma\mathbb{E}[\widehat{P}_t^j(2|2, 1)V_{t-1}^j(2)])$$

$$= (1 - \eta_t) \frac{1}{M} \sum_{j=1}^{M} \mathbb{E}[V_{t-1}^j(2)] + \frac{1}{M} \sum_{j=1}^{M} \eta_t(1 + \gamma p \mathbb{E}[V_{t-1}^j(2)])$$

$$= (1 - \eta_t(1 - \gamma p)) \left( \frac{1}{M} \sum_{j=1}^{M} \mathbb{E}[V_{t-1}^j(2)] \right) + \eta_t. \tag{32}$$

Eqns. (30) and (32) together imply that for all $t \in [T]$,

$$\left( \frac{1}{M} \sum_{m=1}^{M} \mathbb{E}[V_t^m(2)] \right) = (1 - \eta_t(1 - \gamma p)) \left( \frac{1}{M} \sum_{m=1}^{M} \mathbb{E}[V_{t-1}^m(2)] \right) + \eta_t. \tag{33}$$

On unrolling the above recursion with $V_0^m = 0$ for all $m \in [M]$, we obtain the desired claim (26).

### B.3.2  Large learning rates with small $\frac{\eta_T}{BM}$

In this subsection, we prove the lower bound for case of large learning rates when the ratio $\frac{\eta_T}{BM}$ is small. For the analysis in this part, we focus on the dynamics of state 1. Unless otherwise specified, throughout the section we implicitly assume that the state is 1.

We further define a key parameter that will play a key role in the analysis:

$$\tau := \min\{k \in \mathbb{N} : \forall\, t \geq k, \eta_t \leq \eta_k \leq 3\eta_t\}. \tag{34}$$

It can be noted that for constant step size sequence $\tau = 1$ and for rescaled linear stepsize $\tau = T/3$.

**Step 1: introducing an auxiliary sequence.** We define an auxiliary sequence $\widehat{Q}_t^m(a)$ for $a \in \{1, 2\}$ and all $t = 1, 2, \ldots, T$ to aid our analysis, where we drop the dependency with state $s = 1$ for simplicity. The evolution of the sequence $\widehat{Q}_t^m$ is defined in Algorithm 4, where $\widehat{V}_t^m = \max_{a \in \{1,2\}} \widehat{Q}_t^m(a)$. In other words, the iterates $\{\widehat{Q}_t^m\}$ evolve exactly as the iterates of Algorithm 1 except for the fact that sequence $\{\widehat{Q}_t^m\}$ is initialized at the optimal $Q$-function of the MDP. We would like to point out that we assume that the underlying stochasticity is also identical in the sense that the evolution of both $Q_t^m$ and $\widehat{Q}_t^m$ is governed by the same $\widehat{P}_t^m$ matrices. The following lemma controls the error between the iterates $Q_t^m$ and $\widehat{Q}_t^m$, allowing us to focus only on $\widehat{Q}_t^m$.

---

**Algorithm 4:** Evolution of $\widehat{Q}$

1: Input : $T, R, \{\eta_t\}_{t=1}^{T}, \mathcal{C} = \{t_r\}_{r=1}^{R}, B$
2: Set $\widehat{Q}_0^m(a) \leftarrow Q^\star(1, a)$ for $a \in \{1, 2\}$ and all agents $m$      // Different initialization
3: **for** $t = 1, 2, \ldots, T$ **do**
4:     **for** $m = 1, 2, \ldots, M$ **do**
5:         Compute $\widehat{Q}_{t-\frac{1}{2}}^m$ according to Eqn. (7)
6:         Compute $\widehat{Q}_t^m$ according to Eqn. (8)
7:     **end for**
8: **end for**

---

**Lemma 2.** *The following relation holds for all agents $m \in [M]$, all $t \in [T]$ and $a \in \{1, 2\}$:*

$$Q_t^m(1, a) - \widehat{Q}_t^m(a) \geq -\frac{1}{1-\gamma} \prod_{i=1}^{t} (1 - \eta_i(1 - \gamma)).$$

By Lemma 2, bounding the error of the sequence $\widehat{Q}_t^m$ allows us to obtain a bound on the error of $Q_t^m$. To that effect, we define the following terms for any $t \leq T$ and all $m \in [M]$:

$$\Delta_t^m(a) := \widehat{Q}_t^m(a) - Q^\star(1, a); \quad a = 1, 2;$$

$$\Delta^m_{t,\mathrm{max}} = \max_{a \in \{1,2\}} \Delta^m_t(a).$$

In addition, we use $\overline{\Delta}_t = \frac{1}{M} \sum_{m=1}^M \Delta^m_t$ to denote the error of the averaged iterate[1], and similarly,

$$\overline{\Delta}_{t,\mathrm{max}} := \max_{a \in \{1,2\}} \overline{\Delta}_t(a). \tag{35}$$

We first derive a basic recursion regarding $\Delta^m_t(a)$. From the iterative update rule in Algorithm 4, we have,

$$\begin{aligned}
\Delta^m_t(a) &= (1 - \eta_t)\Delta^m_{t-1}(a) + \eta_t(1 + \gamma \widehat{P}^m_t(1|1,a)\widehat{V}^m_{t-1} - Q^\star(1,a)) \\
&= (1 - \eta_t)\Delta^m_{t-1}(a) + \eta_t \gamma(\widehat{P}^m_t(1|1,a)\widehat{V}^m_{t-1} - pV^\star(1)) \\
&= (1 - \eta_t)\Delta^m_{t-1}(a) + \eta_t \gamma(p(\widehat{V}^m_{t-1} - V^\star(1)) + (\widehat{P}^m_t(1|1,a) - p)\widehat{V}_{t-1}) \\
&= (1 - \eta_t)\Delta^m_{t-1}(a) + \eta_t \gamma(p\Delta^m_{t-1,\mathrm{max}} + (\widehat{P}^m_t(1|1,a) - p)\widehat{V}^m_{t-1}).
\end{aligned}$$

Here in the last line, we used the following relation:

$$\Delta^m_{t,\mathrm{max}} = \max_{a \in \{1,2\}} (\widehat{Q}^m_t(a) - Q^\star(1,a)) = \max_{a \in \{1,2\}} \widehat{Q}^m_t(a) - V^\star(1) = \widehat{V}^m_{t-1} - V^\star(1),$$

as $Q^\star(1,1) = Q^\star(1,2) = V^\star(1)$.

Recursively unrolling the above expression, and using the expression (17), we obtain the following relation: for any $t' < t$ when there is no averaging during the interval $(t', t)$

$$\Delta^m_t(a) = \left( \prod_{k=t'+1}^t (1 - \eta_k) \right) \Delta^m_{t'}(a) + \sum_{k=t'+1}^t \widetilde{\eta}^{(t)}_k \gamma(p\Delta^m_{k-1,\mathrm{max}} + (\widehat{P}^m_k(1|1,a) - p)\widehat{V}^m_{k-1}). \tag{36}$$

For any $t', t$ with $t' < t$, we define the notation

$$\varphi_{t',t} := \prod_{k=t'+1}^t (1 - \eta_k), \tag{37}$$

$$\xi^m_{t',t}(a) := \sum_{k=t'+1}^t \widetilde{\eta}^{(t)}_k \gamma(\widehat{P}^m_k(1|1,a) - p)\widehat{V}^m_{k-1}, \quad a = 1, 2; \tag{38}$$

$$\xi^m_{t',t,\mathrm{max}} := \max_{a \in \{1,2\}} \xi^m_{t',t}(a). \tag{39}$$

Note that by definition, $\mathbb{E}[\xi^m_{t',t}(a)] = 0$ for $a \in \{1, 2\}$ and all $m$, $t'$ and $t$. Plugging them into the previous expression leads to the simplified expression

$$\Delta^m_t(a) = \varphi_{t',t}\Delta^m_{t'}(a) + \left[ \sum_{k=t'+1}^t \widetilde{\eta}^{(t)}_k \gamma p\Delta^m_{k-1,\mathrm{max}} \right] + \xi^m_{t',t}(a).$$

We specifically choose $t'$ and $t$ to be the consecutive averaging instants to analyze the behaviour of $\Delta^m_t$ across two averaging instants. Consequently, we can rewrite the above equation as

$$\Delta^m_t(a) = \varphi_{t',t}\overline{\Delta}_{t'}(a) + \left[ \sum_{k=t'+1}^t \widetilde{\eta}^{(t)}_k \gamma p\Delta^m_{k-1,\mathrm{max}} \right] + \xi^m_{t',t}(a). \tag{40}$$

Furthermore, after averaging, we obtain,

$$\overline{\Delta}_t(a) = \varphi_{t',t}\overline{\Delta}_{t'}(a) + \frac{1}{M} \sum_{m=1}^M \left[ \sum_{k=t'+1}^t \widetilde{\eta}^{(t)}_k \gamma p\Delta^m_{k-1,\mathrm{max}} \right] + \frac{1}{M} \sum_{m=1}^M \xi^m_{t',t}(a). \tag{41}$$

---

[1]We use this different notation in appendix as opposed to the half-time indices used in the main text to improve readability of the proof.

**Step 2: deriving a recursive bound for** $\mathbb{E}[\overline{\Delta}_{t,\max}]$. Bounding (40), we obtain,

$$\Delta_{t,\max}^m \geq \varphi_{t',t}\overline{\Delta}_{t',\max} + \left[\sum_{k=t'+1}^{t} \widetilde{\eta}_k^{(t)}\gamma p\Delta_{k-1,\max}^m\right] + \xi_{t',t,\max}^m - \varphi_{t',t}|\overline{\Delta}_{t'}(1) - \overline{\Delta}_{t'}(2)|, \quad (42a)$$

$$\Delta_{t,\max}^m \leq \varphi_{t',t}\overline{\Delta}_{t',\max} + \left[\sum_{k=t'+1}^{t} \widetilde{\eta}_k^{(t)}\gamma p\Delta_{k-1,\max}^m\right] + \xi_{t',t,\max}^m, \quad (42b)$$

where in the first step we used the fact that

$$\max\{a_1 + b_1, a_2 + b_2\} \geq \min\{a_1, a_2\} + \max\{b_1, b_2\} = \max\{a_1, a_2\} + \max\{b_1, b_2\} - |a_1 - a_2|. \quad (43)$$

On taking expectation, we obtain,

$$\mathbb{E}[\Delta_{t,\max}^m] \geq \varphi_{t',t}\mathbb{E}[\overline{\Delta}_{t',\max}] + \left[\sum_{k=t'+1}^{t} \widetilde{\eta}_k^{(t)}\gamma p\mathbb{E}[\Delta_{k-1,\max}^m]\right] + \mathbb{E}[\xi_{t',t,\max}^m] - \varphi_{t',t}\mathbb{E}[|\overline{\Delta}_{t'}(1) - \overline{\Delta}_{t'}(2)|], \quad (44a)$$

$$\mathbb{E}[\Delta_{t,\max}^m] \leq \varphi_{t',t}\mathbb{E}[\overline{\Delta}_{t',\max}] + \left[\sum_{k=t'+1}^{t} \widetilde{\eta}_k^{(t)}\gamma p\mathbb{E}[\Delta_{k-1,\max}^m]\right] + \mathbb{E}[\xi_{t',t,\max}^m]. \quad (44b)$$

Similarly, using (41) and (43) we can write,

$$\overline{\Delta}_{t,\max} \geq \varphi_{t',t}\overline{\Delta}_{t',\max} + \frac{1}{M}\sum_{m=1}^{M}\left[\sum_{k=t'+1}^{t} \widetilde{\eta}_k^{(t)}\gamma p\Delta_{k-1,\max}^m\right] - \varphi_{t',t}|\overline{\Delta}_{t'}(1) - \overline{\Delta}_{t'}(2)|$$

$$+ \max\left\{\frac{1}{M}\sum_{m=1}^{M}\xi_{t',t}^m(1), \frac{1}{M}\sum_{m=1}^{M}\xi_{t',t}^m(2)\right\} \quad (45a)$$

$$\implies \mathbb{E}[\overline{\Delta}_{t,\max}] \geq \varphi_{t',t}\mathbb{E}[\overline{\Delta}_{t',\max}] + \frac{1}{M}\sum_{m=1}^{M}\left[\sum_{k=t'+1}^{t} \widetilde{\eta}_k^{(t)}\gamma p\mathbb{E}[\Delta_{k-1,\max}^m]\right] - \varphi_{t',t}\mathbb{E}[|\overline{\Delta}_{t'}(1) - \overline{\Delta}_{t'}(2)|]$$

$$+ \mathbb{E}\left[\max\left\{\frac{1}{M}\sum_{m=1}^{M}\xi_{t',t}^m(1), \frac{1}{M}\sum_{m=1}^{M}\xi_{t',t}^m(2)\right\}\right]. \quad (45b)$$

On combining (44b) and (45b), we obtain,

$$\mathbb{E}[\overline{\Delta}_{t,\max}] \geq \frac{1}{M}\sum_{m=1}^{M}\left[\mathbb{E}[\Delta_{t,\max}^m] - \mathbb{E}[\xi_{t',t,\max}^m]\right] - \varphi_{t',t}\mathbb{E}[|\overline{\Delta}_{t'}(1) - \overline{\Delta}_{t'}(2)|]$$

$$+ \mathbb{E}\left[\max\left\{\frac{1}{M}\sum_{m=1}^{M}\xi_{t',t}^m(1), \frac{1}{M}\sum_{m=1}^{M}\xi_{t',t}^m(2)\right\}\right]. \quad (46)$$

In order to simplify (46), we make use of the following lemmas.

**Lemma 3.** *Let $t' < t$ be two consecutive averaging instants. Then for all $m \in [M]$,*

$$\mathbb{E}[\Delta_{t,\max}^m] - \mathbb{E}[\xi_{t',t,\max}^m] \geq \left(\prod_{k=t'+1}^{t}(1 - \eta_k(1 - \gamma p))\right)\mathbb{E}[\overline{\Delta}_{t',\max}] + \mathbb{E}[\xi_{t',t,\max}^m]\left[\sum_{k=t'+1}^{t}\eta_k^{(t)} - 1\right]_+$$

$$- \varphi_{t',t}\mathbb{E}[|\overline{\Delta}_{t'}(1) - \overline{\Delta}_{t'}(2)|],$$

*where $[x]_+ = \max\{x, 0\}$.*

**Lemma 4.** *For all consecutive averaging instants $t', t$ satisfying $t - \max\{t', \tau\} \geq 1/\eta_\tau$ and all $m \in [M]$, we have,*

$$\mathbb{E}[\xi_{t',t,\max}^m] \geq \frac{1}{240\log\left(\frac{180B}{\eta_T(1-\gamma)}\right)} \cdot \frac{\nu}{\nu + 1},$$

$$\mathbb{E}\left[\max\left\{\frac{1}{M}\sum_{m=1}^{M}\xi_{t',t}^{m}(1), \frac{1}{M}\sum_{m=1}^{M}\xi_{t',t}^{m}(2)\right\}\right] \geq \frac{1}{240\log\left(\frac{180BM}{\eta_T(1-\gamma)}\right)} \cdot \frac{\nu}{\nu+\sqrt{M}},$$

*where* $\nu := \sqrt{\dfrac{20\eta_T}{B(1-\gamma)}}.$

**Lemma 5.** *For all* $t \in \{t_r\}_{r=1}^{R}$, *we have*

$$\mathbb{E}[|\overline{\Delta}_t(1) - \overline{\Delta}_t(2)|] \leq \sqrt{\frac{8\eta_T}{3BM(1-\gamma)}}.$$

Thus, on combining the results from Lemmas 3, 4, and 5 and plugging them into (46), we obtain the following relation for $t, t' \geq \tau$:

$$\mathbb{E}[\overline{\Delta}_{t,\max}] \geq \left(\prod_{k=t'+1}^{t}(1 - \eta_k(1 - \gamma p))\right)\mathbb{E}[\overline{\Delta}_{t',\max}] + \mathbb{E}[\xi_{t',t,\max}^{m}]\left[\sum_{k=t'+1}^{t}\eta_k^{(t)} - 1\right]_{+}$$

$$- 2\varphi_{t',t}\mathbb{E}[|\overline{\Delta}_{t'}(1) - \overline{\Delta}_{t'}(2)|] + \mathbb{E}\left[\max\left\{\frac{1}{M}\sum_{m=1}^{M}\xi_{t',t}^{m}(1), \frac{1}{M}\sum_{m=1}^{M}\xi_{t',t}^{m}(2)\right\}\right]$$

$$\geq (1 - \eta_\tau(1 - \gamma p))^{t-t'}\mathbb{E}[\overline{\Delta}_{t',\max}] + \left(\frac{1 - (1 - \eta_\tau(1 - \gamma p))^{t-t'}}{5760\log\left(\frac{180B}{\eta_T(1-\gamma)}\right)(1 - \gamma p)}\right) \cdot \frac{\nu}{\nu+1} \cdot \mathbb{1}\left\{t - t' \geq \frac{8}{\eta_\tau}\right\}$$

$$- 2(1 - \eta_T)^{t-t'}\sqrt{\frac{8\eta_T}{3BM(1-\gamma)}} + \frac{1}{240\log\left(\frac{180BM}{\eta_T(1-\gamma)}\right)} \cdot \frac{\nu}{\nu+\sqrt{M}} \cdot \mathbb{1}\left\{t - t' \geq \frac{8}{\eta_\tau}\right\},$$

$$(47)$$

where we used the relation $\varphi_{t',t} \leq (1 - \eta_T)^{t-t'}$, as well as the value of $\nu$ as defined in Lemma 4 along with the fact

$$\sum_{k=t'+1}^{t}\eta_k^{(t)} - 1 \geq \frac{1 - (1 - \eta_\tau(1 - \gamma p))^{t-t'}}{24(1 - \gamma p)} \qquad (48)$$

for all $t, t' \geq \tau$ such that $t - t' \geq 8/\eta_\tau$.

**Proof of** (48). We have,

$$\sum_{k=t'+1}^{t}\eta_k^{(t)} - 1 = \sum_{k=t'+1}^{t}\left(\eta_k\prod_{i=k+1}^{t}(1 - \eta_i(1 - \gamma p))\right) - 1$$

$$\geq \sum_{k=t'+1}^{t}\left(\eta_t\prod_{i=k+1}^{t}(1 - \eta_\tau(1 - \gamma p))\right) - 1$$

$$\geq \eta_t\sum_{k=t'+1}^{t}(1 - \eta_\tau(1 - \gamma p))^{t-k} - 1$$

$$\geq \eta_t \cdot \left(\frac{1 - (1 - \eta_\tau(1 - \gamma p))^{t-t'}}{\eta_\tau(1 - \gamma p)}\right) - 1$$

$$\geq \frac{1 - (1 - \eta_\tau(1 - \gamma p))^{t-t'}}{3(1 - \gamma p)} - 1. \qquad (49)$$

To show (48), it is sufficient to show that $\dfrac{1 - (1 - \eta_\tau(1 - \gamma p))^{t-t'}}{3(1 - \gamma p)} \geq \dfrac{8}{7}$ for $t - t' \geq 8/\eta_\tau$. Thus, for $t - t' \geq 8/\eta_\tau$ we have,

$$\frac{1 - (1 - \eta_\tau(1 - \gamma p))^{t-t'}}{3(1 - \gamma p)} \geq \frac{1 - \exp(-\eta_\tau(1 - \gamma p) \cdot (t - t'))}{3(1 - \gamma p)}$$

$$\geq \frac{1 - \exp(-8(1 - \gamma p))}{3(1 - \gamma p)}.\tag{50}$$

Since $\gamma \geq 5/6$, $1 - \gamma p \leq 2/9$. For $x \leq 2/9$, the function $f(x) = \frac{1 - e^{-8x}}{3x} \geq 8/7$, proving the claim.

**Step 3: lower bounding $\mathbb{E}[\overline{\Delta}_{T,\max}]$.** We are now interested in evaluating $\mathbb{E}[\overline{\Delta}_{T,\max}]$ based on the recursion (47). To this effect, we introduce some notation to simplify the presentation. Let

$$R_\tau := \min\{r : t_r \geq \tau\}.\tag{51}$$

For $r = R_\tau, \ldots, R$, we define the following terms:

$$
\begin{aligned}
x_r &:= \mathbb{E}[\overline{\Delta}_{t_r,\max}],\\
\alpha_r &:= (1 - \eta_\tau(1 - \gamma p))^{t_r - t_{r-1}},\\
\beta_r &:= (1 - \eta_T)^{t_r - t_{r-1}},\\
\mathcal{I}_r &:= \{r \geq r' > R_\tau : t_{r'} - t_{r'-1} \geq 8/\eta_\tau\},\\
C_1 &:= \frac{1}{5760 \log\left(\frac{180B}{\eta_T(1-\gamma)}\right)(1 - \gamma p)} \cdot \frac{\nu}{\nu + 1},\\
C_2 &:= \sqrt{\frac{32\eta_T}{3BM(1-\gamma)}},\\
C_3 &:= \frac{1}{240 \log\left(\frac{180BM}{\eta_T(1-\gamma)}\right)} \cdot \frac{\nu}{\nu + \sqrt{M}}.
\end{aligned}
$$

With these notations in place, the recursion in (47) can be rewritten as

$$x_r \geq \alpha_r x_{r-1} - \beta_r C_2 + C_3 \mathbb{1}\{r \in \mathcal{I}_r\} + (1 - \alpha_r)C_1 \mathbb{1}\{r \in \mathcal{I}_r\},\tag{52}$$

for all $r \geq R_\tau$. We claim that $x_r$ satisfies the following relation for all $r \geq R_\tau + 1$ (whose proof is deferred to the end of this step):

$$
\begin{aligned}
x_r \geq{}& \left(\prod_{i=R_\tau+1}^{r} \alpha_i\right) x_{R_\tau} - \sum_{k=R_\tau+1}^{r} \beta_k \left(\prod_{i=k+1}^{r} \alpha_i\right) C_2 + \sum_{k=R_\tau+1}^{r} \left(\prod_{i=k+1}^{r} \alpha_i\right) \mathbb{1}\{k \in \mathcal{I}_k\}C_3\\
&+ C_1 \left(\prod_{i\notin\mathcal{I}_r} \alpha_i\right)\left(1 - \prod_{i\in\mathcal{I}_r} \alpha_i\right),
\end{aligned}\tag{53}
$$

where we recall that if there is no valid index for a product, its value is taken to be $1$.

Invoking (53) for $r = R$ and using the relation $x_{R_\tau-1} \geq 0$, we obtain,

$$
\begin{aligned}
x_R \geq{}& - \sum_{k=R_\tau}^{R} \beta_k \left(\prod_{i=k+1}^{R} \alpha_i\right) C_2 + \sum_{k=R_\tau}^{R} \left(\prod_{i=k+1}^{R} \alpha_i\right) C_3 \mathbb{1}\{k \in \mathcal{I}_k\} + C_1 \left(\prod_{i\notin\mathcal{I}_R} \alpha_i\right)\left(1 - \prod_{i\in\mathcal{I}_R} \alpha_i\right)\\
\geq{}& -RC_2 + C_1 \left(\prod_{i\notin\mathcal{I}_R} \alpha_i\right)\left(1 - \prod_{i\in\mathcal{I}_R} \alpha_i\right)\\
\geq{}& -R \cdot \sqrt{\frac{32\eta_T}{3BM(1-\gamma)}} + \left(\prod_{i\notin\mathcal{I}_R} \alpha_i\right)\left(1 - \prod_{i\in\mathcal{I}_R} \alpha_i\right) \cdot \frac{1}{5760 \log\left(\frac{180B}{\eta_T(1-\gamma)}\right)(1 - \gamma p)} \cdot \frac{\nu}{\nu + 1},
\end{aligned}\tag{54}
$$

where we used the fact $\beta_k \left(\prod_{i=k+1}^{R} \alpha_i\right) \leq 1$ and that $C_3 \geq 0$. Consider the expression

$$\prod_{i\notin\mathcal{I}_R} \alpha_i = \prod_{i\notin\mathcal{I}_R} (1 - \eta_\tau(1 - \gamma p))^{t_i - t_{i-1}} \geq 1 - \eta_\tau(1 - \gamma p) \cdot \underbrace{\sum_{i\notin\mathcal{I}_R} (t_i - t_{i-1})}_{=:T_1}.\tag{55}$$

Consequently,

$$\left(1 - \prod_{i \in \mathcal{I}_R} \alpha_i\right) = 1 - (1 - \eta_\tau (1 - \gamma p))^{T - \tau - T_1} \geq 1 - \exp\left(-\eta_\tau (1 - \gamma p)\left(T - \tau - T_1\right)\right). \quad (56)$$

Note that $T_1$ satisfies the following bound

$$T_1 := \sum_{i \notin \mathcal{I}_R} (t_i - t_{i-1}) \leq (R - |\mathcal{I}_R|) \cdot \frac{8}{\eta_\tau} \leq \frac{8R}{\eta_\tau}. \quad (57)$$

We split the remainder of the analysis based on the step size schedule.

- For the constant step size schedule, i.e., $\eta_t = \eta \geq \frac{1}{(1 - \gamma)T}$, we have, $R_\tau = 0$, with $\tau = 0$ and $t_0 = 0$ (as all agents start at the same point). If $R \leq \frac{1}{96000(1-\gamma)\log\left(\frac{180B}{\eta(1-\gamma)}\right)}$, then, (55), (56) and (57) yield the following relations:

$$T_1 \leq \frac{8R}{\eta} \leq \frac{T}{12000 \log(180N)},$$

$$\prod_{i \notin \mathcal{I}_R} \alpha_i \geq 1 - \eta(1 - \gamma p) \cdot T_1 \geq 1 - \frac{32R(1 - \gamma)}{3} \geq 1 - \frac{1}{9000 \log(180N)},$$

$$\left(1 - \prod_{i \in \mathcal{I}_R} \alpha_i\right) \geq 1 - \exp\left(-\eta(1 - \gamma p)\left(T - T_1\right)\right) \geq 1 - \exp\left(-\frac{4}{3}\left(1 - \frac{1}{9000 \log(180N)}\right)\right).$$

On plugging the above relations into (54), we obtain

$$x_R \geq \frac{\sqrt{40}}{96000 \log\left(\frac{180B}{\eta(1-\gamma)}\right)(1 - \gamma)} \cdot \left(\frac{\nu}{\nu + 1} - \frac{\nu}{5\sqrt{M}}\right) \quad (58)$$

where recall that $\nu := \sqrt{\frac{20\eta}{3B(1-\gamma)}}$. Consider the function $f(x) = \frac{x}{x+1} - \frac{x}{5\sqrt{M}}$. We claim that for $x \in [0, \sqrt{M}]$ and all $M \geq 2$,

$$f(x) \geq \frac{7}{20} \min\{x, 1\}. \quad (59)$$

The proof of the above claim is deferred to the end of the section. In light of the above claim, we have,

$$x_R \geq \frac{\sqrt{40}}{96000 \log\left(\frac{180B}{\eta(1-\gamma)}\right)(1 - \gamma)} \cdot \frac{7}{20} \cdot \min\left\{1, \sqrt{\frac{20\eta}{3B(1-\gamma)}}\right\}$$

$$\geq \frac{\sqrt{40}}{96000 \log(180N)} \cdot \frac{7}{20} \cdot \min\left\{\frac{1}{1 - \gamma}, \sqrt{\frac{20}{3(1-\gamma)^4 N}}\right\}, \quad (60)$$

where we used the fact that $M \geq 2$, $\frac{\sqrt{x}}{\log(1/x)}$ is an increasing function and the relation $\frac{\nu}{M} = \frac{20\eta}{3BM(1-\gamma)} \leq \frac{1}{15} \leq 1$.

- Next, we consider the rescaled linear step size schedule, where $\tau = T/3$ (cf. (34)). To begin, we assume $t_{R_\tau} \leq \max\{\frac{3T}{4}, T - \frac{1}{6\eta_\tau(1-\gamma p)}\}$. It is straightforward to note that

$$\max\left\{\frac{3T}{4}, T - \frac{1}{6\eta_\tau(1 - \gamma p)}\right\} = \begin{cases} \frac{3T}{4} & \text{if } c_\eta \geq 3 \\ T - \frac{1}{6\eta_\tau(1 - \gamma p)} & \text{if } c_\eta < 3. \end{cases}$$

If $R \leq \frac{1}{384000(1-\gamma)\log\left(\frac{180B}{\eta_T(1-\gamma)}\right)\cdot(5+c_\eta)}$ then, (55), (56) and (57) yield the following relations:

$$T_1 \leq \frac{8R}{\eta_\tau}, \qquad \prod_{i \notin \mathcal{I}_R} \alpha_i \geq 1 - \eta_\tau(1 - \gamma p) \cdot T_1 \geq 1 - \frac{32R(1 - \gamma)}{3} \geq 1 - \frac{1}{36000}.$$

For $c_\eta \geq 3$, we have,

$$\left(1 - \prod_{i \in \mathcal{I}_R} \alpha_i\right) \geq 1 - \exp\left(-\eta_\tau(1 - \gamma p)\left(T - t_{R_\tau} - T_1\right)\right)$$

$$\geq 1 - \exp\left(-\frac{(1-\gamma)T}{(3 + c_\eta(1-\gamma)T)} + \frac{32R(1-\gamma)}{3}\right)$$

$$\geq \frac{1}{2(3 + c_\eta)},$$

where we used $T \geq \frac{1}{1-\gamma}$ in the second step. Similarly, for $c_\eta < 3$, we have,

$$\left(1 - \prod_{i \in \mathcal{I}_R} \alpha_i\right) \geq 1 - \exp\left(-\eta_\tau(1 - \gamma p)\left(T - t_{R_\tau} - T_1\right)\right)$$

$$\geq 1 - \exp\left(-\frac{1}{6} + \frac{32R(1-\gamma)}{3}\right)$$

$$\geq \frac{1}{10}.$$

On plugging the above relations into (54), we obtain

$$x_R \geq \frac{18\sqrt{1.6}}{384000 \log\left(\frac{180B}{\eta_T(1-\gamma)}\right)(1-\gamma)(5+c_\eta)} \cdot \left(\frac{\nu}{\nu+1} - \frac{\nu}{18\sqrt{M}}\right)$$

$$\geq \frac{18\sqrt{1.6}}{384000 \log\left(\frac{180B}{\eta_T(1-\gamma)}\right)(1-\gamma)(5+c_\eta)} \cdot \frac{7}{20} \cdot \min\left\{1, \sqrt{\frac{20\eta_T}{3B(1-\gamma)}}\right\}$$

$$\geq \frac{18\sqrt{1.6}}{384000 \log\left(\frac{180B}{\eta_T(1-\gamma)}\right)(5+c_\eta)} \cdot \frac{7}{20} \cdot \min\left\{\frac{1}{1-\gamma}, \sqrt{\frac{20\eta_T}{3B(1-\gamma)^3}}\right\}$$

$$\geq \frac{18\sqrt{1.6}}{384000 \log\left(180N(1+\log N)\right)(5+\log N)} \cdot \frac{7}{20} \cdot \min\left\{\frac{1}{1-\gamma}, \sqrt{\frac{20}{3B(1+\log N)(1-\gamma)^4 N}}\right\},$$

$$(61)$$

where we again used the facts that $M \geq 2$, $c_\eta \leq \log N$, $\frac{\sqrt{x}}{\log(1/x)}$ is an increasing function and the relation $\frac{\nu}{M} = \frac{20\eta_T}{3BM(1-\gamma)} \leq 1$.

- Last but not least, let us consider the rescaled linear step size schedule case when $t_{R_\tau} > \max\{\frac{3T}{4}, T - \frac{1}{6\eta_\tau(1-\gamma p)}\}$. The condition implies that the time between the communication rounds $R_\tau - 1$ and $R_\tau$ is at least $T_0 := \max\{\frac{5T}{12}, \frac{2T}{3} - \frac{1}{6\eta_\tau(1-\gamma p)}\}$. Thus, (47) yields that

$$\mathbb{E}[\overline{\Delta}_{t_{R_\tau}}] \geq \left(\frac{1 - (1 - \eta_\tau(1-\gamma p))^{T_0}}{5760 \log\left(\frac{180}{B\eta_T(1-\gamma)}\right)(1-\gamma p)}\right) \cdot \frac{\nu}{\nu+1} - 2(1-\eta_T)^{T_0}\sqrt{\frac{8\eta_T}{3BM(1-\gamma)}}.$$

$$(62)$$

Using the above relation along with (53), we can conclude that

$$x_R \geq (1 - \eta_\tau(1-\gamma p))^{T-t_{R_\tau}}\left(\frac{1 - (1 - \eta_\tau(1-\gamma p))^{T_0}}{5760 \log\left(\frac{180}{B\eta_T(1-\gamma)}\right)(1-\gamma p)}\right) \cdot \frac{\nu}{\nu+1}$$

$$- 2(1-\eta_T)^{T_0} \cdot (1 - \eta_\tau(1-\gamma p))^{T-t_{R_\tau}}\sqrt{\frac{8\eta_T}{3BM(1-\gamma)}} - RC_2. \quad (63)$$

In the above relation, we used the trivial bounds $C_1, C_3 \geq 0$ and a crude bound on the term corresponding to $C_2$, similar to (54). Let us first consider the case of $c_\eta \geq 3$. We have,

$$1 - (1 - \eta_\tau(1 - \gamma p))^{T_0} \geq 1 - \exp\left(-\eta_\tau(1 - \gamma p)5T/12\right) \geq 1 - \exp\left(-\frac{5(1 - \gamma)T}{3(3 + c_\eta(1 - \gamma)T)}\right) \geq \frac{1}{3 + c_\eta},$$

$$(1 - \eta_\tau(1 - \gamma p))^{T - t_{R_\tau}} \geq 1 - \eta_\tau(1 - \gamma p)\frac{T}{4} \geq 1 - \frac{(1 - \gamma)T}{(3 + c_\eta(1 - \gamma)T)} \geq 1 - \frac{1}{c_\eta} \geq \frac{2}{3}.$$

Similarly, for $c_\eta < 3$, we have,

$$1 - (1 - \eta_\tau(1 - \gamma p))^{T_0} \geq 1 - \exp\left(-\eta_\tau(1 - \gamma p)\frac{2T}{3} + \frac{1}{6}\right)$$

$$\geq 1 - \exp\left(-\frac{8(1 - \gamma)T}{3(3 + c_\eta(1 - \gamma)T)} + \frac{1}{6}\right) \geq 1 - e^{-5/18},$$

$$(1 - \eta_\tau(1 - \gamma p))^{T - t_{R_\tau}} \geq 1 - \frac{\eta_\tau(1 - \gamma p)}{6\eta_\tau(1 - \gamma p)} \geq \frac{5}{6}.$$

The above relations implies that $(1 - \eta_\tau(1 - \gamma p))^{T - t_{R_\tau}}(1 - (1 - \eta_\tau(1 - \gamma p))^{T_0}) \geq c$ for some constant $c$, which only depends on $c_\eta$. On plugging this into (63), we obtain a relation that is identical to that in (54) up to leading constants. Thus, by using a similar sequence of argument as used to obtain (61), we arrive at the same conclusion as for the case of $t_{R_\tau} \leq \max\{\frac{3T}{4}, T - \frac{1}{6\eta_\tau(1 - \gamma p)}\}$.

**Step 4: finishing up the proof.** Thus, (60), (61) along with the above conclusion together imply that there exists a numerical constant $c_0 > 0$ such that

$$\mathbb{E}[|\widehat{V}_T^m(1) - V^\star(1)|] \geq \mathbb{E}[\overline{\Delta}_{T,\max}] \geq \frac{c_0}{\log^3 N} \cdot \min\left\{\frac{1}{1 - \gamma}, \sqrt{\frac{1}{(1 - \gamma)^4 N}}\right\}. \qquad (64)$$

The above equation along with Lemma 2 implies

$$\mathbb{E}[|V_T^m - V^\star(1)|] \geq \frac{c_0}{\log^3 N} \cdot \min\left\{\frac{1}{1 - \gamma}, \sqrt{\frac{1}{(1 - \gamma)^4 N}}\right\} - \frac{1}{1 - \gamma}\prod_{i=1}^{T}(1 - \eta_i(1 - \gamma)). \qquad (65)$$

On the other hand, from (21) we know that

$$\mathbb{E}[|V_T^m(3) - V^\star(3)|] \geq \frac{1}{1 - \gamma}\prod_{i=1}^{T}(1 - \eta_i(1 - \gamma)). \qquad (66)$$

Hence,

$$\mathbb{E}[\|Q_T^m - Q^\star\|_\infty] \geq \mathbb{E}\left[\max\left\{|V_T^m(3) - V^\star(3)|, |V_T^m(1) - V^\star(1)|\right\}\right]$$

$$\geq \max\left\{\mathbb{E}\left[|V_T^m(3) - V^\star(3)|\right], \mathbb{E}\left[|V_T^m(1) - V^\star(1)|\right]\right\}$$

$$\geq \max\left\{\frac{1}{1 - \gamma}\prod_{i=1}^{T}(1 - \eta_i(1 - \gamma)), \min\left\{\frac{1}{1 - \gamma}, \sqrt{\frac{1}{(1 - \gamma)^4 N}}\right\} - \frac{1}{1 - \gamma}\prod_{i=1}^{T}(1 - \eta_i(1 - \gamma))\right\}$$

$$\geq \frac{1}{2}\min\left\{\frac{1}{1 - \gamma}, \sqrt{\frac{1}{(1 - \gamma)^4 N}}\right\}, \qquad (67)$$

where the third step follows from (65) and (66) and the fourth step uses $\max\{a, b\} \geq (a + b)/2$.

Thus, from (28) and (67) we can conclude that whenever $\mathsf{CC}_{\mathsf{round}} = \mathcal{O}\left(\frac{1}{(1 - \gamma)\log^2 N}\right)$, $\mathsf{ER}(\mathscr{A}; N, M) = \Omega\left(\frac{1}{\log^3 N\sqrt{N}}\right)$ for all values of $M \geq 2$. In other words, for any algorithm to achieve any collaborative gain, its communication complexity should satisfy $\mathsf{CC}_{\mathsf{round}} = \Omega\left(\frac{1}{(1 - \gamma)\log^2 N}\right)$, as required.

**Proof of (53).** We now return to establish (53) using induction. For the base case, (52) yields

$$x_{R_\tau+1} \geq \alpha_{R_\tau+1} x_{R_\tau} - \beta_{R_\tau+1} C_2 + C_3 \mathbb{1}\{R_\tau + 1 \in \mathcal{I}_{R_\tau+1}\} + (1 - \alpha_{R_\tau+1}) C_1 \mathbb{1}\{R_\tau + 1 \in \mathcal{I}_{R_\tau+1}\}.$$

$$(68)$$

Note that this is identical to the expression in (53) for $r = R_\tau + 1$ as

$$\left( \prod_{i \notin \mathcal{I}_{R_\tau+1}} \alpha_i \right) \left( 1 - \prod_{i \in \mathcal{I}_{R_\tau+1}} \alpha_i \right) = (1 - \alpha_{R_\tau+1}) \mathbb{1}\{R_\tau + 1 \in \mathcal{I}_{R_\tau+1}\}$$

based on the adopted convention for products with no valid indices. For the induction step, assume (53) holds for some $r \geq R_\tau + 1$. On combining (52) and (53), we obtain,

$$x_{r+1} \geq \alpha_{r+1} x_r - \beta_{r+1} C_2 + C_3 \mathbb{1}\{(r+1) \in \mathcal{I}_{r+1}\} + (1 - \alpha_{r+1}) C_1 \mathbb{1}\{r + 1 \in \mathcal{I}_{r+1}\}$$

$$\geq \alpha_{r+1} \left( \prod_{i=R_\tau+1}^{r} \alpha_i \right) x_{R_\tau} - \alpha_{r+1} \sum_{k=R_\tau+1}^{r} \beta_k \left( \prod_{i=k+1}^{r} \alpha_i \right) C_2 + \alpha_{r+1} \sum_{k=R_\tau+1}^{r} \left( \prod_{i=k+1}^{r} \alpha_i \right) C_3 \mathbb{1}\{k \in \mathcal{I}_k\}$$

$$+ \alpha_{r+1} C_1 \left( \prod_{i \notin \mathcal{I}_r} \alpha_i \right) \left( 1 - \prod_{i \in \mathcal{I}_r} \alpha_i \right) - \beta_{r+1} C_2 + C_3 \mathbb{1}\{(r+1) \in \mathcal{I}_{r+1}\} + (1 - \alpha_{r+1}) C_1 \mathbb{1}\{(r+1) \in \mathcal{I}_{r+1}\}$$

$$\geq \left( \prod_{i=R_\tau+1}^{r+1} \alpha_i \right) x_{R_\tau} - \sum_{k=R_\tau+1}^{r+1} \beta_k \left( \prod_{i=k+1}^{r+1} \alpha_i \right) C_2 + \sum_{k=R_\tau+1}^{r+1} \left( \prod_{i=k+1}^{r+1} \alpha_i \right) C_3 \mathbb{1}\{k \in \mathcal{I}_k\}$$

$$+ \alpha_{r+1} C_1 \left( \prod_{i \notin \mathcal{I}_r} \alpha_i \right) \left( 1 - \prod_{i \in \mathcal{I}_r} \alpha_i \right) + (1 - \alpha_{r+1}) C_1 \mathbb{1}\{(r+1) \in \mathcal{I}_{r+1}\}. \qquad (69)$$

If $(r + 1) \notin \mathcal{I}_{r+1}$, then $\left(1 - \prod_{i \in \mathcal{I}_r} \alpha_i\right) = \left(1 - \prod_{i \in \mathcal{I}_{r+1}} \alpha_i\right)$ and $\alpha_{r+1} \left( \prod_{i \notin \mathcal{I}_r} \alpha_i \right) = \left( \prod_{i \notin \mathcal{I}_{r+1}} \alpha_i \right)$. Consequently,

$$\alpha_{r+1} C_1 \left( \prod_{i \notin \mathcal{I}_r} \alpha_i \right) \left( 1 - \prod_{i \in \mathcal{I}_r} \alpha_i \right) + (1 - \alpha_{r+1}) C_1 \mathbb{1}\{(r+1) \in \mathcal{I}_{r+1}\} = C_1 \left( \prod_{i \notin \mathcal{I}_{r+1}} \alpha_i \right) \left( 1 - \prod_{i \in \mathcal{I}_{r+1}} \alpha_i \right).$$

$$(70)$$

On the other hand, if $(r + 1) \in \mathcal{I}_{r+1}$, then $\left(\prod_{i \notin \mathcal{I}_r} \alpha_i\right) = \left(\prod_{i \notin \mathcal{I}_{r+1}} \alpha_i\right)$. Consequently, we have,

$$\alpha_{r+1} C_1 \left( \prod_{i \notin \mathcal{I}_r} \alpha_i \right) \left( 1 - \prod_{i \in \mathcal{I}_r} \alpha_i \right) + (1 - \alpha_{r+1}) C_1 \mathbb{1}\{(r+1) \in \mathcal{I}_{r+1}\}$$

$$= \alpha_{r+1} C_1 \left( \prod_{i \notin \mathcal{I}_{r+1}} \alpha_i \right) \left( 1 - \prod_{i \in \mathcal{I}_r} \alpha_i \right) + (1 - \alpha_{r+1}) C_1$$

$$\geq C_1 \left( \prod_{i \notin \mathcal{I}_{r+1}} \alpha_i \right) \left[ \alpha_{r+1} \left( 1 - \prod_{i \in \mathcal{I}_r} \alpha_i \right) + (1 - \alpha_{r+1}) \right]$$

$$\geq C_1 \left( \prod_{i \notin \mathcal{I}_{r+1}} \alpha_i \right) \left( 1 - \prod_{i \in \mathcal{I}_{r+1}} \alpha_i \right). \qquad (71)$$

Combining (69), (70) and (71) proves the claim.

**Proof of (59).** To establish this result, we separately consider the cases $x \leq 1$ and $x \geq 1$.

- When $x \leq 1$, we have

$$f(x) = \frac{x}{x+1} - \frac{1}{5\sqrt{M}} \geq x \cdot \left( \frac{1}{2} - \frac{x}{5\sqrt{M}} \right) \geq \frac{7x}{20}, \qquad (72)$$

where in the last step, we used the relation $M \geq 2$.

- Let us now consider the case $x \geq 1$. The second derivative of $f$ is given by $f''(x) = -\frac{1}{2(x+1)^3}$. Clearly, for all $x \geq 1$, $f'' < 0$ implying that $f$ is a concave function. It is well-known that a continuous, bounded, concave function achieves its minimum values over a compact interval at the end points of the interval (Bauer's minimum principle). For all $M \geq 2$, we have,

$$f(1) = \frac{1}{2} - \frac{1}{5\sqrt{M}} \geq \frac{7}{20}; \quad f(\sqrt{M}) = \frac{\sqrt{M}}{\sqrt{M}+1} - \frac{1}{5} \geq \frac{7}{20}.$$

Consequently, we can conclude that for all $x \in [1, \sqrt{M}]$,

$$f(x) \geq \frac{7}{20}. \tag{73}$$

Combining (72) and (73) proves the claim.

### B.3.3 Large learning rates with large $\frac{\eta_T}{BM}$

In order to bound the error in this scenario, note that $\frac{\eta_T}{BM}$ controls the variance of the stochastic updates in the fixed point iteration. Thus, when $\frac{\eta_T}{BM}$ is large, the variance of the iterates is large, resulting in a large error. To demonstrate this effect, we focus on the dynamics of state 2. This part of the proof is similar to the large learning rate case of Li et al. [2023]. For all $t \in [T]$, define:

$$\overline{V}_t(2) := \frac{1}{M} \sum_{m=1}^{M} V_t^m(2). \tag{74}$$

Thus, from (33), we know that $\mathbb{E}[\overline{V}_t(2)]$ obeys the following recursion:

$$\mathbb{E}[\overline{V}_t(2)] = (1 - \eta_t(1 - \gamma p))\mathbb{E}[\overline{V}_{t-1}(2)] + \eta_t.$$

Upon unrolling the recursion, we obtain,

$$\mathbb{E}[\overline{V}_T(2)] = \left( \prod_{k=t+1}^{T} (1 - \eta_k(1 - \gamma p)) \right) \mathbb{E}[\overline{V}_t(2)] + \sum_{k=t+1}^{T} \eta_k^{(T)}.$$

Thus, the above relation along with (16) and the value of $V^\star(2)$ yields us,

$$V^\star(2) - \mathbb{E}[\overline{V}_T(2)] = \prod_{k=t+1}^{T} (1 - \eta_k(1 - \gamma p)) \left( \frac{1}{1 - \gamma p} - \mathbb{E}[\overline{V}_t(2)] \right). \tag{75}$$

Similar to Li et al. [2023], we define

$$\tau' := \min \left\{ 0 \leq t' \leq T - 2 \,\middle|\, \mathbb{E}[(\overline{V}_t)^2] \geq \frac{1}{4(1 - \gamma)^2} \text{ for all } t' + 1 \leq t \leq T \right\}.$$

If such a $\tau'$ does not exist, it implies that either $\mathbb{E}[(\overline{V}_T)^2] < \frac{1}{4(1-\gamma)^2}$ or $\mathbb{E}[(\overline{V}_{T-1})^2] < \frac{1}{4(1-\gamma)^2}$. If the former is true, then,

$$V^\star(2) - \mathbb{E}[\overline{V}_T(2)] = \frac{3}{4(1 - \gamma)} - \sqrt{\mathbb{E}[(\overline{V}_T)^2]} > \frac{1}{4(1 - \gamma)}. \tag{76}$$

Similarly, if $\mathbb{E}[(\overline{V}_{T-1})^2] < \frac{1}{4(1-\gamma)^2}$, it implies $\mathbb{E}[\overline{V}_{T-1}] < \frac{1}{2(1-\gamma)}$. Using (33), we have,

$$\mathbb{E}[\overline{V}_T(2)] = (1 - \eta_T(1 - \gamma p))\mathbb{E}[\overline{V}_{T-1}(2)] + \eta_T \leq \mathbb{E}[\overline{V}_{T-1}(2)] + 1 < \frac{1}{2(1 - \gamma)} + \frac{1}{6(1 - \gamma)} = \frac{2}{3(1 - \gamma)}.$$

Consequently,

$$V^\star(2) - \mathbb{E}[\overline{V}_T(2)] > \frac{3}{4(1 - \gamma)} - \frac{2}{3(1 - \gamma)} > \frac{1}{12(1 - \gamma)}. \tag{77}$$

For the case when $\tau'$ exists, we divide the proof into two cases.

- We first consider the case when the learning rates satisfy:

$$\prod_{k=\tau'+1}^{T} (1 - \eta_k(1 - \gamma p)) \geq \frac{1}{2}. \tag{78}$$

The analysis for this case is identical to that considered in Li et al. [2023]. We explicitly write the steps for completeness. Specifically,

$$
\begin{aligned}
V^\star(2) - \mathbb{E}[\overline{V}_T(2)] &= \left( \prod_{k=\tau'+1}^{T} (1 - \eta_k(1 - \gamma p)) \right) \left( \frac{1}{1 - \gamma p} - \mathbb{E}[\overline{V}_{\tau'}(2)] \right) \\
&\geq \frac{1}{2} \cdot \left( \frac{3}{4(1 - \gamma)} - \sqrt{\mathbb{E}[(\overline{V}_{\tau'}(2))^2]} \right) \\
&\geq \frac{1}{2} \cdot \left( \frac{3}{4(1 - \gamma)} - \frac{1}{2(1 - \gamma)} \right) \geq \frac{1}{8(1 - \gamma)},
\end{aligned} \tag{79}
$$

where the first line follows from (75), the second line from the condition on step sizes and the third line from the definition of $\tau'$.

- We now consider the other case where,

$$0 \leq \prod_{k=\tau'+1}^{T} (1 - \eta_k(1 - \gamma p)) < \frac{1}{2}. \tag{80}$$

Using [Li et al., 2023, Eqn.(134)], for any $t' < t$ and all agents $m$, we have the relation

$$
\begin{aligned}
V_t^m(2) = \frac{1}{1 - \gamma p} &- \prod_{k=t'+1}^{t} (1 - \eta_k(1 - \gamma p)) \left( \frac{1}{1 - \gamma p} - V_{t''}^m(2) \right) \\
&+ \sum_{k=t'+1}^{t} \eta_k^{(t)} \gamma (\hat{P}_k^m(2|2) - p) V_{k-1}^m(2).
\end{aligned}
$$

The above equation is directly obtained by unrolling the recursion in (24) along with noting that $Q_t(2, 1) = V_t(2)$ for all $t$. Consequently, we have,

$$
\begin{aligned}
\overline{V}_T(2) = \frac{1}{1 - \gamma p} &- \prod_{k=t'+1}^{T} (1 - \eta_k(1 - \gamma p)) \left( \frac{1}{1 - \gamma p} - \overline{V}_{t'}(2) \right) \\
&+ \frac{1}{M} \sum_{m=1}^{M} \sum_{k=t'+1}^{T} \eta_k^{(T)} \gamma (\hat{P}_k^m(2|2) - p) V_{k-1}^m(2).
\end{aligned} \tag{81}
$$

Let $\{\mathscr{F}_t\}_{t=0}^{T}$ be a filtration such that $\mathscr{F}_t$ is the $\sigma$-algebra corresponding to $\{\{\hat{P}_s^m(2|2)\}_{m=1}^{M}\}_{s=1}^{t}$. It is straightforward to note that $\left\{ \frac{1}{M} \sum_{m=1}^{M} \eta_k^{(T)} \gamma (\hat{P}_k^m(2|2) - p) V_{k-1}^m(2) \right\}_k$ is a martingale sequence adapted to the filtration $\mathscr{F}_k$. Thus, using the result from [Li et al., 2023, Eqn.(139)], we can conclude that

$$\text{Var}(\overline{V}_T(2)) \geq \mathbb{E}\left[ \sum_{k=\tau'+2}^{T} \text{Var}\left( \frac{1}{M} \sum_{m=1}^{M} \eta_k^{(T)} \gamma (\hat{P}_k^m(2|2) - p) V_{k-1}^m(2) \,\Big|\, \mathscr{F}_{k-1} \right) \right]. \tag{82}$$

We have,

$$
\begin{aligned}
&\text{Var}\left( \frac{1}{M} \sum_{m=1}^{M} \eta_k^{(T)} \gamma (\hat{P}_k^m(2|2) - p) V_{k-1}^m(2) \,\Big|\, \mathscr{F}_{k-1} \right) \\
&= \frac{1}{M^2} \sum_{m=1}^{M} \text{Var}\left( \eta_k^{(T)} \gamma (\hat{P}_k^m(2|2) - p) V_{k-1}^m(2) \,\Big|\, \mathscr{F}_{k-1} \right) \\
&= \frac{(\eta_k^{(T)})^2}{BM} \gamma^2 p(1 - p) \left( \frac{1}{M} \sum_{m=1}^{M} (V_{k-1}^m(2))^2 \right)
\end{aligned}
$$

$$\geq \frac{(1-\gamma)(4\gamma-1)}{9BM} \cdot (\eta_k^{(T)})^2 \cdot (\overline{V}_{k-1}(2))^2, \tag{83}$$

where the first line follows from that fact that variance of sum of i.i.d. random variables is the sum of their variances, the second line from variance of Binomial random variable and the third line from Jensen's inequality. Thus, (82) and (83) together yield,

$$\mathrm{Var}(\overline{V}_T(2)) \geq \frac{(1-\gamma)(4\gamma-1)}{9BM} \cdot \sum_{k=\tau'+2}^{T} (\eta_k^{(T)})^2 \cdot \mathbb{E}[(\overline{V}_{k-1}(2))^2]$$

$$\geq \frac{(1-\gamma)(4\gamma-1)}{9BM} \cdot \frac{1}{4(1-\gamma)^2} \cdot \sum_{k=\max\{\tau,\tau'\}+2}^{T} (\eta_k^{(T)})^2, \tag{84}$$

where the second line follows from the definition of $\tau'$. We focus on bounding the third term in the above relation. We have,

$$\sum_{k=\max\{\tau',\tau\}+2}^{T} \left(\eta_k^{(T)}\right)^2 \geq \sum_{k=\max\{\tau',\tau\}+2}^{T} \left(\eta_k \prod_{i=k+1}^{T} (1-\eta_i(1-\gamma p))\right)^2$$

$$\geq \sum_{k=\max\{\tau',\tau\}+2}^{T} \left(\eta_T \prod_{i=k+1}^{t} (1-\eta_\tau(1-\gamma p))\right)^2$$

$$= \eta_T^2 \sum_{k=\max\{\tau',\tau\}+2}^{T} (1-\eta_\tau(1-\gamma p))^{2(t-k)}$$

$$\geq \eta_T^2 \cdot \frac{1 - (1-\eta_\tau(1-\gamma p))^{2(T-\max\{\tau',\tau\}-1)}}{\eta_\tau(1-\gamma p)(2 - \eta_\tau(1-\gamma p))}$$

$$\geq \eta_T \cdot \frac{1}{4(1-\gamma)} \cdot c', \tag{85}$$

where the second line follows from monotonicity of $\eta_t$ and the numerical constant $c'$ in the fifth step is given by the following claim whose proof is deferred to the end of the section:

$$1 - (1-\eta_\tau(1-\gamma p))^{2(T-\max\{\tau',\tau\}-1)} \geq \begin{cases} 1 - e^{-8/9} & \text{for constant step sizes,} \\ 1 - \exp\left(-\frac{8}{3\max\{1,c_\eta\}}\right) & \text{for linearly rescaled step sizes} \end{cases}. \tag{86}$$

Thus, (84) and (85) together imply

$$\mathrm{Var}(\overline{V}_T(2)) \geq \frac{(4\gamma-1)}{36BM(1-\gamma)} \cdot \sum_{k=\tau'+2}^{T} (\eta_k^{(T)})^2$$

$$\geq \frac{c'(4\gamma-1)}{144(1-\gamma)} \cdot \frac{\eta_T}{BM(1-\gamma)} \geq \frac{c'(4\gamma-1)}{144(1-\gamma)} \cdot \frac{1}{100}, \tag{87}$$

where the last inequality follows from the bound on $\frac{\eta_T}{BM}$.

Thus, for all $N \geq 1$, we have,

$$\mathbb{E}[(V^\star(2) - \overline{V}_T(2))^2] = \mathbb{E}[(V^\star(2) - \mathbb{E}[\overline{V}_T(2)])^2] + \mathrm{Var}(\overline{V}_T(2)) \geq \frac{c''}{(1-\gamma)N},$$

for some numerical constant $c''$. Similar to the small learning rate case, the error rate is bounded away from a constant value irrespective of the number of agents and the number of communication rounds. Thus, even with $\mathsf{CC}_{\mathsf{round}} = \Omega(T)$, we will not observe any collaborative gain in this scenario.

**Proof of** (86). To establish the claim, we consider two cases:

- $\tau' \geq \tau$: Under this case, we have,

$$(1 - \eta_\tau(1 - \gamma p))^{2(T - \max\{\tau', \tau\} - 1)} = (1 - \eta_\tau(1 - \gamma p))^{2(T - \tau' - 1)}$$

$$\leq (1 - \eta_\tau(1 - \gamma p))^{T - \tau'} \leq \prod_{k = \tau' + 1}^{T} (1 - \eta_k(1 - \gamma p)) \leq \frac{1}{2}, \tag{88}$$

where the last inequality follows from (80).

- $\tau \geq \tau'$: For this case, we have

$$(1 - \eta_\tau(1 - \gamma p))^{2(T - \max\{\tau', \tau\} - 1)} = (1 - \eta_\tau(1 - \gamma p))^{2(T - \tau - 1)}$$

$$\leq (1 - \eta_\tau(1 - \gamma p))^{T - \tau} \leq \exp\left(-\frac{2T\eta_\tau(1 - \gamma p)}{3}\right). \tag{89}$$

For the constant stepsize schedule, we have,

$$\exp\left(-\frac{2T\eta_\tau(1 - \gamma p)}{3}\right) \leq \exp\left(-\frac{2T}{3} \cdot \frac{1}{(1 - \gamma)T} \cdot \frac{4(1 - \gamma)}{3}\right) = \exp\left(-\frac{8}{9}\right) \tag{90}$$

For linearly rescaled stepsize schedule, we have,

$$\exp\left(-\frac{2T\eta_\tau(1 - \gamma p)}{3}\right) \leq \exp\left(-\frac{2T}{3} \cdot \frac{1}{1 + c_\eta(1 - \gamma)T/3} \cdot \frac{4(1 - \gamma)}{3}\right) = \exp\left(-\frac{8}{3\max\{1, c_\eta\}}\right) \tag{91}$$

On combining (88), (89), (90) and (91), we arrive at the claim.

## B.4 Generalizing to larger state action spaces

We now elaborate on how we can extend the result to general state-action spaces along with the obtaining the lower bound on the bit level communication complexity. For the general case, we instead consider the following MDP. For the first four states $\{0, 1, 2, 3\}$, the probability transition kernel and reward function are given as follows.

$$
\begin{array}{llll}
\mathcal{A}_0 = \{1\} & P(0|0, 1) = 1 & & r(0, 1) = 0, & (92a) \\
\mathcal{A}_1 = \{1, 2, \ldots, |\mathcal{A}|\} & P(1|1, a) = p & P(0|1, a) = 1 - p & r(1, a) = 1, \forall\, a \in \mathcal{A} & (92b) \\
\mathcal{A}_2 = \{1\} & P(2|2, 1) = p & P(0|2, 1) = 1 - p & r(2, 1) = 1, & (92c) \\
\mathcal{A}_3 = \{1\} & P(3|3, 1) = 1 & & r(3, 1) = 1, & (92d)
\end{array}
$$

where the parameter $p = \dfrac{4\gamma - 1}{3\gamma}$. The overall MDP is obtained by creating $|\mathcal{S}|/4$ copies of the above MDP for all sets of the form $\{4r, 4r + 1, 4r + 2, 4r + 3\}$ for $r \leq |\mathcal{S}|/4 - 1$. It is straightforward to note that the lower bound on the number of communication rounds immediately transfers to the general case as well. Moreover, note that the bound on $\mathsf{CC}_{\mathsf{round}}$ implies the bound $\mathsf{CC}_{\mathsf{bit}} = \Omega\left(\frac{1}{(1 - \gamma)\log^2 N}\right)$ as every communication entails sending $\Omega(1)$ bits.

To obtain the general lower bound on bit level communication complexity, note that we can carry out the analysis in the previous section for all $|\mathcal{A}|/2$ pairs of actions in state 1 corresponding to the set of states $\{0, 1, 2, 3\}$. Moreover, the algorithm $\mathscr{A}$, needs to ensure that the error is low across all the $|\mathcal{A}|/2$ pairs. Since the errors are independent across all these pairs, each of them require $\Omega\left(\frac{1}{(1 - \gamma)\log^2 N}\right)$ bits of information to be transmitted during the learning horizon leading to a lower bound of $\Omega\left(\frac{|\mathcal{A}|}{(1 - \gamma)\log^2 N}\right)$. Note that since we require a low $\ell_\infty$ error, $\mathscr{A}$ needs to ensure that the error is low across all the pairs, resulting in a communication cost linearly growing with $|\mathcal{A}|$. Upon repeating the argument across all $|\mathcal{S}|/4$ copies of the MDP, we arrive at the lower bound of $\mathsf{CC}_{\mathsf{bit}} = \Omega\left(\frac{|\mathcal{S}||\mathcal{A}|}{(1 - \gamma)\log^2 N}\right)$.

## B.5 Proofs of auxiliary lemmas

### B.5.1 Proof of Lemma 2

Note that a similar relationship is also derived in Li et al. [2023], but needing to take care of the averaging over multiple agents, we present the entire arguments for completeness. We prove the claim using an induction over $t$. It is straightforward to note that the claim is true for $t = 0$ and all agents $m \in \{1, 2, \ldots, M\}$. For the inductive step, we assume that the claim holds for $t - 1$ for all clients. Using the induction hypothesis, we have the following relation between $V_{t-1}^m(1)$ and $\widehat{V}_{t-1}^m$:

$$V_{t-1}^m(1) = \max_{a \in \{1,2\}} Q_{t-1}^m(1, a) \geq \max_{a \in \{1,2\}} \widehat{Q}_{t-1}^m(a) - \frac{1}{1-\gamma} \prod_{i=1}^{t-1}(1 - \eta_i(1-\gamma)) = \widehat{V}_{t-1}^m - \frac{1}{1-\gamma} \prod_{i=1}^{t-1}(1 - \eta_i(1-\gamma)). \tag{93}$$

For $t \notin \{t_r\}_{r=1}^R$ and $a \in \{1, 2\}$, we have,

$$\begin{aligned}
Q_t^m(1, a) - \widehat{Q}_t^m(a) &= Q_{t-1/2}^m(1, a) - \widehat{Q}_{t-1/2}^m(a) \\
&= (1 - \eta_t)Q_{t-1}^m(1, a) + \eta_t(1 + \gamma \widehat{P}_t^m(1|1, a)V_{t-1}^m(1)) \\
&\quad - \left[ (1 - \eta_t)\widehat{Q}_{t-1}^m(a) + \eta_t(1 + \gamma \widehat{P}_t^m(1|1, a)\widehat{V}_{t-1}^m) \right] \\
&= (1 - \eta_t)(Q_{t-1}^m(1|1, a) - \widehat{Q}_{t-1}^m(a)) + \eta_t \gamma \widehat{P}_t^m(1|1, a)(V_{t-1}^m(1) - \widehat{V}_{t-1}^m) \\
&\geq -\frac{(1 - \eta_t)}{1 - \gamma} \prod_{i=1}^{t-1}(1 - \eta_i(1-\gamma)) - \widehat{P}_t^m(1|1, a) \cdot \frac{\eta_t \gamma}{1 - \gamma} \prod_{i=1}^{t-1}(1 - \eta_i(1-\gamma)) \\
&\geq -\frac{(1 - \eta_t)}{1 - \gamma} \prod_{i=1}^{t-1}(1 - \eta_i(1-\gamma)) - \frac{\eta_t \gamma}{1 - \gamma} \prod_{i=1}^{t-1}(1 - \eta_i(1-\gamma)) \\
&\geq -\frac{1}{1 - \gamma} \prod_{i=1}^{t}(1 - \eta_i(1-\gamma)). \tag{94}
\end{aligned}$$

For $t \in \{t_r\}_{r=1}^R$ and $a \in \{1, 2\}$, we have,

$$\begin{aligned}
Q_t^m(1, a) - \widehat{Q}_t^m(a) &= \frac{1}{M} \sum_{m=1}^M Q_{t-1/2}^m(1, a) - \frac{1}{M} \sum_{m=1}^M \widehat{Q}_{t-1/2}^m(a) \\
&= \frac{1}{M} \sum_{m=1}^M \left[ (1 - \eta_t)Q_{t-1}^m(1, a) + \eta_t(1 + \gamma \widehat{P}_t^m(1|1, a)V_{t-1}^m(1)) \right] \\
&\quad - \frac{1}{M} \sum_{m=1}^M \left[ (1 - \eta_t)\widehat{Q}_{t-1}^m(a) + \eta_t(1 + \gamma \widehat{P}_t^m(1|1, a)\widehat{V}_{t-1}^m) \right] \\
&= \frac{1}{M} \sum_{m=1}^M \left[ (1 - \eta_t)(Q_{t-1}^m(1, a) - \widehat{Q}_{t-1}^m(a)) + \eta_t \gamma \widehat{P}_t^m(1|1, a)(V_{t-1}^m(1) - \widehat{V}_{t-1}^m) \right] \\
&\geq -\frac{1}{1 - \gamma} \prod_{i=1}^{t}(1 - \eta_i(1-\gamma)), \tag{95}
\end{aligned}$$

where the last step follows using the same set of arguments as used in (94). The inductive step follows from (94) and (95).

### B.5.2 Proof of Lemma 3

In order to bound the term $\mathbb{E}[\Delta_{t,\max}^m] - \mathbb{E}[\xi_{t',t,\max}^m]$, we make use of the relation in (44a), which we recall

$$\mathbb{E}[\Delta_{t,\max}^m] \geq \varphi_{t',t}\mathbb{E}[\overline{\Delta}_{t',\max}] + \left[ \sum_{k=t'+1}^{t} \widetilde{\eta}_k^{(t)} \gamma p \mathbb{E}[\Delta_{k-1,\max}^m] \right] + \mathbb{E}[\xi_{t',t,\max}^m] - \varphi_{t',t}\mathbb{E}[|\overline{\Delta}_{t'}(1) - \overline{\Delta}_{t'}(2)|].$$

- To aid the analysis, we consider the following recursive relation for any fixed agent $m$:

$$y_t = (1 - \eta_t)y_{t-1} + \eta_t(\gamma p y_{t-1} + \mathbb{E}[\xi_{t',t,\max}^m]). \tag{96}$$

Upon unrolling the recursion, we obtain,

$$y_t = \left(\prod_{k=t'+1}^{t}(1 - \eta_k)\right)y_{t'} + \sum_{k=t'+1}^{t}\left(\eta_k \prod_{i=k+1}^{t}(1 - \eta_i)\right)\gamma p y_{k-1}$$

$$+ \sum_{k=t'+1}^{t}\left(\eta_k \prod_{i=k+1}^{t}(1 - \eta_i)\right)\mathbb{E}[\xi_{t',t,\max}^m]$$

$$= \varphi_{t',t}y_{t'} + \sum_{k=t'+1}^{t}\widetilde{\eta}_k^{(t)}\gamma p y_{k-1} + \sum_{k=t'+1}^{t}\widetilde{\eta}_k^{(t)}\mathbb{E}[\xi_{t',t,\max}^m]. \tag{97}$$

Initializing $y_{t'} = \mathbb{E}[\overline{\Delta}_{t',\max}]$ in (97) and plugging this into (44a), we have

$$\mathbb{E}[\Delta_{t,\max}^m] \geq y_t - \varphi_{t',t}\mathbb{E}[|\overline{\Delta}_{t'}(1) - \overline{\Delta}_{t'}(2)|],$$

where we used $\sum_{k=t'+1}^{t}\widetilde{\eta}_k^{(t)} \leq 1$ (cf. (18)). We now further simply the expression of $y_t$. By rewriting (96) as

$$y_t = (1 - \eta_t(1 - \gamma p))y_{t-1} + \eta_t\mathbb{E}[\xi_{t',t,\max}^m],$$

it is straight forward to note that $y_t$ is given as

$$y_t = \left(\prod_{k=t'+1}^{t}(1 - \eta_k(1 - \gamma p))\right)y_{t'} + \mathbb{E}[\xi_{t',t,\max}^m]\left[\sum_{k=t'+1}^{t}\eta_k^{(t)}\right]. \tag{98}$$

Consequently, we have,

$$\mathbb{E}[\Delta_{t,\max}^m] - \mathbb{E}[\xi_{t',t,\max}^m] \geq \left(\prod_{k=t'+1}^{t}(1 - \eta_k(1 - \gamma p))\right)\mathbb{E}[\overline{\Delta}_{t',\max}]$$

$$+ \mathbb{E}[\xi_{t',t,\max}^m]\left[\sum_{k=t'+1}^{t}\eta_k^{(t)} - 1\right] - \varphi_{t',t}\mathbb{E}[|\overline{\Delta}_{t'}(1) - \overline{\Delta}_{t'}(2)|]. \tag{99}$$

- We can consider a slightly different recursive sequence defined as

$$w_t = (1 - \eta_t)w_{t-1} + \eta_t(\gamma p w_{t-1}). \tag{100}$$

Using a similar sequence of arguments as outlined in (96)-(98), we can conclude that if $w_{t'} = \mathbb{E}[\overline{\Delta}_{t',\max}]$, then $\mathbb{E}[\Delta_{t,\max}^m] \geq w_t + \mathbb{E}[\xi_{t',t,\max}^m] - \varphi_{t',t}\mathbb{E}[|\overline{\Delta}_{t'}(1) - \overline{\Delta}_{t'}(2)|]$ and consequently,

$$\mathbb{E}[\Delta_{t,\max}^m] \geq \left(\prod_{k=t'+1}^{t}(1 - \eta_k(1 - \gamma p))\right)\mathbb{E}[\overline{\Delta}_{t',\max}] + \mathbb{E}[\xi_{t',t,\max}^m] - \varphi_{t',t}\mathbb{E}[|\overline{\Delta}_{t'}(1) - \overline{\Delta}_{t'}(2)|]. \tag{101}$$

On combining (99) and (101), we arrive at the claim.

### B.5.3 Proof of Lemma 4

We begin with bounding the first term $\mathbb{E}[\xi_{t',t,\max}^m]$; the second bound follows in an almost identical derivation.

**Step 1: applying Freedman's inequality.** Using the relation $\max\{a,b\} = \frac{a+b+|a-b|}{2}$, we can rewrite $\mathbb{E}[\xi_{t',t,\max}^m]$ as

$$
\begin{aligned}
\mathbb{E}[\xi_{t',t,\max}^m] &= \mathbb{E}\left[\frac{\xi_{t',t}^m(1) + \xi_{t',t}^m(2)}{2} + \left|\frac{\xi_{t',t}^m(1) - \xi_{t',t}^m(2)}{2}\right|\right] \\
&= \frac{1}{2}\mathbb{E}\left[\left|\frac{\xi_{t',t}^m(1) - \xi_{t',t}^m(2)}{2}\right|\right] \\
&= \frac{1}{2}\mathbb{E}\left[\left|\underbrace{\sum_{k=t'+1}^{t} \widetilde{\eta}_k^{(t)}\gamma(\widehat{P}_k^m(1|1,1) - \widehat{P}_k^m(1|1,2))\widehat{V}_{k-1}^m}_{=:\zeta_{t',t}^m}\right|\right],
\end{aligned}
\tag{102}
$$

where we used the definition in (38) and the fact that $\mathbb{E}[\xi_{t',t}^m(1)] = \mathbb{E}[\xi_{t',t}^m(2)] = 0$. Decompose $\zeta_{t',t}^m$ as

$$
\zeta_{t',t}^m = \sum_{k=t'+1}^{t}\sum_{b=1}^{B} \widetilde{\eta}_k^{(t)}\frac{\gamma}{B}(P_{k,b}^m(1|1,1) - P_{k,b}^m(1|1,2))\widehat{V}_{k-1}^m =: \sum_{l=1}^{L} z_l,
\tag{103}
$$

where for all $1 \leq l \leq L$

$$
z_l := \frac{\gamma}{B}(P_{k(l),b(l)}^m(1|1,1) - P_{k(l),b(l)}^m(1|1,2))\widehat{V}_{k(l)-1}^m
$$

with

$$
k(l) := \lfloor l/B \rfloor + t' + 1; \quad b(l) = ((l-1) \mod B) + 1; \quad L = (t-t')B.
$$

Let $\{\mathscr{F}_l\}_{l=1}^{L}$ be a filtration such that $\mathscr{F}_l$ is the $\sigma$-algebra corresponding to $\{P_{k(j),b(j)}^m(1|1,1), P_{k(j),b(j)}^m(1|1,2)\}_{j=1}^{l}$. It is straightforward to note that $\{z_l\}_{l=1}^{L}$ is a martingale sequence adapted to the filtration $\{\mathscr{F}_l\}_{l=1}^{L}$. We will use the Freedman's inequality [Freedman, 1975, Li et al., 2023] to obtain a high probability bound on $|\zeta_{t',t}^m|$.

- To that effect, note that

$$
\begin{aligned}
\sup_l |z_l| &\leq \sup_l \left|\widetilde{\eta}_{k(l)}^{(t)} \cdot \frac{\gamma}{B} \cdot (P_{k(l),b(l)}^m(1|1,1) - P_{k(l),b(l)}^m(1|1,2)) \cdot \widehat{V}_{k(l)-1}^m\right| \\
&\leq \widetilde{\eta}_{k(l)}^{(t)} \cdot \frac{\gamma}{B(1-\gamma)} \\
&\leq \frac{\eta_t}{B(1-\gamma)},
\end{aligned}
\tag{104}
$$

where the second step follows from the bounds $|(P_{k(l),b(l)}^m(1|1,1) - P_{k(l),b(l)}^m(1|1,2))| \leq 1$ and $\widehat{V}_{k(l)-1}^m \leq \frac{1}{1-\gamma}$ and the third step uses $c_\eta \leq \frac{1}{1-\gamma}$ and the fact that $\widetilde{\eta}_k^{(T)}$ is increasing in $k$ in this regime. (cf. (19)).

- Similarly,

$$
\begin{aligned}
\mathrm{Var}(z_l|\mathscr{F}_{l-1}) &\leq \left(\widetilde{\eta}_{k(l)}^{(t)}\right)^2\frac{\gamma^2}{B^2} \cdot \left(\widehat{V}_{k(l)-1}^m\right)^2 \cdot \mathrm{Var}(P_{k(l),b(l)}^m(1|1,1) - P_{k(l),b(l)}^m(1|1,2)) \\
&\leq \left(\widetilde{\eta}_{k(l)}^{(t)}\right)^2\frac{\gamma^2}{B^2(1-\gamma)^2} \cdot 2p(1-p) \\
&\leq \frac{2\left(\widetilde{\eta}_{k(l)}^{(t)}\right)^2}{3B^2(1-\gamma)}.
\end{aligned}
\tag{105}
$$

Using the above bounds (104) and (105) along with Freedman's inequality yield that

$$
\Pr\left(|\zeta_{t',t}^m| \geq \sqrt{\frac{8\log(2/\delta)}{3B^2(1-\gamma)}\sum_{l=1}^{L}\left(\widetilde{\eta}_{k(l)}^{(t)}\right)^2} + \frac{4\eta_t\log(2/\delta)}{3B(1-\gamma)}\right) \leq \delta.
\tag{106}
$$

Setting $\delta_0 = \frac{(1-\gamma)^2}{2} \cdot \mathbb{E}[|\zeta_{t',t}^m|^2]$, with probability at least $1 - \delta_0$, it holds

$$|\zeta_{t',t}^m| \geq \sqrt{\frac{8 \log(2/\delta_0)}{3B(1-\gamma)} \sum_{k=t'+1}^{t} \left(\widetilde{\eta}_k^{(t)}\right)^2} + \frac{4\eta_t \log(2/\delta_0)}{3B(1-\gamma)} =: D. \tag{107}$$

Consequently, plugging this back to (102), we obtain

$$\begin{aligned}
\mathbb{E}[\xi_{t',t,\max}^m] &= \frac{1}{2}\mathbb{E}[|\zeta_{t',t}^m|] \\
&\geq \frac{1}{2}\mathbb{E}[|\zeta_{t',t}^m|\mathbb{1}\{|\zeta_{t',t}^m| \leq D\}] \\
&\geq \frac{1}{2D}\mathbb{E}[|\zeta_{t',t}^m|^2\mathbb{1}\{|\zeta_{t',t}^m| \leq D\}] \\
&\geq \frac{1}{2D}\left(\mathbb{E}[|\zeta_{t',t}^m|^2] - \mathbb{E}[|\zeta_{t',t}^m|^2\mathbb{1}\{|\zeta_{t',t}^m| > D\}]\right) \\
&\geq \frac{1}{2D}\left(\mathbb{E}[|\zeta_{t',t}^m|^2] - \frac{\Pr(|\zeta_{t',t}^m| > D)}{(1-\gamma)^2}\right) \geq \frac{1}{4D} \cdot \mathbb{E}[|\zeta_{t',t}^m|^2]. \tag{108}
\end{aligned}$$

Here, the penultimate step used the fact that $|\zeta_{t',t}^m| \leq \sum_{k=t'+1}^{t} \frac{\widetilde{\eta}_k^{(t)}}{(1-\gamma)} \leq \frac{1}{(1-\gamma)}$, and the last step used the definition of $\delta_0$. Thus, it is sufficient to obtain a lower bound on $\mathbb{E}[|\zeta_{t',t}^m|^2]$ in order obtain a lower bound for $\mathbb{E}[\xi_{t',t,\max}^m]$.

**Step 2: lower bounding $\mathbb{E}[|\zeta_{t',t}^m|^2]$.** To proceed, we introduce the following lemma pertaining to lower bounding $\widehat{V}_t^m$ that will be useful later.

**Lemma 6.** *For all time instants $t \in [T]$ and all agent $m \in [M]$:*

$$\mathbb{E}\left[\left(\widehat{V}_t^m\right)^2\right] \geq \frac{1}{2(1-\gamma)^2}.$$

We have,

$$\begin{aligned}
\mathbb{E}[|\zeta_{t',t}^m|^2] &= \mathbb{E}\left[\sum_{l=1}^{L}\operatorname{Var}(z_l|\mathscr{F}_{l-1})\right] = \mathbb{E}\left[\sum_{l=1}^{L}\mathbb{E}[z_l^2|\mathscr{F}_{l-1}]\right] \\
&\geq \sum_{l=1}^{L}\left(\widetilde{\eta}_{k(l)}^{(t)}\right)^2 \frac{\gamma^2}{B^2} \cdot 2p(1-p) \cdot \mathbb{E}\left[\left(\widehat{V}_{k(l)-1}^m\right)^2\right] \\
&\geq \sum_{l=1}^{L}\left(\widetilde{\eta}_{k(l)}^{(t)}\right)^2 \frac{\gamma^2}{B^2} \cdot 2p(1-p) \cdot \frac{1}{2(1-\gamma)^2} \\
&\geq \frac{2}{9B(1-\gamma)} \cdot \sum_{k=\max\{t',\tau\}+1}^{t}\left(\widetilde{\eta}_k^{(t)}\right)^2, \tag{109}
\end{aligned}$$

where the third line follows from Lemma 6 and the fourth line uses $\gamma \geq 5/6$.

**Step 3: finishing up.** We finish up the proof by bounding $\sum_{k=\max\{t',\tau\}+1}^{t}\left(\widetilde{\eta}_k^{(t)}\right)^2$ for $t - \max\{t',\tau\} \geq 1/\eta_\tau$. We have

$$\sum_{k=\max\{t',\tau\}+1}^{t}\left(\widetilde{\eta}_k^{(t)}\right)^2 \geq \sum_{k=\max\{t',\tau\}+1}^{t}\left(\eta_k \prod_{i=k+1}^{t}(1-\eta_i)\right)^2$$

$$\overset{(i)}{\geq} \sum_{k=\max\{t',\tau\}+1}^{t} \left( \eta_t \prod_{i=k+1}^{t} (1 - \eta_\tau) \right)^2$$

$$= \eta_t^2 \sum_{k=\max\{t',\tau\}+1}^{t} (1 - \eta_\tau)^{2(t-k)}$$

$$\geq \eta_t^2 \cdot \frac{1 - (1 - \eta_\tau)^{2(t-\max\{t',\tau\})}}{\eta_\tau (2 - \eta_\tau)}$$

$$\geq \eta_t \cdot \frac{1 - \exp(-2)}{6} \geq \frac{\eta_t}{10} \geq \frac{\eta_T}{10}, \tag{110}$$

where (i) follows from the monotonicity of $\eta_k$. Plugging (110) into the expressions of $D$ (cf. (107)) we have

$$D = \sqrt{\frac{8 \log(2/\delta_0)}{3B(1-\gamma)} \sum_{k=t'+1}^{t} \left( \widetilde{\eta}_k^{(t)} \right)^2} + \frac{4\eta_t \log(2/\delta_0)}{3B(1-\gamma)}$$

$$\leq \frac{9}{2} \mathbb{E}[|\zeta_{t',t}^m|^2] \cdot \sqrt{\frac{8 \log(2/\delta_0)}{3}} \left( \frac{1}{B(1-\gamma)} \sum_{k=t'+1}^{t} \left( \widetilde{\eta}_k^{(t)} \right)^2 \right)^{-1/2} + 60 \cdot \mathbb{E}[|\zeta_{t',t}^m|^2] \cdot \log(2/\delta_0)$$

$$\leq 3\mathbb{E}[|\zeta_{t',t}^m|^2] \cdot \log(2/\delta_0) \left[ \sqrt{\frac{60B(1-\gamma)}{\eta_t}} + 20 \right]$$

$$\leq 60\mathbb{E}[|\zeta_{t',t}^m|^2] \cdot \log(2/\delta_0) \left[ \sqrt{\frac{3B(1-\gamma)}{20\eta_T}} + 1 \right],$$

where the second line follows from (109) and (110), and the third line follows from (110). On combining the above bound with (108), we obtain,

$$\mathbb{E}[\xi_{t',t,\max}^m] \geq \frac{1}{240 \log(2/\delta_0)} \cdot \frac{\nu}{\nu + 1}, \tag{111}$$

where $\nu := \sqrt{\frac{20\eta_T}{3B(1-\gamma)}}$. Note that we have,

$$\delta_0 = \frac{(1-\gamma)^2}{2} \cdot \mathbb{E}[|\zeta_{t',t}^m|^2] \geq \frac{(1-\gamma)}{9B} \cdot \sum_{k=t'+1}^{t} \left( \widetilde{\eta}_k^{(t)} \right)^2 \geq \frac{\eta_T(1-\gamma)}{90B}.$$

Combining the above bound with (111) yields us the required bound.

**Step 4: repeating the argument for the second claim.** We note that second claim in the theorem, i.e., the lower bound on $\mathbb{E}\left[ \max\left\{ \frac{1}{M} \sum_{m=1}^{M} \xi_{t',t}^m(1), \frac{1}{M} \sum_{m=1}^{M} \xi_{t',t}^m(2) \right\} \right]$ follows through an identical series of arguments where the bounds in Eqns. (104) and (105) contain an additional factor of $M$ in the denominator (effectively replacing $B$ with $BM$), which is carried through in all the following steps.

### B.5.4 Proof of Lemma 5

Using Eqns. (41) and (38), we can write

$$\overline{\Delta}_t(1) - \overline{\Delta}_t(2) = \left( \prod_{k=t'+1}^{t} (1 - \eta_k) \right) (\overline{\Delta}_{t'}(1) - \overline{\Delta}_{t'}(2))$$

$$+ \frac{1}{M} \sum_{m=1}^{M} \sum_{k=t'+1}^{t} \left( \eta_k \prod_{i=k+1}^{t} (1 - \eta_i) \right) \gamma (\widehat{P}_k^m(1|1,1) - \widehat{P}_k^m(1|1,2)) \widehat{V}_{k-1}^m.$$

Upon unrolling the recursion, we obtain,

$$\overline{\Delta}_t(1) - \overline{\Delta}_t(2) = \sum_{k=1}^{t} \sum_{m=1}^{M} \left( \eta_k \prod_{i=k+1}^{t} (1 - \eta_i) \right) \frac{\gamma}{M} (\widehat{P}_k^m(1|1,1) - \widehat{P}_k^m(1|1,2)) \widehat{V}_{k-1}^m.$$

If we define a filtration $\mathscr{F}_k$ as the $\sigma$-algebra corresponding to $\{\widehat{P}_l^1(1|1,1), \widehat{P}_l^1(1|1,2), \ldots, \widehat{P}_l^M(1|1,1), \widehat{P}_l^M(1|1,2)\}_{l=1}^{k}$, then it is straightforward to note that $\{\overline{\Delta}_t(1) - \overline{\Delta}_t(2)\}_t$ is a martingale sequence adapted to the filtration $\{\mathscr{F}_t\}_t$. Using Jensen's inequality, we know that if $\{Z_t\}_t$ is a martingale adapted to a filtration $\{\mathscr{G}_t\}_t$, then for a convex function $f$ such that $f(Z_t)$ is integrable for all $t$, $\{f(Z_t)\}_t$ is a sub-martingale adapted to $\{\mathscr{G}_t\}_t$. Since $f(x) = |x|$ is a convex function, $\{|\overline{\Delta}_t(1) - \overline{\Delta}_t(2)|\}_t$ is a submartingale adapted to the filtration $\{\mathscr{F}_t\}_t$. As a result,

$$\sup_{1 \le t \le T} \mathbb{E}[|\overline{\Delta}_t(1) - \overline{\Delta}_t(2)|] \le \mathbb{E}[|\overline{\Delta}_T(1) - \overline{\Delta}_T(2)|] \le \left( \mathbb{E}[(\overline{\Delta}_T(1) - \overline{\Delta}_T(2))^2] \right)^{1/2}. \tag{112}$$

We use the following observation about a martingale sequence $\{X_i\}_{i=1}^{t}$ adapted to a filtration $\{\mathscr{G}_i\}_{i=1}^{t}$ to evaluate the above expression. We have,

$$\mathbb{E}\left[ \left( \sum_{i=1}^{t} X_i \right)^2 \right] = \mathbb{E}\left[ \mathbb{E}\left[ \left( \sum_{i=1}^{t} X_i \right)^2 \Big| \mathscr{G}_{t-1} \right] \right]$$

$$= \mathbb{E}\left[ \mathbb{E}\left[ X_t^2 + 2X_t \left( \sum_{i=1}^{t-1} X_i \right) + \left( \sum_{i=1}^{t-1} X_i \right)^2 \Big| \mathscr{G}_{t-1} \right] \right]$$

$$= \mathbb{E}[X_t^2] + \mathbb{E}\left[ \left( \sum_{i=1}^{t-1} X_i \right)^2 \right]$$

$$= \sum_{i=1}^{t} \mathbb{E}[X_i^2], \tag{113}$$

where the third step uses the facts that $\left( \sum_{i=1}^{t-1} X_i \right)$ is $\mathscr{G}_{t-1}$ measure and $\mathbb{E}[X_t | \mathscr{G}_{t-1}] = 0$ and fourth step is obtained by recursively applying second and third steps. Using the relation in Eqn. (113) in Eqn. (112), we obtain,

$$\sup_{1 \le t \le T} \mathbb{E}[|\overline{\Delta}_t(1) - \overline{\Delta}_t(2)|] \le \left( \mathbb{E}[(\overline{\Delta}_T(1) - \overline{\Delta}_T(2))^2] \right)^{1/2}$$

$$\le \left( \sum_{k=1}^{T} \mathbb{E}\left[ \left( \sum_{m=1}^{M} \widetilde{\eta}_k^{(T)} \cdot \frac{\gamma}{M} \cdot (\widehat{P}_k^m(1|1,1) - \widehat{P}_k^m(1|1,2)) \widehat{V}_{k-1}^m \right)^2 \right] \right)^{1/2}$$

$$\le \left( \sum_{k=1}^{T} \left( \widetilde{\eta}_k^{(T)} \right)^2 \cdot \frac{2\gamma^2 p(1-p)}{BM^2} \cdot \sum_{m=1}^{M} \mathbb{E}\left[ \left( \widehat{V}_{k-1}^m \right)^2 \right] \right)^{1/2}$$

$$\le \left( \sum_{k=1}^{T} \left( \widetilde{\eta}_k^{(T)} \right)^2 \cdot \frac{2\gamma^2 p(1-p)}{BM(1-\gamma)^2} \right)^{1/2}. \tag{114}$$

Let us focus on the term involving the step sizes. We separately consider the scenario for constant step sizes and linearly rescaled step sizes. For constant step sizes, we have,

$$\sum_{k=1}^{T} \left( \widetilde{\eta}_k^{(T)} \right)^2 = \sum_{k=1}^{T} \left( \eta_k \prod_{i=k+1}^{T} (1 - \eta_i) \right)^2 = \sum_{k=1}^{T} \eta^2 (1 - \eta)^{2(T-k)} \le \frac{\eta^2}{1 - (1 - \eta)^2} \le \eta. \tag{115}$$

Similarly, for linearly rescaled step sizes, we have,

$$\sum_{k=1}^{T} \left( \widetilde{\eta}_k^{(T)} \right)^2 = \sum_{k=1}^{\tau} \left( \widetilde{\eta}_k^{(T)} \right)^2 + \sum_{k=\tau+1}^{T} \left( \eta_k \prod_{i=k+1}^{T} (1 - \eta_i) \right)^2$$

$$\leq \sum_{k=1}^{\tau} \left( \widetilde{\eta}_\tau^{(T)} \right)^2 + \sum_{k=\tau+1}^{T} \eta_k^2 (1 - \eta_T)^{2(T-k)}$$

$$\leq \eta_\tau^2 (1 - \eta_T)^{2(T-\tau)} \cdot \tau + \eta_\tau^2 \cdot \frac{1}{\eta_T (2 - \eta_T)}$$

$$\leq 3\eta_T \cdot \eta_T \cdot T \cdot \exp\left( -\frac{4T\eta_T}{3} \right) + 3\eta_T$$

$$\leq \frac{9}{4e} \eta_T + 3\eta_T$$

$$\leq 4\eta_T, \tag{116}$$

where the second step uses $c_\eta \leq \log N \leq \frac{1}{1-\gamma}$ and the fact that $\widetilde{\eta}_k^{(T)}$ is increasing in $k$ in this regime. (See Eqn. (19)) and fifth step uses $xe^{-4x/3} \leq 3/4e$. On plugging results from Eqns. (115) and (116) into Eqn. (114) along with the value of $p$, we obtain,

$$\sup_{1 \leq t \leq T} \mathbb{E}[|\overline{\Delta}_t(1) - \overline{\Delta}_t(2)|] \leq \sqrt{\frac{8\eta_T}{3BM(1-\gamma)}}, \tag{117}$$

as required.

### B.5.5 Proof of Lemma 6

For the proof, we fix an agent $m$. In order to obtain the required lower bound on $\widehat{V}_t^m$, we define an auxiliary sequence $\overline{Q}_t^m$ that evolves as described in Algorithm 5. Essentially, $\overline{Q}_t^m$ evolves in a manner almost identical to $\widehat{Q}_t^m$ except for the fact that there is only one action and hence there is no maximization step in the update rule.

---

**Algorithm 5:** Evolution of $\overline{Q}$

---

1: $r \leftarrow 1, \overline{Q}_0^m = Q^\star(1,1)$ for all $m \in \{1, 2, \ldots, M\}$
2: **for** $t = 1, 2, \ldots, T$ **do**
3:     **for** $m = 1, 2, \ldots, M$ **do**
4:         $\overline{Q}_{t-1/2}^m \leftarrow (1 - \eta_t)\overline{Q}_{t-1}^m(a) + \eta_t(1 + \widehat{P}_t^m(1|1,1)\overline{Q}_{t-1}^m)$
5:         Compute $\overline{Q}_t^m$ according to Eqn. (8)
6:     **end for**
7: **end for**

---

It is straightforward to note that $\widehat{Q}_t^m(1) \geq \overline{Q}_t^m$, which can be shown using induction. From the initialization, it follows that $\widehat{Q}_0^m(1) \geq \overline{Q}_0^m$. Assuming the relation holds for $t-1$, we have,

$$\widehat{Q}_{t-1/2}^m(1) = (1 - \eta_t)\widehat{Q}_{t-1}^m(1) + \eta_t(1 + \gamma\widehat{P}_t^m(1|1,1)\widehat{V}_{t-1}^m)$$

$$\geq (1 - \eta_t)\widehat{Q}_{t-1}^m(1) + \eta_t(1 + \gamma\widehat{P}_t^m(1|1,1)\widehat{Q}_{t-1}^m(1))$$

$$\geq (1 - \eta_t)\overline{Q}_{t-1}^m + \eta_t(1 + \gamma\widehat{P}_t^m(1|1,1)\overline{Q}_{t-1}^m)$$

$$= \overline{Q}_{t-1/2}^m.$$

Since $\widehat{Q}_t^m$ and $\overline{Q}_t^m$ follow the same averaging schedule, it immediately follows from the above relation that $\widehat{Q}_t^m(1) \geq \overline{Q}_t^m$. Since $\widehat{V}_t^m \geq \widehat{Q}_t^m(1) \geq \overline{Q}_t^m$, we will use the sequence $\overline{Q}_t^m$ to establish the required lower bound on $\widehat{V}_t^m$.

We claim that for all time instants $t$ and all agents $m$,

$$\mathbb{E}[\overline{Q}_t^m] = \frac{1}{1 - \gamma p}. \tag{118}$$

Assuming (118) holds, we have

$$\mathbb{E}[(\widehat{V}_t^m)^2] \geq \left( \mathbb{E}[\widehat{V}_t^m] \right)^2 \geq \left( \mathbb{E}[\overline{Q}_t^m] \right)^2 \geq \left( \frac{1}{1 - \gamma p} \right)^2 \geq \frac{1}{2(1 - \gamma)^2},$$

as required. In the above expression, the first inequality follows from Jensen's inequality, the second from the relation $\widehat{V}_t^m \geq \overline{Q}_t^m \geq 0$ and the third from (118).

We now move now to prove the claim (118) using induction. For the base case, $\mathbb{E}[\overline{Q}_0^m] = \frac{1}{1-\gamma p}$ holds by choice of initialization. Assume that $\mathbb{E}[\overline{Q}_{t-1}^m] = \frac{1}{1-\gamma p}$ holds for some $t-1$ for all $m$.

- If $t$ is not an averaging instant, then for any client $m$,

$$\overline{Q}_t^m = (1 - \eta_t)\overline{Q}_{t-1}^m + \eta_t(1 + \gamma \widehat{P}_t^m(1|1,1)\overline{Q}_{t-1}^m)$$

$$\implies \mathbb{E}[\overline{Q}_t^m] = (1 - \eta_t)\mathbb{E}[\overline{Q}_{t-1}^m] + \eta_t(1 + \gamma\mathbb{E}[\widehat{P}_t^m(1|1,1)\overline{Q}_{t-1}^m])$$

$$= (1 - \eta_t)\mathbb{E}[\overline{Q}_{t-1}^m] + \eta_t(1 + \gamma p\mathbb{E}[\overline{Q}_{t-1}^m])$$

$$= \frac{(1 - \eta_t)}{1 - \gamma p} + \eta_t\left(1 + \frac{\gamma p}{1 - \gamma p}\right) = \frac{1}{1 - \gamma p}. \tag{119}$$

The third line follows from the independence of $\widehat{P}_t^m(1|1,1)$ and $\overline{Q}_{t-1}^m$ and the fourth line uses the inductive hypothesis.

- If $t$ is an averaging instant, then for all clients $m$,

$$\overline{Q}_t^m = \frac{(1 - \eta_t)}{M} \sum_{j=1}^M \overline{Q}_{t-1}^j + \eta_t\frac{1}{M}\sum_{j=1}^M(1 + \gamma \widehat{P}_t^j(1|1,1)\overline{Q}_{t-1}^j)$$

$$\implies \mathbb{E}[\overline{Q}_t^m] = \frac{(1 - \eta_t)}{M} \sum_{j=1}^M \mathbb{E}[\overline{Q}_{t-1}^j] + \eta_t\frac{1}{M}\sum_{j=1}^M(1 + \gamma\mathbb{E}[\widehat{P}_t^j(1|1,1)\overline{Q}_{t-1}^j])$$

$$= \frac{(1 - \eta_t)}{M} \sum_{j=1}^M \frac{1}{1 - \gamma p} + \eta_t\frac{1}{M}\sum_{j=1}^M\left(1 + \frac{\gamma p}{1 - \gamma p}\right) = \frac{1}{1 - \gamma p}, \tag{120}$$

where we again make use of independence and the inductive hypothesis.

Thus, (119) and (120) taken together complete the inductive step.

## C    Analysis of Fed-DVR-Q

In this section, we prove Theorem 2 that outlines the performance guarantees of Fed-DVR-Q. There are two main parts of the proof. The first part deals with establishing that for the given choice of parameters described in Section 4.1.3, the output of the algorithm is an $\varepsilon$-optimal estimate of $Q^\star$ with probability $1 - \delta$. The second part deals with deriving the bounds on the sample and communication complexity based on the choice of prescribed parameters. We begin with the second part, which is easier of the two.

### C.1    Establishing the sample and communication complexity bounds

**Establishing the communication complexity.**    We begin with bounding $\mathsf{CC}_{\mathsf{round}}$. From the description of Fed-DVR-Q, it is straightforward to note that each epoch, i.e., each call to the REFINEES-TIMATE routine, involves $I + 1$ rounds of communication, one for estimating $\mathcal{T}\overline{Q}$ and the remaining ones during the iterative updates of the Q-function. Since there are a total of $K$ epochs,

$$\mathsf{CC}_{\mathsf{round}}(\mathsf{Fed\text{-}DVR\text{-}Q}; \varepsilon, M, \delta) \leq (I + 1)K \leq \frac{16}{\eta(1 - \gamma)} \log_2\left(\frac{1}{(1 - \gamma)\varepsilon}\right),$$

where the second bound follows from the prescribed choice of parameters in Sec. 4.1.3. Similarly, since the quantization step is designed to compress each coordinate into $J$ bits, each message transmitted by an agent has a size of no more than $J \cdot |\mathcal{S}||\mathcal{A}|$ bits. Consequently,

$$\mathsf{CC}_{\mathsf{bit}}(\mathsf{Fed\text{-}DVR\text{-}Q}; \varepsilon, M, \delta) \leq J \cdot |\mathcal{S}||\mathcal{A}| \cdot \mathsf{CC}_{\mathsf{round}}(\mathsf{Fed\text{-}DVR\text{-}Q}; \varepsilon, M, \delta)$$

$$\leq \frac{32|\mathcal{S}||\mathcal{A}|}{\eta(1 - \gamma)} \log_2\left(\frac{1}{(1 - \gamma)\varepsilon}\right)\log_2\left(\frac{70}{\eta(1 - \gamma)}\sqrt{\frac{4}{M}\log\left(\frac{8KI|\mathcal{S}||\mathcal{A}|}{\delta}\right)}\right),$$

where once again in the second step we plugged in the choice of $J$ from Sec. 4.1.3.

**Establishing the sample complexity.** In order to establish the bound on the sample complexity, note that during epoch $k$, each agent takes a total of $\lceil L_k/M \rceil + I \cdot B$ samples, where the first term corresponds to approximating $\widetilde{\mathcal{T}}_L(Q^{(k-1)})$ and the second term corresponds to the samples taken during the iterative update scheme. Thus, the total sample complexity is obtained by summing up over all the $K$ epochs. We have,

$$\mathsf{SC}(\mathsf{Fed\text{-}DVR\text{-}Q}; \varepsilon, M, \delta) \leq \sum_{k=1}^{K} \left( \left\lceil \frac{L_k}{M} \right\rceil + I \cdot B \right) \leq I \cdot B \cdot K + \frac{1}{M} \sum_{k=1}^{K} L_k + K.$$

To continue, notice that

$$\frac{1}{M} \sum_{k=1}^{K} L_k \leq \frac{39200}{M(1-\gamma)^2} \log\left( \frac{8KI|\mathcal{S}||\mathcal{A}|}{\delta} \right) \left( \sum_{k=1}^{K_0} 4^k + \sum_{k=K_0+1}^{K} 4^{k-K_0} \right)$$

$$\leq \frac{39200}{3M(1-\gamma)^2} \log\left( \frac{8KI|\mathcal{S}||\mathcal{A}|}{\delta} \right) \left( 4^{K_0} + 4^{K-K_0} \right)$$

$$\leq \frac{156800}{3M(1-\gamma)^2} \log\left( \frac{8KI|\mathcal{S}||\mathcal{A}|}{\delta} \right) \left( \frac{1}{1-\gamma} + \frac{1}{(1-\gamma)\varepsilon^2} \right),$$

where the first line follows from the choice of $L_k$ in Sec. 4.1.3 and the last line follows from $K_0 = \lceil \frac{1}{2} \log_2(\frac{1}{1-\gamma}) \rceil$. Plugging this relation and the choices of $I$ and $B$ (cf. Sec. 4.1.3) into the previous bound yields

$$\mathsf{SC}(\mathsf{Fed\text{-}DVR\text{-}Q}; \varepsilon, M, \delta) \leq \frac{4608}{\eta M(1-\gamma)^3} \log_2\left( \frac{1}{(1-\gamma)\varepsilon} \right) \log\left( \frac{8KI|\mathcal{S}||\mathcal{A}|}{\delta} \right) + K$$

$$+ \frac{156800}{3M(1-\gamma)^2} \log\left( \frac{8KI|\mathcal{S}||\mathcal{A}|}{\delta} \right) \left( \frac{1}{1-\gamma} + \frac{1}{(1-\gamma)\varepsilon^2} \right)$$

$$\leq \frac{313600}{\eta M(1-\gamma)^3 \varepsilon^2} \log_2\left( \frac{1}{(1-\gamma)\varepsilon} \right) \log\left( \frac{8KI|\mathcal{S}||\mathcal{A}|}{\delta} \right) + K.$$

Plugging in the choice of $K$ finishes the proof.

## C.2 Establishing the error guarantees

In this section, we show that the Q-function estimate returned by the $\mathsf{Fed\text{-}DVR\text{-}Q}$ algorithm is $\varepsilon$-optimal with probability at least $1 - \delta$. We claim that the estimates of the Q-function generated by the algorithm across different epochs satisfy the following relation for all $k \leq K$ with probability $1 - \delta$:

$$\|Q^{(k)} - Q^\star\|_\infty \leq \frac{2^{-k}}{1-\gamma}. \tag{121}$$

The required bound on $\|Q^{(K)} - Q^\star\|_\infty$ immediately follows by plugging in the value of $K$. Thus, for the remainder of the section, we focus on establishing the above claim.

**Step 1: fixed-point contraction of** REFINEESTIMATE. Firstly, note that the variance-reduced update scheme carried out during the REFINEESTIMATE routine resembles that of the classic Q-learning scheme, i.e., fixed-point iteration, with a different operator defined as follows:

$$\mathcal{H}(Q) := \mathcal{T}(Q) - \mathcal{T}(\overline{Q}) + \widetilde{\mathcal{T}}_L(\overline{Q}), \quad \text{for some fixed } \overline{Q}. \tag{122}$$

Thus, the update scheme at step $i \geq 1$ in (11) can then be written as

$$Q_{i-\frac{1}{2}}^m = (1-\eta)Q_{i-1} + \eta \widehat{\mathcal{H}}_i^{(m)}(Q_{i-1}), \tag{123}$$

where $\widehat{\mathcal{H}}_i^{(m)}(Q) := \widehat{\mathcal{T}}_i^{(m)}(Q) - \widehat{\mathcal{T}}_i^{(m)}(\overline{Q}) + \widetilde{\mathcal{T}}_L(\overline{Q})$ is a stochastic, unbiased estimate of the operator $\mathcal{H}$, similar to $\widehat{\mathcal{T}}_i^{(m)}(Q)$. Let $Q_{\mathcal{H}}^\star$ denote the fixed point of $\mathcal{H}$. Then the update scheme in (123) drives the sequence $\{Q_i^m\}_{i\geq 0}$ to $Q_{\mathcal{H}}^\star$; further, as long as $\|Q^\star - Q_{\mathcal{H}}^\star\|_\infty$ is small, the required error $\|Q_i - Q^\star\|_\infty$ can also be controlled. The following lemmas formalize these ideas and pave the path to establish the claim in (121). The proofs are deferred to Appendix C.3.

**Lemma 7.** *Let $\delta \in (0, 1)$. Consider the* REFINEESTIMATE *routine described in Algorithm 3 and let $Q_{\mathcal{H}}^{\star}$ denote the fixed point of the operator $\mathcal{H}$ defined in (122) for some fixed $\overline{Q}$. Then the iterates generated by* REFINEESTIMATE $Q_I$ *satisfy*

$$\|Q_I - Q_{\mathcal{H}}^{\star}\|_{\infty} \leq \frac{1}{6} \left( \|\overline{Q} - Q^{\star}\|_{\infty} + \|Q^{\star} - Q_{\mathcal{H}}^{\star}\|_{\infty} \right) + \frac{D}{70}$$

*with probability $1 - \frac{\delta}{2K}$.*

**Lemma 8.** *Consider the* REFINEESTIMATE *routine described in Alg. 3 and let $Q_{\mathcal{H}}^{\star}$ denote the fixed point of the operator $\mathcal{H}$ defined in Eqn. (122) for a fixed $\overline{Q}$. The following relation holds with probability $1 - \frac{\delta}{2K}$:*

$$\|Q_{\mathcal{H}}^{\star} - Q^{\star}\|_{\infty} \leq \|\overline{Q} - Q^{\star}\|_{\infty} \cdot \sqrt{\frac{16\kappa'}{L(1-\gamma)^2}} + \sqrt{\frac{64\kappa'}{L(1-\gamma)^3}} + \frac{2\kappa'\sqrt{2}}{3L(1-\gamma)^2} + \frac{D}{70},$$

*whenever $L \geq 32\kappa'$, where $\kappa' = \log\left( \frac{12K|\mathcal{S}||\mathcal{A}|}{\delta} \right)$.*

**Step 2: establishing the linear contraction.** We now leverage the above lemmas to establish the desired contraction in (121). Instantiating the operator (122) at each $k$-th epoch by setting $\overline{Q} := Q^{(k-1)}$ and $L := L_k$, we define

$$\mathcal{H}_k(Q) := \mathcal{T}(Q) - \mathcal{T}(Q^{(k-1)}) + \widetilde{\mathcal{T}}_{L_k}(Q^{(k-1)}), \tag{124}$$

whose fixed point is denoted as $Q_{\mathcal{H}_k}^{\star}$. Using the results from Lemmas 7 and 8 with $D := D_k$ and $\mathcal{H} = \mathcal{H}_k$, we obtain

$$
\begin{aligned}
\|Q^{(k)} - Q^{\star}\|_{\infty} &\leq \|Q^{(k)} - Q_{\mathcal{H}_k}^{\star}\|_{\infty} + \|Q_{\mathcal{H}}^{\star} - Q_{\mathcal{H}_k}^{\star}\|_{\infty} \\
&\leq \frac{1}{6} \left( \|Q^{(k-1)} - Q^{\star}\|_{\infty} + \|Q^{\star} - Q_{\mathcal{H}_k}^{\star}\|_{\infty} \right) + \frac{D_k}{70} + \|Q_{\mathcal{H}_k}^{\star} - Q^{\star}\|_{\infty} \\
&= \frac{1}{6} \left( \|Q^{(k-1)} - Q^{\star}\|_{\infty} + 7\|Q^{\star} - Q_{\mathcal{H}_k}^{\star}\|_{\infty} \right) + \frac{D_k}{70} \\
&\leq \|Q^{(k-1)} - Q^{\star}\|_{\infty} \left( \frac{1}{6} + \frac{7}{6} \sqrt{\frac{16\kappa'}{L_k(1-\gamma)^2}} \right) + \frac{7}{6} \left( \sqrt{\frac{64\kappa'}{L_k(1-\gamma)^3}} + \frac{2\sqrt{2}\kappa'}{3L_k(1-\gamma)^2} \right) + \frac{13D_k}{420} \\
&\leq \|Q^{(k-1)} - Q^{\star}\|_{\infty} \left( \frac{1}{6} + \frac{7}{6} \sqrt{\frac{16\kappa'}{L_k(1-\gamma)^2}} \right) + \frac{7}{6} \sqrt{\frac{100\kappa'}{L_k(1-\gamma)^3}} + \frac{13D_k}{420},
\end{aligned}
\tag{125}
$$

holds with probability $1 - \frac{\delta}{K}$. Here, we invoke Lemma 7 in the second step and Lemma 8 in the fourth step corresponding to the REFINEESTIMATE routine during the $k$-th epoch. In the last step, we used the fact that $\frac{L_k(1-\gamma)^2}{\kappa'} \geq 1$.

We now use induction along with the recursive relation in (125) to establish the required claim (121). Let us first consider the case $0 \leq k \leq K_0$. The base case, $\|Q^{(0)} - Q^{\star}\|_{\infty} \leq \frac{1}{1-\gamma}$, holds by definition. Let us assume the relation holds for $k - 1$. Then, from (125) and choice of $L_k$ (Sec. 4.1.3), we have

$$
\begin{aligned}
\|Q^{(k)} - Q^{\star}\|_{\infty} &\leq \|Q^{(k-1)} - Q^{\star}\|_{\infty} \left( \frac{1}{6} + \frac{7}{6} \sqrt{\frac{16\kappa'}{L_k(1-\gamma)^2}} \right) + \frac{7}{6} \sqrt{\frac{100\kappa'}{L_k(1-\gamma)^3}} + \frac{13D_k}{420} \\
&\leq \frac{2^{-(k-1)}}{1-\gamma} \left( \frac{1}{6} + 2^{-k} \cdot \frac{7}{6} \sqrt{\frac{8}{19600}} \right) + 2^{-k} \cdot \frac{7}{6} \sqrt{\frac{50}{19600(1-\gamma)}} + \frac{104}{420} \cdot \frac{2^{-(k-1)}}{1-\gamma} \\
&\leq \frac{2^{-(k-1)}}{1-\gamma} \left( \frac{1}{6} + \frac{7}{6} \sqrt{\frac{91}{39200}} + \frac{1}{4} \right) \\
&\leq \frac{2^{-k}}{1-\gamma}. 
\end{aligned}
\tag{126}
$$

Now we move to the second case, for $k > K_0$. From (125) and choice of $L_k$ (Sec. 4.1.3), we have

$$\|Q^{(k)} - Q^\star\|_\infty \leq \|Q^{(k-1)} - Q^\star\|_\infty \left(\frac{1}{6} + \frac{7}{6}\sqrt{\frac{16\kappa'}{L_k(1-\gamma)^2}}\right) + \frac{7}{6}\sqrt{\frac{100\kappa'}{L_k(1-\gamma)^3}} + \frac{13D_k}{420}$$

$$\leq \frac{2^{-(k-1)}}{1-\gamma}\left(\frac{1}{6} + 2^{-(k-K_0)} \cdot \frac{7}{6}\sqrt{\frac{8}{19600}}\right) + 2^{-(k-K_0)} \cdot \frac{7}{6}\sqrt{\frac{50}{19600(1-\gamma)}} + \frac{104}{420} \cdot \frac{2^{-(k-1)}}{1-\gamma}$$

$$\leq \frac{2^{-(k-1)}}{1-\gamma}\left(\frac{1}{6} + \frac{7}{6}\sqrt{\frac{1}{196}} + \frac{1}{4}\right)$$

$$\leq \frac{2^{-k}}{1-\gamma}. \tag{127}$$

By a union bound argument, we can conclude that the relation $\|Q^{(k)} - Q^\star\|_\infty \leq \frac{2^{-k}}{1-\gamma}$ holds for all $k \leq K$ with probability at least $1 - \delta$.

**Step 3: confirm the compressor bound.** The only thing left to verify is that the inputs to the compressor are always bounded by $D_k$ during the $k$-th epoch, for all $1 \leq k \leq K$. The following lemma provides a bound on the input to the compressor during any run of the REFINEESTIMATE routine.

**Lemma 9.** *Consider the* REFINEESTIMATE *routine described in Algorithm 3 with some for some fixed* $\overline{Q}$. *For all* $i \leq I$ *and all agents* $m$, *the following bound holds with probability* $1 - \frac{\delta}{2K}$:

$$\|Q^m_{i-\frac{1}{2}} - Q_{i-1}\|_\infty \leq \eta\|\overline{Q} - Q^\star_{\mathcal{H}}\|_\infty\left(\frac{7}{6} \cdot (1+\gamma) + 2\gamma\right) + \frac{\eta D(1+\gamma)}{70}.$$

For the $k$-th epoch, it follows that

$$\eta\|Q^{(k-1)} - Q^\star_{\mathcal{H}_k}\|_\infty\left(\frac{7}{6} \cdot (1+\gamma) + 2\gamma\right) + \frac{\eta D_k(1+\gamma)}{70}$$

$$\leq \frac{13}{3}\left(\|Q^{(k-1)} - Q^\star\|_\infty + \|Q^\star - Q^\star_{\mathcal{H}_k}\|_\infty\right) + \frac{D_k(1+\gamma)}{70}$$

$$\leq \frac{13}{3} \cdot \frac{15}{14} \cdot \|Q^{(k-1)} - Q^\star\|_\infty + \frac{2D_k}{70}$$

$$\leq \left(\frac{195}{42} + \frac{16}{70}\right) \cdot \frac{2^{-(k-1)}}{1-\gamma}$$

$$\leq 8 \cdot \frac{2^{-(k-1)}}{1-\gamma} := D_k.$$

In the third step, we used the same sequence of arguments as used in (126) and (127) and, in the fourth step, we used the bound on $\|Q^{(k-1)} - Q^\star\|_\infty$ from (121) and the prescribed value of $D_k$.

## C.3 Proof of auxiliary lemmas

### C.3.1 Proof of Lemma 7

Let us begin with analyzing the evolution of the sequence $\{Q_i\}_{i=1}^I$ during a run of the REFINEESTI-MATE routine. The sequence $\{Q_i\}_{i=1}^I$ satisfies the following recursion:

$$Q_i = Q_{i-1} + \frac{1}{M}\sum_{m=1}^M \mathscr{C}\left(Q^m_{i-\frac{1}{2}} - Q_{i-1}; D, J\right)$$

$$= Q_{i-1} + \frac{1}{M}\sum_{m=1}^M \left(Q^m_{i-\frac{1}{2}} - Q_{i-1} + \zeta^m_i\right)$$

$$= \frac{1}{M} \sum_{m=1}^{M} \left( Q_{i-\frac{1}{2}}^m + \zeta_i^m \right) = (1-\eta)Q_{i-1} + \frac{\eta}{M} \sum_{m=1}^{M} \widehat{\mathcal{H}}_i^{(m)}(Q_{i-1}) + \underbrace{\frac{1}{M} \sum_{m=1}^{M} \zeta_i^m}_{=:\zeta_i}. \quad (128)$$

In the above expression, $\zeta_i^m$ denotes the quantization noise introduced at agent $m$ in the $i$-th update. Subtracting $Q_{\mathcal{H}}^\star$ from both sides of (128), we obtain

$$Q_i - Q_{\mathcal{H}}^\star = (1-\eta)(Q_{i-1} - Q_{\mathcal{H}}^\star) + \frac{\eta}{M} \sum_{m=1}^{M} \left( \widehat{\mathcal{H}}_i^{(m)}(Q_{i-1}) - Q_{\mathcal{H}}^\star \right) + \zeta_i$$

$$= (1-\eta)(Q_{i-1} - Q_{\mathcal{H}}^\star) + \frac{\eta}{M} \sum_{m=1}^{M} \left( \widehat{\mathcal{H}}_i^{(m)}(Q_{i-1}) - \widehat{\mathcal{H}}_i^{(m)}(Q_{\mathcal{H}}^\star) \right)$$

$$+ \frac{\eta}{M} \sum_{m=1}^{M} \left( \widehat{\mathcal{H}}_i^{(m)}(Q_{\mathcal{H}}^\star) - \mathcal{H}(Q_{\mathcal{H}}^\star) \right) + \zeta_i. \quad (129)$$

Consequently,

$$\|Q_i - Q_{\mathcal{H}}^\star\|_\infty \leq (1-\eta)\|Q_{i-1} - Q_{\mathcal{H}}^\star\|_\infty + \frac{\eta}{M} \sum_{m=1}^{M} \left\| \widehat{\mathcal{H}}_i^{(m)}(Q_{i-1}) - \widehat{\mathcal{H}}_i^{(m)}(Q_{\mathcal{H}}^\star) \right\|_\infty$$

$$+ \left\| \frac{\eta}{M} \sum_{m=1}^{M} \left( \widehat{\mathcal{H}}_i^{(m)}(Q_{\mathcal{H}}^\star) - \mathcal{H}(Q_{\mathcal{H}}^\star) \right) \right\|_\infty + \|\zeta_i\|_\infty, \quad (130)$$

which we shall proceed to bound each term separately.

- Regarding the second term, it follows that
$$\left\| \widehat{\mathcal{H}}_i^{(m)}(Q) - \widehat{\mathcal{H}}_i^{(m)}(Q_{\mathcal{H}}^\star) \right\|_\infty = \left\| \widehat{\mathcal{T}}_i^{(m)}(Q) - \widehat{\mathcal{T}}_i^{(m)}(Q_{\mathcal{H}}^\star) \right\|_\infty \leq \gamma \|Q - Q_{\mathcal{H}}^\star\|_\infty, \quad (131)$$
which holds for all $Q$ since $\widehat{\mathcal{T}}_i^{(m)}$ is a $\gamma$-contractive operator.

- Regarding the third term, notice that
$$\frac{1}{M} \sum_{m=1}^{M} \left( \widehat{\mathcal{H}}_i^{(m)}(Q_{\mathcal{H}}^\star) - \mathcal{H}(Q_{\mathcal{H}}^\star) \right) = \frac{1}{MB} \sum_{m=1}^{M} \sum_{z \in \mathcal{Z}_i^{(m)}} \left( \mathcal{T}_z(Q_{\mathcal{H}}^\star) - \mathcal{T}_z(\overline{Q}) - \mathcal{T}(Q_{\mathcal{H}}^\star) + \mathcal{T}(\overline{Q}) \right).$$

Note that $\mathcal{T}_z(Q_{\mathcal{H}}^\star) - \mathcal{T}_z(\overline{Q}) - \mathcal{T}(Q_{\mathcal{H}}^\star) + \mathcal{T}(\overline{Q})$ is a zero-mean random vector satisfying
$$\|\mathcal{T}_z(Q_{\mathcal{H}}^\star) - \mathcal{T}_z(\overline{Q}) - \mathcal{T}(Q_{\mathcal{H}}^\star) + \mathcal{T}(\overline{Q})\|_\infty \leq 2\gamma\|\overline{Q} - Q_{\mathcal{H}}^\star\|_\infty. \quad (132)$$
Thus, each of its coordinate is a $(2\gamma\|\overline{Q} - Q_{\mathcal{H}}^\star\|_\infty)^2$-sub-Gaussian vector. Applying the tail bounds for a maximum of sub-Gaussian random variables [Vershynin, 2018], we obtain that

$$\left\| \frac{1}{M} \sum_{m=1}^{M} \left( \widehat{\mathcal{H}}_i^{(m)}(Q_{\mathcal{H}}^\star) - \mathcal{H}(Q_{\mathcal{H}}^\star) \right) \right\|_\infty \leq 2\gamma\|\overline{Q} - Q_{\mathcal{H}}^\star\|_\infty \cdot \sqrt{\frac{2}{MB} \log\left( \frac{8KI|\mathcal{S}||\mathcal{A}|}{\delta} \right)} \quad (133)$$

holds with probability at least $1 - \frac{\delta}{4KI}$.

- Turning to the last term, by the construction of the compression routine described in Section 4.1.2, it is straightforward to note that $\zeta_i^m$ is a zero-mean random vector whose coordinates are independent, $D^2 \cdot 4^{-J}$-sub-Gaussian random variables. Thus, $\zeta_i$ is also a zero-mean random vector whose coordinates are independent, $\frac{D^2}{M \cdot 4^J}$-sub-Gaussian random variables. Hence, we can similarly conclude that

$$\|\zeta_i\|_\infty \leq D \cdot 2^{-J} \cdot \sqrt{\frac{2}{M} \log\left( \frac{8KI|\mathcal{S}||\mathcal{A}|}{\delta} \right)} \quad (134)$$

holds with probability at least $1 - \frac{\delta}{4KI}$.

Combining the above bounds into (130), and introducing the short-hand notation $\kappa := \log\left(\frac{8KI|\mathcal{S}||\mathcal{A}|}{\delta}\right)$, we obtain with probability at least $1 - \frac{\delta}{2KI}$,

$$\|Q_i - Q_{\mathcal{H}}^\star\|_\infty \leq (1 - \eta(1-\gamma))\|Q_{i-1} - Q_{\mathcal{H}}^\star\|_\infty + 2\eta\gamma\|\overline{Q} - Q_{\mathcal{H}}^\star\|_\infty \cdot \sqrt{\frac{2\kappa}{MB}} + D \cdot 2^{-J} \cdot \sqrt{\frac{2\kappa}{M}}.$$

Unrolling the above recursion over $i = 1, \ldots, I$ yields the following relation, which holds with probability at least $1 - \frac{\delta}{2K}$:

$$\|Q_I - Q_{\mathcal{H}}^\star\|_\infty \leq (1 - \eta(1-\gamma))^I \|Q_0 - Q_{\mathcal{H}}^\star\|_\infty + \sqrt{\frac{2\kappa}{M}}\left(\frac{2\eta\gamma}{\sqrt{B}}\|\overline{Q} - Q_{\mathcal{H}}^\star\|_\infty + D \cdot 2^{-J}\right) \cdot \sum_{i=1}^{I}(1 - \eta(1-\gamma))^{I-i}$$

$$\leq (1 - \eta(1-\gamma))^I \|\overline{Q} - Q_{\mathcal{H}}^\star\|_\infty + \frac{1}{\eta(1-\gamma)}\sqrt{\frac{2\kappa}{M}}\left(\frac{2\eta\gamma}{\sqrt{B}}\|\overline{Q} - Q_{\mathcal{H}}^\star\|_\infty + D \cdot 2^{-J}\right)$$

$$\leq \|\overline{Q} - Q_{\mathcal{H}}^\star\|_\infty\left((1 - \eta(1-\gamma))^I + \frac{2\gamma}{(1-\gamma)}\sqrt{\frac{2\kappa}{MB}}\right) + \frac{D \cdot 2^{-J}}{\eta(1-\gamma)} \cdot \sqrt{\frac{2\kappa}{M}} \quad (135)$$

$$\leq \frac{\|\overline{Q} - Q_{\mathcal{H}}^\star\|_\infty}{6} + \frac{D}{70} \leq \frac{1}{6}\left(\|\overline{Q} - Q^\star\|_\infty + \|Q^\star - Q_{\mathcal{H}}^\star\|_\infty\right) + \frac{D}{70}. \quad (136)$$

Here, the fourth step is obtained by plugging in the prescribed values of $B, I$ and $J$ in Sec. 4.1.3.

### C.3.2 Proof of Lemma 8

Intuitively, the error $\|Q_{\mathcal{H}}^\star - Q^\star\|_\infty$ depends on the error term $\widetilde{\mathcal{T}}_L(\overline{Q}) - \mathcal{T}(\overline{Q})$. If the latter is small, then $\mathcal{H}(Q)$ is close to $\mathcal{T}(Q)$ and consequently so are $Q_{\mathcal{H}}^\star$ and $Q^\star$. Thus, we begin with bounding the term $\widetilde{\mathcal{T}}_L(\overline{Q}) - \mathcal{T}(\overline{Q})$. We have,

$$\widetilde{\mathcal{T}}_L(\overline{Q}) - \mathcal{T}(\overline{Q})$$

$$= \overline{Q} + \frac{1}{M}\sum_{m=1}^{M}\mathscr{C}\left(\widetilde{\mathcal{T}}_L^{(m)}(\overline{Q}) - \overline{Q}\right) - \mathcal{T}(\overline{Q})$$

$$= \frac{1}{M}\sum_{m=1}^{M}\left(\widetilde{\mathcal{T}}_L^{(m)}(\overline{Q}) + \tilde{\zeta}_L^{(m)}\right) - \mathcal{T}(\overline{Q})$$

$$= \frac{1}{M}\sum_{m=1}^{M}\left(\widetilde{\mathcal{T}}_L^{(m)}(\overline{Q}) - \widetilde{\mathcal{T}}_L^{(m)}(Q^\star) - \mathcal{T}(\overline{Q}) + \mathcal{T}(Q^\star)\right) + \frac{1}{M}\sum_{m=1}^{M}\tilde{\zeta}_L^{(m)} + \frac{1}{M}\sum_{m=1}^{M}\left(\widetilde{\mathcal{T}}_L^{(m)}(Q^\star) - \mathcal{T}(Q^\star)\right),$$
$$(137)$$

where once again $\tilde{\zeta}_L^{(m)} := \widetilde{\mathcal{T}}_L^{(m)}(\overline{Q}) - \overline{Q} - \mathscr{C}\left(\widetilde{\mathcal{T}}_L^{(m)}(\overline{Q}) - \overline{Q}\right)$ denotes the quantization error at agent $m$. Similar to the arguments of (133) and (134), we can conclude that each of the following relations hold with probability at least $1 - \frac{\delta}{6K}$:

$$\left\|\frac{1}{M}\sum_{m=1}^{M}\left(\widetilde{\mathcal{T}}_L^{(m)}(\overline{Q}) - \widetilde{\mathcal{T}}_L^{(m)}(Q^\star) - \mathcal{T}(\overline{Q}) + \mathcal{T}(Q^\star)\right)\right\|_\infty \leq 2\gamma\|\overline{Q} - Q^\star\|_\infty \cdot \sqrt{\frac{2}{L}\log\left(\frac{12K|\mathcal{S}||\mathcal{A}|}{\delta}\right)},$$
$$(138)$$

$$\left\|\frac{1}{M}\sum_{m=1}^{M}\tilde{\zeta}_L^{(m)}\right\|_\infty \leq D \cdot 2^{-J} \cdot \sqrt{\frac{2}{M}\log\left(\frac{12K|\mathcal{S}||\mathcal{A}|}{\delta}\right)}.$$
$$(139)$$

For the third term, we can rewrite it as

$$\frac{1}{M}\sum_{m=1}^{M}\left(\widetilde{\mathcal{T}}_L^{(m)}(Q^\star) - \mathcal{T}(Q^\star)\right) = \frac{1}{M\lceil L/M\rceil}\sum_{m=1}^{M}\sum_{l=1}^{\lceil L/M\rceil}\left(\mathcal{T}_{Z_l^{(m)}}(Q^\star) - \mathcal{T}(Q^\star)\right).$$

We will use Bernstein inequality element wise to bound the above term. Let $\boldsymbol{\sigma}^\star \in \mathbb{R}^{|\mathcal{S}|\times|\mathcal{A}|}$ be such that $[\boldsymbol{\sigma}^\star(s,a)]^2 = \mathrm{Var}(\mathcal{T}_Z(Q^\star)(s,a))$, i.e., $(s,a)$-th element of $\boldsymbol{\sigma}$ denotes the standard deviation of the random variable $\mathcal{T}_Z(Q^\star)(s,a)$. Since $\|\mathcal{T}_Z(Q^\star) - \mathcal{T}(Q^\star)\|_\infty \leq \frac{1}{1-\gamma}$ a.s., Bernstein inequality gives us that

$$\left|\frac{1}{M}\sum_{m=1}^{M}\left(\widetilde{\mathcal{T}}_L^{(m)}(Q^\star)(s,a) - \mathcal{T}(Q^\star)(s,a)\right)\right| \leq \boldsymbol{\sigma}^\star(s,a)\sqrt{\frac{2}{L}\log\left(\frac{6K|\mathcal{S}||\mathcal{A}|}{\delta}\right)} + \frac{2}{3L(1-\gamma)}\log\left(\frac{6K|\mathcal{S}||\mathcal{A}|}{\delta}\right).$$
$$(140)$$

holds simultaneously for all $(s,a) \in \mathcal{S}\times\mathcal{A}$ with probability at least $1 - \frac{\delta}{6K}$. On combining (137), (138), (139) and (140), we obtain that

$$\left|\widetilde{\mathcal{T}}_L(\overline{Q})(s,a) - \mathcal{T}(\overline{Q})(s,a)\right| = \|\overline{Q} - Q^\star\|_\infty \cdot \sqrt{\frac{8\kappa'}{L}} + \boldsymbol{\sigma}^\star(s,a)\sqrt{\frac{2\kappa'}{L}} + \frac{2\kappa'}{3L(1-\gamma)} + D\cdot 2^{-J}\cdot\sqrt{\frac{2\kappa'}{M}},$$
$$(141)$$

holds simultaneously for all $(s,a) \in \mathcal{S}\times\mathcal{A}$ with probability at least $1 - \frac{\delta}{2K}$, where $\kappa' = \log\left(\frac{12K|\mathcal{S}||\mathcal{A}|}{\delta}\right)$. We use this bound in (141) to obtain a bound on $\|Q_{\mathcal{H}}^\star - Q^\star\|_\infty$ using the following lemma.

**Lemma 10** (Wainwright [2019b]). *Let $\pi^\star$ and $\pi_{\mathcal{H}}^\star$ respectively denote the optimal policies w.r.t. $Q^\star$ and $Q_{\mathcal{H}}^\star$. Then,*

$$\|Q_{\mathcal{H}}^\star - Q^\star\|_\infty \leq \max\left\{(I - \gamma P^{\pi^\star})^{-1}\left|\widetilde{\mathcal{T}}_L(\overline{Q}) - \mathcal{T}(\overline{Q})\right|, (I - \gamma P^{\pi_{\mathcal{H}}^\star})^{-1}\left|\widetilde{\mathcal{T}}_L(\overline{Q}) - \mathcal{T}(\overline{Q})\right|\right\}.$$

*Here, for any deterministic policy $\pi$, $P^\pi \in \mathbb{R}^{|\mathcal{S}||\mathcal{A}|\times|\mathcal{S}||\mathcal{A}|}$ is given by $(P^\pi Q)(s,a) = \sum_{s'\in\mathcal{S}} P(s'|s,a)Q(s',\pi(s'))$.*

Furthermore, it was shown in Wainwright [2019b, Proof of Lemma 4] that if the error $|\widetilde{\mathcal{T}}_L(\overline{Q})(s,a) - \mathcal{T}(\overline{Q})(s,a)|$ satisfies

$$\left|\widetilde{\mathcal{T}}_L(\overline{Q})(s,a) - \mathcal{T}(\overline{Q})(s,a)\right| \leq z_0\|\overline{Q} - Q^\star\|_\infty + z_1\boldsymbol{\sigma}^\star(s,a) + z_2 \tag{142}$$

for some $z_0, z_1, z_2 \geq 0$ with $z_1 < 1$, then the bound in Lemma 10 can be simplified to

$$\|Q_{\mathcal{H}}^\star - Q^\star\|_\infty \leq \frac{1}{1-z_1}\left(\frac{z_0}{1-\gamma}\|\overline{Q} - Q^\star\|_\infty + \frac{z_1}{(1-\gamma)^{3/2}} + \frac{z_2}{1-\gamma}\right). \tag{143}$$

On comparing, (141) with (142), we obtain

$$z_0 \equiv \sqrt{\frac{8\kappa'}{L}}; \quad z_1 \equiv \sqrt{\frac{2\kappa'}{L}}; \quad z_2 \equiv \frac{2\kappa'}{3L(1-\gamma)} + D\cdot 2^{-J}\cdot\sqrt{\frac{2\kappa'}{M}}.$$

Moreover, the condition $L \geq 32\kappa'$ implies that $z_1 < 1$ and $\frac{1}{1-z_1} \leq \sqrt{2}$. Thus, on plugging in the above values in (143), we can conclude that

$$\|Q_{\mathcal{H}}^\star - Q^\star\|_\infty \leq \|\overline{Q} - Q^\star\|_\infty \cdot \sqrt{\frac{16\kappa'}{L(1-\gamma)^2}} + \sqrt{\frac{64\kappa'}{L(1-\gamma)^3}} + \frac{2\kappa'\sqrt{2}}{3L(1-\gamma)^2} + \frac{D\cdot 2^{-J}}{(1-\gamma)}\cdot\sqrt{\frac{4\kappa'}{M}}$$

$$\leq \|\overline{Q} - Q^\star\|_\infty \cdot \sqrt{\frac{8\kappa'}{L(1-\gamma)^2}} + \sqrt{\frac{32\kappa'}{L(1-\gamma)^3}} + \frac{2\sqrt{2}\kappa'}{3L(1-\gamma)^2} + \frac{D}{40}, \tag{144}$$

where once again we use the value of $J$ in the last step.

### C.3.3   Proof of Lemma 9

From the iterative update rule in (123), for any agent $m$ we have,

$$Q_{i-\frac{1}{2}}^m - Q_{i-1} = \eta(\widehat{\mathcal{H}}_{i-1}^{(m)}(Q_{i-1}) - Q_{i-1})$$

$$= \eta(\widehat{\mathcal{H}}_{i-1}^{(m)}(Q_{i-1}) - \widehat{\mathcal{H}}_{i-1}^{(m)}(Q_{\mathcal{H}}^\star) + \widehat{\mathcal{H}}_{i-1}^{(m)}(Q_{\mathcal{H}}^\star) - \mathcal{H}(Q_{\mathcal{H}}^\star) + Q_{\mathcal{H}}^\star - Q_{i-1}).$$

Thus,

$$\|Q_{i-\frac{1}{2}}^m - Q_{i-1}\|_\infty \le \eta \left( \|\widehat{\mathcal{H}}_{i-1}^{(m)}(Q_{i-1}) - \widehat{\mathcal{H}}_{i-1}^{(m)}(Q_{\mathcal{H}}^\star)\|_\infty + \|\widehat{\mathcal{H}}_{i-1}^{(m)}(Q_{\mathcal{H}}^\star) - \mathcal{H}(Q_{\mathcal{H}}^\star)\|_\infty + \|Q_{\mathcal{H}}^\star - Q_{i-1}\|_\infty \right)$$

$$\le \eta \left( \gamma\|Q_{i-1} - Q_{\mathcal{H}}^\star\|_\infty + 2\gamma\|\overline{Q} - Q_{\mathcal{H}}^\star\|_\infty + \|Q_{\mathcal{H}}^\star - Q_{i-1}\|_\infty \right)$$

$$= \eta \left( (1+\gamma)\|Q_{i-1} - Q_{\mathcal{H}}^\star\|_\infty + 2\gamma\|\overline{Q} - Q_{\mathcal{H}}^\star\|_\infty \right)$$

$$\le \eta\|\overline{Q} - Q_{\mathcal{H}}^\star\|_\infty \left( \frac{7}{6} \cdot (1+\gamma) + 2\gamma \right) + \frac{\eta D(1+\gamma)}{70},$$

holds with probability $1 - \frac{\delta}{2KI}$. Here, the second inequality follows from (131) and (132), The last step in the above relation follows from (135) evaluated at a general value of $i$ and the prescribed value of $J$. By a union bound argument, the above relation holds for all $i$ with probability at least $1 - \frac{\delta}{2K}$.

## D   Numerical Experiments

In this section, we corroborate our theoretical results through simulations. For the simulations, we consider an MDP with 3 states and two actions, i.e., $\mathcal{S} = \{0, 1, 2\}$ and $\mathcal{A} = \{0, 1\}$. The discount parameter is set to $\gamma = 0.9$. The reward and transition kernel of the MDP is based on the hard instance constructed in Appendix B. Specifically, the reward and transition kernel of state 0 is given by the expression in Eqn. (14a). Similarly, the reward and transition kernel corresponding to state 1 and 2 are identical and given by Eqns. (14b) and (14c) with $p = 0.8$.

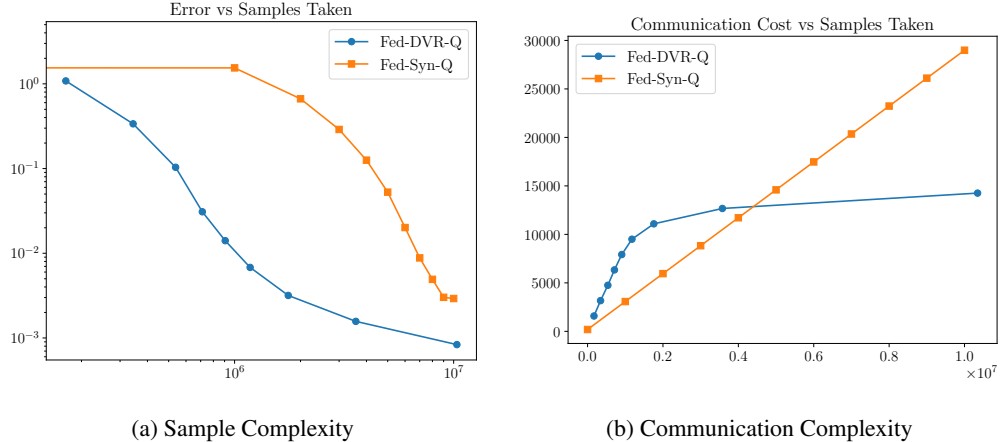

(a) Sample Complexity                    (b) Communication Complexity

Figure 1: Comparison between sample and communication complexities of Fed-DVR-Q and the algorithm Fed-SynQ from Woo et al. [2023].

We perform three empirical studies. In the first study, we compare the proposed algorithm Fed-DVR-Q to the Fed-SynQ algorithm proposed in Woo et al. [2023]. We consider a Federated Q-learning setting with 5 agents. The parameters for both the algorithms were set to the suggested values in the respective papers. Both the algorithms were run with $10^7$ samples at each agent. For the communication cost of Fed-SynQ we assume that each real number is expressed using 32 bits. In Fig 1a, we plot the error rate of the algorithm as a function of the number of samples used. In Fig. 1b we plot the corresponding communication complexities. As evident from Fig 1a, Fed-DVR-Q achieves a smaller error than Fed-SynQ under the same sample budget. Similarly, as suggested by Fig. 1b, Fed-DVR-Q also requires much less communication (measured in terms of the number of bits transmitted) than Fed-SynQ, demonstrating the effectiveness of the proposed approach and corroborating our theoretical results.

In the second study, we examine the effect of the number of agents on the sample and communication complexity of Fed-DVR-Q. We vary the number of agents from 5 to 25 in multiples of 5 and record the sample and communication complexity to achieve an error rate of $\varepsilon = 0.03$. The sample

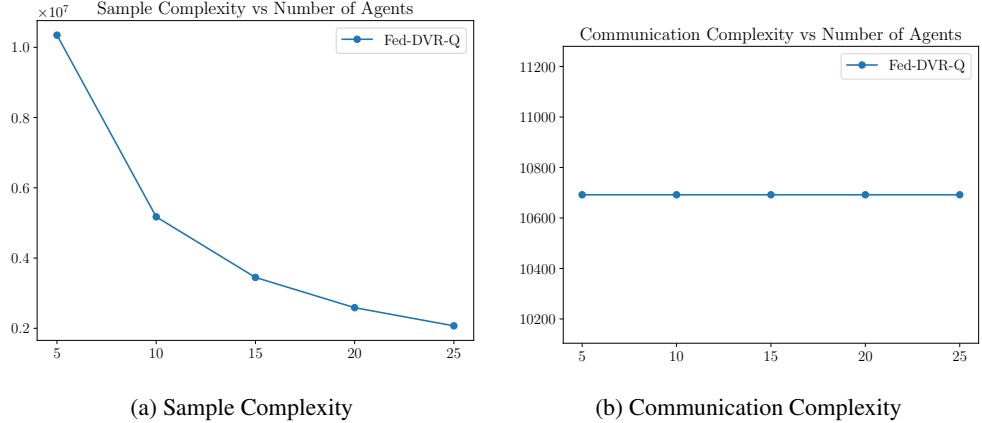

(a) Sample Complexity          (b) Communication Complexity

Figure 2: Dependence of sample and communication complexities of Fed-DVR-Q on the number of agents.

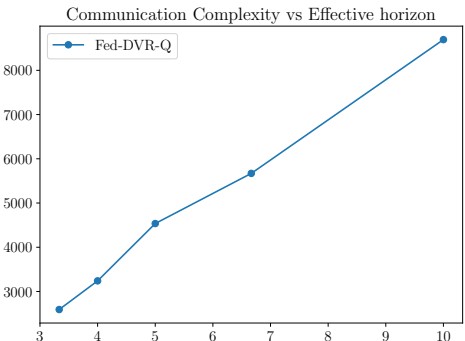

Figure 3: Communication complexity of Fed-DVR-Q as a function of effective horizon, i.e., $\frac{1}{1-\gamma}$.

and communication complexities as a function of number of agents are plotted in Figs. 2a and 2b respectively. The sample complexity decreases as $1/M$ while the communication complexity is independent of the number of agents. This corroborates the linear speedup phenomenon suggested by our theoretical results and the independence between communication complexity and the number of agents.

In the last study, we compare the communication complexity of Fed-DVR-Q as function of the discount parameter $\gamma$. We consider the same setup as in the first study and vary the values of $\gamma$ from $0.7$ to $0.9$ in steps of $0.05$. We run the algorithm to achieve an accuracy of $\varepsilon = 0.1$ with parameter choices prescribed in Sec. 4.1.3. We plot the communication cost of Fed-DVR-Q against the effective horizon, i.e., $\frac{1}{1-\gamma}$ in Fig. 3. As evident from the figure, the communication scales linearly with the effective horizon, which matches the theoretical claim in Theorem 2.

