# OpenReview forum: "The Sample-Communication Complexity Trade-off in Federated Q-Learning"
_NeurIPS.cc/2024/Conference — NeurIPS 2024 oral_

### Official Review · Reviewer_YcCi · 2024-07-10

**Soundness:** 3
**Presentation:** 3
**Contribution:** 4
**Rating:** 8
**Confidence:** 3

**Summary:**

This paper addresses the challenge of Federated Q-learning, focusing on the trade-off between sample complexity and communication complexity. Federated Q-learning involves multiple agents collaboratively learning the optimal Q-function for an infinite horizon Markov Decision Process (MDP) with finite state and action spaces.

The paper proves the lower bound result on the number of rounds, which shows that linear speedup in sample complexity with respect to the number of agents requires at least $\Omega(\frac{1}{1-\gamma})$ rounds of communication.

The second contribution is the algorithm that shows that this bound is tight.

**Strengths:**

The authors consider the problem interesting and well-motivated. The findings establish a good understanding of tradeoffs in Federated Q-learning. The results are well-presented, and the theoretical part of the lower and upper bound looks solid.

**Weaknesses:**

While the main focus of this work is theoretical, the paper could benefit from the experimental evaluation of the algorithm.

**Questions:**

There are also studies on distributed multi-armed bandits, such as 'Parallel Best Arm Identification in Heterogeneous Environments' and 'Communication-efficient Collaborative Best Arm Identification,' which are relevant to Q-learning and RL problems. Could you elaborate on the main differences in techniques used in these studies?

---

> ### Author Rebuttal · Authors · 2024-08-07
>
> Thank you for reviewing our paper and your constructive feedback. We appreciate the time and effort you spent on our paper and the helpful comments you provided. Please find our itemized responses to your questions below.
>
> - _While the main focus of this work is theoretical, the paper could benefit from the experimental evaluation of the algorithm._
>
> Based on your suggestion, we performed some empirical studies and have included the results in the rebuttal. We consider a MDP with 3 states, namely $\{0,1,2\}$ and 2 actions, where the reward and transition kernel of states $0$ and $1$ are identical to those in the MDP outlined in Appendix B.1 in the paper. The reward and transition kernel for state $2$ is identical to that of state $1$. The values of $\gamma$ and $p$ are set to $0.9$ and $0.8$ respectively. We perform two empirical studies. In the first study, we compare the proposed algorithm Fed-DVR-Q to the Fed-SynQ algorithm proposed in Woo et al., 2023. The parameters for both the algorithms were set to the suggested values in the respective papers. As evident from Fig 1, Fed-DVR-Q achieves a smaller error than Fed-SynQ in the same sample budget. Similarly, Fed-DVR-Q also requires much lesser communication (measured in number of bits transmitted) than Fed-SynQ demonstrating the effectiveness of the proposed approach and corroborating our theoretical results. In the second experiment, we study the effect of the number of agents on the sample and communication complexity of Fed-DVR-Q. The sample complexity decreases as $1/M$ demonstrating the linear speed-up while the communication complexity is independent of the number of agents. Both these results confirm our theoretical findings.
>
> Thank you for your suggestion. We will add the empirical results in the final version of our paper.
>
> - _There are also studies on distributed multi-armed bandits, such as 'Parallel Best Arm Identification in Heterogeneous Environments' and 'Communication-efficient Collaborative Best Arm Identification,' which are relevant to Q-learning and RL problems. Could you elaborate on the main differences in techniques used in these studies?_
>
> Thank you for pointing this additional related work. Both the studies mentioned by the reviewer focus on best-arm identification in bandits, which is different problem from learning the optimal Q-function in RL using Q-learning due the markovian structure of the responses and the different objective functions. As a result, both the algorithmic design and analysis are quite different in these papers from those in our work. The lower bound in [1] is established using the heterogeneity across clients. On the other hand, the lower bound in our work is based on the bias-variance trade-off of Q-learning. Similarly, the algorithm design in both [1] and [2] are based on arm-elimination approaches which is different from the variance reduced stochastic fixed point iteration used in our work. We will add a discussion on this in the final version of the paper.
>
> [1] Nikolai Karpov and Qin Zhang, "Parallel Best Arm Identification in Heterogeneous Environments"
>
> [2] Nikolai Karpov and Qin Zhang, "Communication-efficient Collaborative Best Arm Identification"

---

> > ### Author Response · Authors · 2024-08-10
> > **Response to Reviewer**
> >
> > Thank you once again for taking out time to review our paper. We hope that our rebuttal satisfactorily addressed all your concerns. If you have any additional or follow-up questions based on the rebuttal, we would be happy to answer them!

---

> > > ### Comment · Reviewer_YcCi · 2024-08-14
> > >
> > > Thank you for the response.  Yes, it is satisfactory.

---

### Official Review · Reviewer_buwU · 2024-07-13

**Soundness:** 4
**Presentation:** 4
**Contribution:** 3
**Rating:** 8
**Confidence:** 4

**Summary:**

This paper discusses the sample and communication complexity of federated tabular Q-learning. The main contributions can be summarized as follows. First, the paper provides a lower bound on the communication
cost to guarantee a linear speed-up with respect to the number of agents. Then, it proposes a novel  Federated Q-learning algorithm, called Fed-DVR-Q, which simultaneously achieves optimal order sample and communication complexities.

**Strengths:**

S1. The paper provides a lower bound in terms of communication complexity. This would be helpful to the community.

S2. The work provides a novel algorithm that incorporates the variance reduction technique. It is shown that this algorithm has optimal order from both the sample complexity and communication complexity perspectives and achieves a linear speedup in terms of the number of agents.

**Weaknesses:**

W1. Both the lower and upper bounds only apply to the case of synchronous Q-learning with IID samples of the $(s_k, a_k, r_k, s_{k + 1})$ sequence at each agent. Moreover, it only applies to the tabular setup.

**Questions:**

Q1. Can you quantify the benefit of variance reduction in the upper bound? That is, what would the sample complexity be if we looked at a variant of your algorithm without the variance reduction part of the update rule? Doesn't the variance reduction technique typically lead to an improvement in terms of the constants in the sample complexity? I am a bit surprised to see that variance reduction is needed even to guarantee **order** optimal sample and communication complexities.

Q2. Typically, to get convergence to an $\epsilon$-neighbourhood, the stepsize $\eta$ should be chosen depending on $\epsilon.$ For example, see Theorem 2 (a) in [BRS18]. However, in your Theorem 2, it appears $\eta$ can be any number in $(0, 1).$ This seems a bit surprising. Can you elaborate on why that is the case? Am I overlooking something?

[BRS18]: Bhandari, J., Russo, D. and Singal, R., 2021. A Finite Time Analysis of Temporal Difference Learning with Linear Function Approximation. Operations Research, 69(3), pp.950-973.

Q3. Also, it is unclear to me why you claim that your upper bound matches the lower bound. The lower bound is in terms of $N,$ while the upper bound is in terms of $\epsilon?$ Can you formally show that the two orders match?

Q4. Finally, assuming your lower and upper bounds match, can you explain whether the proposed Fed-DVR-Q is a parameter-free algorithm? That is, does it need any knowledge of the unknown parameters of the underlying MDP to achieve order-optimal sample complexity?

I would be happy to increase my score based on your response to my above question.

**Limitations:**

Yes, the authors have discussed the limitations of their work.

---

> ### Author Rebuttal · Authors · 2024-08-06
>
> Thank you for reviewing our paper and your constructive feedback. We appreciate the time and effort you spent on our paper and the helpful comments you provided. Please find our itemized responses to your questions below.
>
> Q1. Can you quantify the benefit of variance reduction in the upper bound? That is, what would the sample complexity be if we looked at a variant of your algorithm without the variance reduction part of the update rule? Doesn't the variance reduction technique typically lead to an improvement in terms of the constants in the sample complexity? I am a bit surprised to see that variance reduction is needed even to guarantee order optimal sample and communication complexities.
>
> A1. For Q-learning, it is well-known that some form of variance reduction presents in all existing algorithm designs that achieve the optimal sample complexity with respect to $\gamma$. In [LCCWC23], the authors demonstrated that vanilla Q-learning, i.e., without variance reduction, has a sample complexity that is necessarily sub-optimal by a factor of $1/(1-\gamma)$. Thus, some form of variance reduction is crucial to achieve the optimal sample complexity. In absence of variance reduction, our algorithm will achieve a sample complexity that is greater than the current one by a factor of $1/(1-\gamma)$ (along with logarithmic factors).
>
> Q2. Typically, to get convergence to an $\epsilon$-neighbourhood, the stepsize $\eta$ should be chosen depending on $\epsilon.$ For example, see Theorem 2 (a) in [BRS18]. However, in your Theorem 2, it appears $\eta$ can be any number in $(0, 1).$ This seems a bit surprising. Can you elaborate on why that is the case? Am I overlooking something?
>
> A2. The Theorem 2(a) in [BRS18] is true when the learner updates the Q-function (or the parameter $\theta$ in their case) after *each* data point/observation from the environment. In other words, the mini-batch size is $1$. On the other hand, our algorithm takes a mini-batch ($\gg 1$) of samples, collates the information and then updates the Q-function. In both the approaches, the fundamental motivation is to ensure that the variance of stochastic updates is small. In [BRS18], the authors use an update with a large variance and balance it by choosing a small step size. In our work, we allow larger step sizes but require updates with smaller variance (obtained through mini-batching) to ensure updates with low variance.
>
> The dependence of $\varepsilon$ is directly through the number of epochs $K$ and indirectly through the choice of $B$ and $J$, as they depend on the parameter $K$.
>
> Q3. Also, it is unclear to me why you claim that your upper bound matches the lower bound. The lower bound is in terms of $N,$ while the upper bound is in terms of $\epsilon?$ Can you formally show that the two orders match?
>
> A3. The variable $N$ in Theorem 1 is the sample complexity of the algorithm and its relation with the error $\varepsilon$ can be obtained through equations (4) and (5). Our Theorem 1 states that if the number of communication rounds in algorithm is $\mathcal{O}(\frac{1}{1-\gamma})$ (upto logarithmic factors), then the error of the final output is $\Omega(1/\sqrt{N})$, where $N$ is the number of samples taken for each state-action pair at each agent. In other words, in order to obtain $\varepsilon$-optimal Q-function, each agent needs to take at least $\Omega(1/\varepsilon^2)$ samples, i.e., the algorithm offers no linear speed up w.r.t. the number of agents. This is equivalent to saying that if any algorithm is designed such that it only takes $\mathcal{O}(1/M\varepsilon^2)$ samples per agent (or offers _any_ speed-up w.r.t. number of agents), then it must have at least $\Omega(\frac{1}{1-\gamma})$ rounds of communication.
>
> Our Theorem 2 states that our proposed algorithm Fed-DVR-Q is such that it takes $\mathcal{O}(1/M\varepsilon^2)$ samples per state-action pair at each agents and has $\mathcal{O}(\frac{1}{1-\gamma})$ rounds of communication. Note that this order matches that in the statement of the lower bound, thereby establishing the optimality of communication complexity. The optimality of the sample complexity follows immediately from the lower bound in (Azar et al. 2013).
>
> Q4. Finally, assuming your lower and upper bounds match, can you explain whether the proposed Fed-DVR-Q is a parameter-free algorithm? That is, does it need any knowledge of the unknown parameters of the underlying MDP to achieve order-optimal sample complexity?
>
> A4. Our algorithm Fed-DVR-Q is parameter-free in the sense that it does not require any knowledge of the parameters of underlying MDP. Our algorithm, however, does have several parameters, whose values have been specified in Sec. 4.1.3. of the paper. We also use a hyperparameter $\eta \in (0,1)$ corresponding to the step size of the updates. As evident from the bounds in Theorem 2, it is preferable to have values of $\eta$ close to $1$.
>
> [LCCWC23]: G. Li, C. Cai, Y. Chen, Y. Wei, and Y. Chi. Is q-learning minimax optimal? a tight sample complexity analysis. Operations Research, 2023.

---

> > ### Author Response · Authors · 2024-08-10
> > **Response to Reviewer**
> >
> > Thank you once again for taking out time to review our paper. We hope that our rebuttal satisfactorily addressed all your concerns. If you have any additional or follow-up questions based on the rebuttal, we would be happy to answer them!

---

### Official Review · Reviewer_UVy4 · 2024-07-15

**Soundness:** 3
**Presentation:** 3
**Contribution:** 3
**Rating:** 7
**Confidence:** 3

**Summary:**

This paper investigates the sample and communication complexities of federated Q-learning with intermittent central aggregation of the Q-value function.
The authors demonstrate that to achieve any speedup in sample complexity through federated collaboration, the communication complexity must be at least $\Omega(1 /(1-\gamma ))$.
Additionally, the paper introduces a novel federated Q-learning algorithm incorporating variance reduction and minibatching techniques, achieving order-optimal sample and communication complexities.

**Strengths:**

- This paper considers the important trade-off problem between sample complexity speedup and communication cost in federated reinforcement learning. It provides a complete characterization of this trade-off in federated Q-learning, including the communication cost in bits.
- Not only do the authors provide a complete characterization of the sample-communication trade-off and design a novel federated Q-learning algorithm that achieves order-optimal sample and communication complexities, but they also provide insights and intuitions into how infrequent communication fails to speed up sample complexity, and how their algorithm balances this trade-off.

**Weaknesses:**

- Several papers report that a _one-shot_ average is sufficient to achieve linear speedups in federated reinforcement learning (FedRL) [1,2]. This seems to contrast starkly with the authors' claim that infrequent communication does not speed up sample complexity. Could the authors clarify this discrepancy and discuss how their work relates to these results?
- The related work section on Distributed RL is generally comprehensive; however, it omits some recent works on heterogeneous FedRL [3,4].
- There is a lack of discussion on the technical difficulties and novelties of the proposed approach. The authors mentioned that Theorem 1 is inspired **by** the analysis of single-agent Q-learning [5] and Theorem 2 is based on the analysis of variance-reduced Q-learning [6]. The authors should elaborate on how their analysis differs from the single-agent case, what new challenges arise in the federated setting, and what novel techniques are employed to overcome these challenges. Additionally, can these techniques be generalized to other settings?
- In Eq. (7), should $\widehat{\mathcal{T}}$  be $\mathcal{T}$? Otherwise it is not defined.

Overall, I am satisfied with the paper, with the first point being my primary concern. I would be happy to raise my score if the authors provide satisfactory clarifications on the above points.

### References

[1] Liu, R., & Olshevsky, A. (2023). Distributed TD (0) with almost no communication. _IEEE Control Systems Letters_, _7_, 2892-2897.
[2] Tian, H., Paschalidis, I. C., & Olshevsky, A. (2024). One-Shot Averaging for Distributed TD (λ) Under Markov Sampling. _IEEE Control Systems Letters_.
[3] Xie, Z., & Song, S. (2023). FedKL: Tackling data heterogeneity in federated reinforcement learning by penalizing KL divergence. _IEEE Journal on Selected Areas in Communications_, _41_(4), 1227-1242.
[4] Zhang, C., Wang, H., Mitra, A., & Anderson, J. (2024). Finite-time analysis of on-policy heterogeneous federated reinforcement learning. In _International Conference on Learning Representations_. PMLR.
[5] Li, G., Cai, C., Chen, Y., Wei, Y., & Chi, Y. (2024). Is Q-learning minimax optimal? a tight sample complexity analysis. _Operations Research_, _72_(1), 222-236.
[6] Wainwright, M. J. (2019). Variance-reduced $ Q $-learning is minimax optimal. _arXiv preprint arXiv:1906.04697_.

**Questions:**

- While I understand that the authors focus on homogeneous i.i.d. data to highlight the main ideas, could the authors comment on the difficulty of generalizing their results to the asynchronous sampling setting? Specifically, the authors mention that the lower bound applies to the asynchronous setting. A similar remark on Theorem 2 would be helpful.

Please see Section Weaknesses for other questions.

**Limitations:**

Limitations and open problems are adequately discussed.

---

> ### Author Rebuttal · Authors · 2024-08-06
>
> Thank you for reviewing our paper and your constructive feedback. We appreciate the time and effort you spent on our paper and the helpful comments you provided. Please find our itemized responses to your questions below.
>
> - _Several papers report that a one-shot average is sufficient to achieve linear speedups in federated reinforcement learning (FedRL) [1,2]. This seems to contrast starkly with the authors' claim that infrequent communication does not speed up sample complexity. Could the authors clarify this discrepancy and discuss how their work relates to these results?_
>
> Our result does not violate the results obtained in the existing studies on federated TD learning [1,2]. The key difference here is that the results in [1,2] consider TD learning to learn the value function directly instead of Q-function. If TD-learning is used to learn the value function directly, then the resultant algorithm aims to learn the fixed point of the operator $\mathcal{T}\_{TD} : \mathbb{R}^{|\mathcal{S}|} \to \mathbb{R}^{|\mathcal{S}|}$ given as $\mathcal{T}\_{TD}(V)(s) := r(s) + \sum_{s'} P(s,s')V(s')$, where $r$ and $P$ are reward and probability transition matrices respectively. Note that this is a **linear** function of $V$. On the other hand, our work focuses on learning the optimal Q-function via Q-learning. Specifically, we learn the fixed point of the operator $\mathcal{T}\_{QL} : \mathbb{R}^{|\mathcal{S}| |\mathcal{A}|} \to \mathbb{R}^{|\mathcal{S}| |\mathcal{A}|}$ given by $\mathcal{T}\_{QL}(Q)(s,a) := r(s,a) + \sum_{s'} P(s'|s,a) [\max\_{a'} Q(s',a')]$. Note that this function is a **non-linear** function of $Q$.
>
> The difference in the communication requirement stems from the fact that the linearity of the Bellman operator in terms of the value allows one-shot averaging to be sufficient to achieve optimal error rates. On the other hand, the non-linearity of Bellman operator with respect to the Q-function results in one-shot averaging to be no longer sufficient to achieve optimal error rates.
>
> If the operator whose fixed point is to be found is linear in the decision variable (e.g., the value function in TD learning) then the fixed point update only induces a variance term corresponding to the noise. However, if the operator is non-linear, then in addition to the variance term, we also obtain a *bias* term in the fixed point update. While the variance term can be controlled with one-shot averaging, more frequent communication is necessary to ensure that the bias term is small enough. A discussion regarding this difference between TD learning and Q-learning can also be found in [5] (from your comment) where the authors show that TD learning achieves the optimal sample complexity but Q-learning (without variance reduction) is necessarily sub-optimal in terms on dependence on $\gamma$.
>
> Thank you for highlighting this interesting point. We will add this discussion in the revised version of the paper.
>
> - _The related work section on Distributed RL is generally comprehensive; however, it omits some recent works on heterogeneous FedRL [3,4]._
>
> Thank you for pointing out the additional papers. We will add them in the related work section. [3] adopts a policy optimization perspective, which is different from the Q-learning paradigm considered in this work. Moreover, the algorithm in [3] obtains a linear communication cost, which is worse than that obtained in our work. Similarly, [4] focuses on on-policy learning and incurs a communication cost that depends polynomially on the required error $\varepsilon$.

---

> ### Author Response · Authors · 2024-08-06
> **Rebuttal by Authors contd.**
>
> - _There is a lack of discussion on the technical difficulties and novelties of the proposed approach. The authors mentioned that Theorem 1 is inspired by the analysis of single-agent Q-learning [5] and Theorem 2 is based on the analysis of variance-reduced Q-learning [6]. The authors should elaborate on how their analysis differs from the single-agent case, what new challenges arise in the federated setting, and what novel techniques are employed to overcome these challenges. Additionally, can these techniques be generalized to other settings?_
>
> One of the challenges in federated setting over the results in [5] and [6] is the design of the communication schedule and its interplay with the convergence of the algorithm to the optimal Q-function. We mention that our analysis is inspired by [5] as we use the same hard instance used in that work and the authors in [5] also use bias-variance trade-off to establish sub-optimality of sample complexity of Q-learning. However, in terms of establishing our lower bound, none of the lemmas from the analysis of single agent Q-learning in [5] can be trivially adopted for the federated learning scenario considered in our work as behaviour of all agents affects that of the others. This requires us to establish all technical results from scratch. Moreover, the communication schedule directly affects the bias-variance trade-off in the federated setting which needs to be carefully analyzed and balanced. Establishing the impact of communication on this trade-off is central to establishing the lower bound in our work. In our analysis, we establish how the time interval between two communication rounds affects the bias and the variance terms. This allows us to show that infrequent communication results in a higher bias term preventing linear speed-up with the number of agents. This analyses and conclusions are completely novel compared to the single-agent analysis in [5] especially because there is no communication involved in single-agent setting and the focus of their analysis is on establishing the sample complexity. Furthermore, the interplay of communication and bias-variance trade-off can be used to conclude communication bounds for more general problems of distributed stochastic fixed point iteration, specifically with non-linear operators. For example, a similar analysis yields that distributed optimization of strongly convex functions using SGD requires a communication cost proportional to the condition number of the function. Thus, the proposed techniques in this work have implications beyond RL.
>
> A direct extension of the algorithm in [6] to the federated setting results in a sub-optimal sample and communication complexities similar to (Woo et al, 2023). The novelty in our work to show that using minibatching as opposed local updates helps manage the bias-variance trade-off (referred to the in lower bound) much better enabling us to achieve optimal sample and communication complexities. As mentioned earlier, this observation carries forward to other distributed stochastic fixed point iteration problems, thereby providing a template to design algorithms that operate at the optimal point in sample-communication complexity trade-off curve. Since our algorithm design is different from that in [6], we need to derive newer results to establish Theorem 2. Lastly, the impact of communication and quantization prevents us from directly adopting the results in [6] and requires a more careful and novel analysis to ensure the optimal convergence rates.
>
> Thank you for pointing this out. We will add a discussion based on these lines in the final version of this paper.
>
>
> - _In Eq. (7), $\hat{\mathcal{T}}$ should be $\mathcal{T}$? Otherwise it is not defined_.
>
> That is correct. Thank you pointing it out. We will fix the typo.

---

> ### Author Response · Authors · 2024-08-06
> **Rebuttal by Authors contd.**
>
> - _While I understand that the authors focus on homogeneous i.i.d. data to highlight the main ideas, could the authors comment on the difficulty of generalizing their results to the asynchronous sampling setting? Specifically, the authors mention that the lower bound applies to the asynchronous setting. A similar remark on Theorem 2 would be helpful._
>
> It is reasonably straightforward to extend the results to the asynchronous sampling setting. At a high level, note that after a burn-in period depending on the mixing time of the behaviour policy, the state visitation distribution will be close to the true stationary distribution. From hereon, we can run the algorithm almost as is with a different choice of mini-batch sizes and recentering sample size parameters. Note that the only difference in the asynchronous setting as compared to the generative model will be that the number of samples for each state-action pair will depend on the behaviour policy. By an appropriate choice of mini-batch sizes and recentering sample sizes, we can ensure that the error still decreases by a factor of 2 every epoch. Consequently, similar conclusions on the sample and communication complexity will also hold for the asynchronous setting. We would also like to point out that the sample complexity will be inversely proportional to the minimum *average* state-action visitation probability, similar to (Woo et al, 2023). As shown in that work, that is the best one can hope for and does not require each agent to cover all state-action pairs.
>
> Thank you for you helpful question. We will also add a discussion on this in the final paper.

---

> > ### Author Response · Authors · 2024-08-10
> > **Response to Reviewer**
> >
> > Thank you once again for taking out time to review our paper. We hope that our rebuttal satisfactorily addressed all your concerns. If you have any additional or follow-up questions based on the rebuttal, we would be happy to answer them!

---

> > ### Comment · Reviewer_UVy4 · 2024-08-12
> >
> > I thank the authors for the comprehensive rebuttal and am content with the clarifications, discussions, and promised revisions. I have raised my score and will support the acceptance of this work.

---

### Author Rebuttal · Authors · 2024-08-07

Based on the suggestion by Reviewer YcCi, we have performed two empirical studies and included their results in the attached PDF. We refer the reader to the response to Reviewer YcCi for additional details about the experiments and a discussion of the results.

---

### Decision · Program_Chairs · 2024-09-25

**Decision:**

Accept (oral)

**Comment:**

The paper explores the trade-off between sample complexity and communication complexity in federated Q-learning. The authors focus on a specific yet intuitive class of algorithms characterized by intermittent communication, where each agent independently runs the Q-learning algorithm and engages in scheduled communication rounds to exchange their estimated Q-functions. These algorithms are defined by their step-sizes and communication schedules. For this class, the authors derive a lower bound on communication complexity necessary to achieve a given speed-up in sample complexity (Theorem 1). They also introduce an algorithm that achieves a sample complexity speed-up by a factor of $M$ (the number of agents), with a communication complexity of $\tilde{O}(1/(1-\gamma))$. Notably, these performance guarantees align with the established lower bound.

The reviewers unanimously recognize the significance and depth of the results. They have also provided several suggestions to enhance the clarity and depth of the paper through additional discussions. Please consider these suggestions when preparing the camera-ready version of your paper. Congratulations on this very nice paper.